# Biomass burning sources control ambient particulate matter but traffic and industrial sources control volatile organic compound emissions and secondary pollutant formation during extreme pollution events in Delhi

Arpit Awasthi[1], Baerbel Sinha[1], Haseeb Hakkim[1], Sachin Mishra[1], Varkrishna Mummidivarapu[1], Gurmanjot Singh[1], Sachin D. Ghude[2], Vijay Kumar Soni[3], Narendra Nigam[3], Vinayak Sinha[1], Madhavan N. Rajeevan[4]

[1]Department of Earth and Environmental Sciences, Indian Institute of Science Education and Research Mohali, Sector 81, S.A.S Nagar, Manauli PO, Punjab, 140306, India
[2]Indian Institute of Tropical Meteorology, Pashan, Pune 411008, Ministry of Earth Sciences, India
[3]India Meteorological Department, New Delhi 110003, Ministry of Earth Sciences, India
[4]Ministry of Earth Sciences, Government of India, New Delhi 110003, India

*Correspondence to*: Baerbel Sinha (bsinha@iisermohali.ac.in)

**Abstract.** Volatile organic compounds (VOCs) and particulate matter (PM) are major constituents of smog. Delhi experiences severe smog during post-monsoon season, but a quantitative understanding of VOCs and PM sources is still lacking. Here, we source-apportioned VOCs and PM, using a high-quality recent (2022) dataset of 111 VOCs, $PM_{2.5}$, and $PM_{10}$ in a positive matrix factorization (PMF) model. Contrasts between clean-monsoon and polluted-post-monsoon air, VOC source fingerprints, and molecular-tracers, enabled differentiating paddy-residue burning from other biomass-burning sources, which
has hitherto been impossible. Fresh paddy-residue burning and residential heating & waste-burning contributed the highest to observed $PM_{10}$ (25% & 23%), $PM_{2.5}$ (23% & 24%), followed by heavy-duty CNG-vehicles 15% $PM_{10}$ and 11% $PM_{2.5}$. For ambient VOCs, ozone, and SOA formation potentials, top sources were petrol-4-wheelers (20%, 25%, 30%), petrol-2-wheelers (14%, 12%, 20%), industrial emissions (12%, 14%, 15%), solid fuel-based cooking (10%, 10%, 8%) and road construction (8%, 6%, 9%). Emission inventories tended to overestimate residential-biofuel emissions at least by a factor of 2, relative to
the PMF output. The major source of PM pollution was regional biomass burning, whereas traffic and industries governed VOC emissions and secondary pollutant formation. Our novel source-apportionment method quantitatively resolved even similar biomass and fossil-fuel sources, offering insights into both VOC and PM sources affecting extreme-pollution events. It represents a notable advancement over current source apportionment approaches, and would be of great relevance for future studies in other polluted cities/regions of the world with complex source mixtures.

# 1 Introduction

The Delhi National Capital Region (NCR) is located in the Indo-Gangetic plains and experiences some of the highest air pollution events worldwide, exposing its inhabitants to hazardous air quality. New Delhi had the world's highest population-weighted annual average $PM_{2.5}$ exposures of 217.6 µg m$^{-3}$ and the sixth-highest $PM_{2.5}$-attributable death (85 deaths per lakh) (Pandey et al., 2021). India is currently among the world's foremost developing countries and Delhi being its capital has witnessed rapid population growth and urbanization in the past decade, but a significant fraction of the population still lacks access to cleaner technologies for cooking and heating (Thakur M. 2023; Fadly et al., 2023). Delhi with a population of 31.7 million people (UN World Population Prospects 2022), sees an addition of over six hundred thousand vehicles per year (2022 VAHAN-Ministry of Road Transport and Highways (MoRTH), Government of India). The sources of air pollutants over the region have received much attention recently and a number of source apportionment methods have been applied. Several studies have relied on chemical mass balance models (CMB) that are unable to sniff out unknown fugitive sources since their application rests on prior knowledge of all relevant sources and their source profiles (Prakash et al., 2021; Srivastava et al., 2008). Clearly, in a dynamic developing world megacity like Delhi, where wide disparities exist in terms of access to clean energy and waste burning, and many other activities continue to be carried out by the informal sector, the CMB approach may misattribute emissions only to known sources, with no possibility of identifying other major sources that may be active. While much information has come to light through previous aerosol mass spectrometry-based source apportionment studies, a key limitation of the previous studies has been an inability to distinguish between different similar types of fossil fuel and biomass-burning sources (Kumar et al., 2022; Mishra et al., 2023). The VOC source-fingerprints of many combustion sources are well constraint and understood, and have recently been used in PMF-based studies to source apportion co-emitted greenhouse gasses such as methane, $CO_2$ and $N_2O$ (Guha, et al. 2015; Assan et al. 2018; Schulze et al. 2023). We now extend the use of this promising new technique towards source-apportionment of co-emitted $PM_{2.5}$ and $PM_{10}$. This helps us overcome another major limitation of existing studies which has been the piece-meal approach where either VOCs (Jain et al., 2022) or PM or a subset thereof have been investigated, that too only on datasets that were acquired in 2019 or earlier, i.e. pre-COVID19 period after which significant changes have been implemented. For example, the Bharat Stage VI which complies with the Euro VI norms was implemented in 2018 in Delhi and 2019 for Delhi NCR (Gajbhiye et al., 2023). This significant decision was prompted by the severe air pollution challenges faced by Delhi, particularly worsening around 2019 (Gajbhiye et al., 2023). Still air pollution continues to pose major health risks. Overall, a continued lack of strategic knowledge and inability to pinpoint the exact sources and their contribution, hampers efforts to propose evidence-based strategies for mitigation of major sources. In our previous studies from another site in the Indo-Gangetic Plain (Pallavi et al., 2019; Singh et al., 2023), we demonstrated that source apportionment carried out by PMF when combined with measured VOC chemical fingerprints of sources, can distinguish and quantify the contribution of even similar types of sources (e.g. within traffic source: to distinguish 4-wheelers from 2-wheelers and diesel vehicles; and within biomass burning sources to distinguish paddy stubble burning from residential biofuel combustion). We improve upon those studies that were carried out on datasets acquired using a unity mass resolution

VOC proton transfer reaction mass spectrometer by recent new data acquired using the latest state-of-the-art enhanced volatile range high mass resolution and high sensitivity PTR-TOF-10 K technology over Delhi (Mishra et al., 2024).

The dataset used for source apportionment in this study using the positive matrix factorization modelling includes the high sensitivity (few ppt), high mass resolution (>10000) real-time acquisition of 111 speciated volatile organic compounds measured (15th August 2022–26th November 2022) using a Proton Transfer Reaction Time of Flight Mass Spectrometer 10 K (PTR-TOF10K-MS) instrument in Delhi, along with hourly averaged $PM_{2.5}$ and $PM_{10}$ measurements. This dataset is novel in that it contains all major known gas phase molecular tracers for varied sources and VOC profiles of major agricultural and

urban sources extant over Indo-Gangetic Plain. The dataset covered the relatively cleaner monsoon season which provides a baseline air pollution over the city and the post-monsoon season when post-harvest agricultural paddy residue burning in the Indo-Gangetic Plain perturbs the atmospheric chemical composition by providing an additional source of VOC and PM emissions. This comprehensive approach ensured that the positive matrix factorization model, which provides the advantage of determining air pollution sources without any prior knowledge of the source fingerprints, was able to quantify the

contribution of different sources to the ambient VOC, $PM_{2.5}$, and $PM_{10}$ mass concentrations reliably as its solutions are sensitive to contrasts in ambient time series data. The statistical solution obtained using the model were verified against real-world measured source profiles from the region and thus presents a significant advancement over previous PMF source apportionment studies reported from the Delhi-NCR region. Furthermore, by combining this molecular tracer-based methodology and analyses with additional air mass back trajectory and statistical analyses, we also constrain the location of

the major pollution sources and regions and compare the results of our source apportionment study with two widely used gridded emission inventories in chemical transport models, namely the Emission Database for Global Atmospheric Research (EDGARv6.1) (Crippa et al., 2022), and the Regional Emission inventory in Asia (REAS v3.2.1 (Kurokawa & Ohara, 2020).

## 2 Methodology

### 2.1 Measurement site and meteorological conditions

The new PTR-TOF-MS 10 K enhanced volatility range mass spectrometer, as well as the primary VOC dataset and site, have already been described and analyzed in detail in the companion paper (Mishra et al., 2024). Hence only a brief description of these aspects and complementary aspects such as the air mass flow trajectories at the site during the study period from August 2022 to November 2022 are provided below.

Ambient air was sampled into the instruments from the roof-top of a tall building (28.5896°N-77.2210°E) at ~35 m above

ground, located within the premises of the Indian Meteorological Department (IMD) at Lodhi Road, New Delhi situated in Central Delhi. The sampling site is a typical urban area surrounded by green spaces, government offices, and residential areas, but not in the direct vicinity of any major industries (Fig. S1) and representative of the airflow patterns observed in Delhi seasonally. Figure 1 shows the location of the site and the 120 h back trajectories of air masses arriving at the site that were grouped according to the dominant synoptic regional scale transport into a) south-westerly (orange and yellow) flows carrying

emissions from southern Punjab, Haryana, Uttar Pradesh, Madhya Pradesh, Rajasthan and Gujarat towards the receptor, b) north-westerly (light and dark blue) flows carrying emissions from Pakistan Punjab, Indian Punjab, Haryana, Western Uttar Pradesh, Himachal Pradesh, and Uttarakhand towards the receptor, and c) south-easterly flows (light and dark red) carrying emissions from Haryana, Southern Uttarakhand, Uttar Pradesh, Bihar and Nepal towards the receptor. Figure 1d shows a Google Earth image with a spatial map of the daily fire counts in the region for the post-monsoon season alongside with the maximum 24-h fetch region for each of these synoptic flow situations marked by coloured square. Figure 1e-h shows the e) photosynthetic active radiation, f) daily fire counts in the fetch region (21-32°N, 72-88°E), g) temperature and relative humidity, and h) the ventilation coefficient and the sum of the daily rainfall during the study period (15th August 2022– 26th November 2022). Wind speed, wind direction, ambient temperature, relative humidity, and photosynthetic active radiation were measured using meteorological sensors (Campbell Scientific portable sensors equipped with CS215 RH and temperature sensor, PQS1 PAR sensor, TE525-L40 v rain gauge, Campbell Scientific Inc.). Boundary layer height was taken from the ERA5 dataset (Hersbach et al., 2023) and the ventilation coefficient was calculated as the product of the measured wind speed and boundary layer height. Fire counts were obtained using the Visible Infrared Imaging Radiometer Suite (VIIRS) 375 m thermal anomalies / active fire product data from the VIIRS sensor aboard the joint NASA/NOAA Suomi National Polar-orbiting Partnership (Suomi NPP) and NOAA-20 satellites, for high and normal confidence intervals only. The back trajectories in Fig. 1 showing the 5-day runs were obtained using Hysplit Desktop, version 5.2.1 (Stein et al., 2015; Rolph et al., 2017) with GFSv1 0.25° resolution meteorological fields as input data. The model was initialized every 3 hours (0, 3, 6, 9, 12, 15, 18, and 21 UTC) at 50 m above ground level for the year 2022 and trajectories were subjected to back trajectory cluster analysis via k-means clustering (Bow, 1984) with Euclidean distance metrics using the open-air package (v2.11, Carslaw & Ropkins, 2012). Three basic air transport situations occur at this site, namely from the South West (Fig. 1a), North-West (Fig. 1b), and South-East (Fig. 1c). These regional transport situations in the shared air-shed have been described for another receptor site located 300 km north of Delhi previously in great detail (Pawar et al, 2015). At Delhi, each of these large-scale flow patterns can occur with three different transport speeds; fast (darkest colour), medium (intermediate colour) and slow (lighter colour), resulting in 9 clusters.

During the monsoon season (15.08-30.09.2022), the air masses from the south-west direction (western arm of the monsoon) were more prevalent than air masses reaching the site from the south-east (Bay of Bengal arm of the monsoon). During the post-monsoon season (01.10-26.11.2022) air masses remain confined over the NW-IGP for prolonged periods and primarily reach the site from the north-west (Fig. 1b), except during the passage of western disturbances (05.10-10.10.2022 and 04.11-10.11.2022), which result in brief periods with south westerly and south-easterly flow and rain (Fig. 1h). Figure 1f shows that paddy residue burning of short-duration varieties commences even before the monsoon withdrawal on 29th September 2022, however, the burning peaks during the harvest of late varieties in late October and early November. During this period a drop in temperature (Fig. 1g) and increased fire activity (Fig. 1f) results in the build-up of a persistent haze layer leading to suppressed photosynthetically active radiation (Fig. 1e). This is associated with prolonged periods of poor ventilation (Fig. 1h).

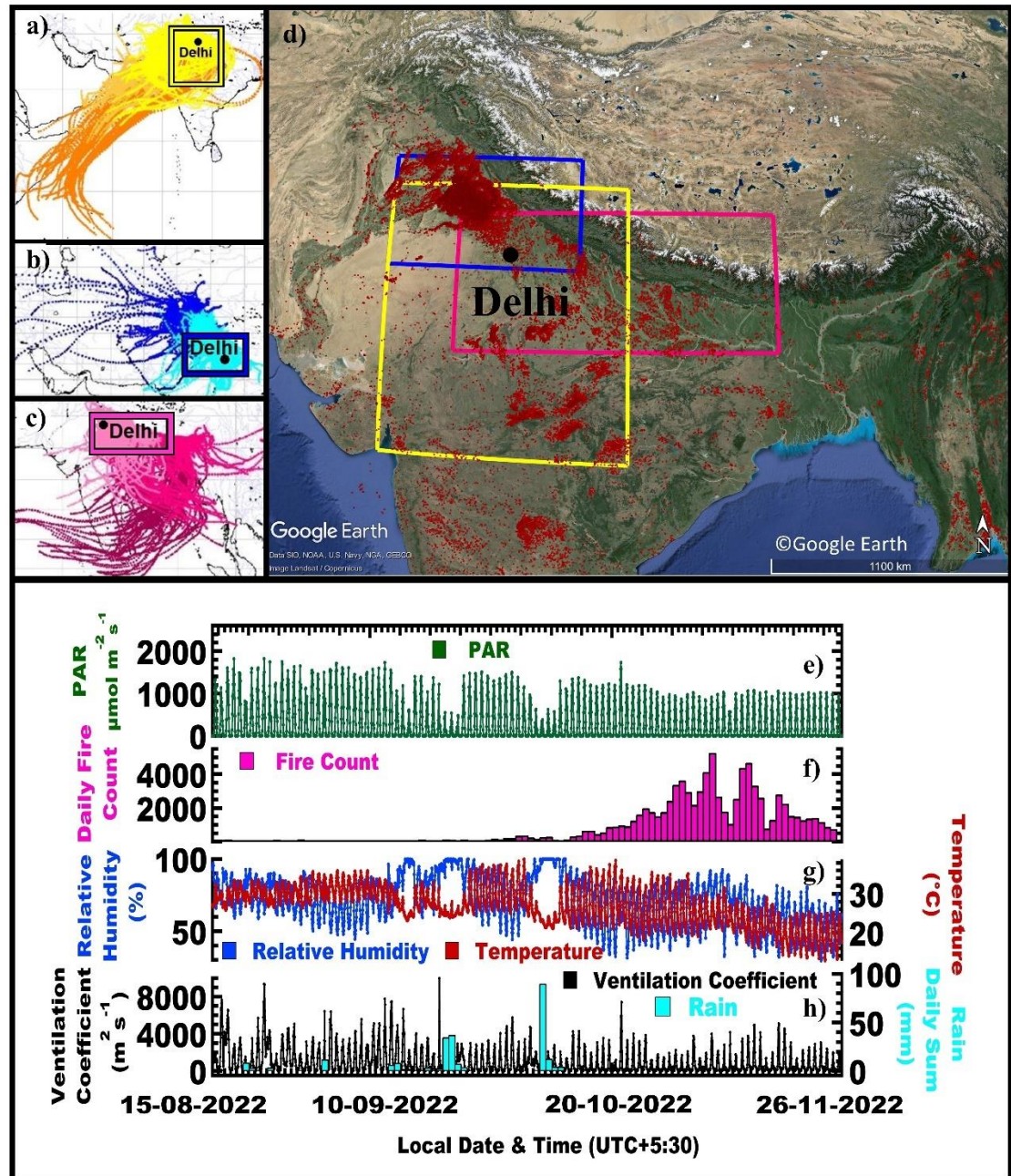

**Figure 1: 120 h back trajectory air mass reaching receptor site at Mausam Bhawan building (28.5896°N-77.2210°E, 50 m above ground level) grouped according to the dominant synoptic scale transport into a) South-Westerly, b) North-Westerly, and c) South-Easterly flow. d) spatial map of the daily fire counts in the region for the post-monsoon season with square boxes indicating the fetch region from which air masses typically reach the receptor site within 24 hrs for a given flow situation (© Google Earth). The bottom panels show the e) photosynthetically active radiation (PAR), f) daily fire counts in the fetch region, g) temperature and relative humidity, and h) the ventilation coefficient and the sum of the daily rainfall for the study period.**

**2.2 Measurement of Volatile Organic Compounds, trace gases, PM$_{2.5}$ and PM$_{10}$ mass concentrations**

Measurements of volatile organic compounds were performed using a high mass resolution and high sensitivity proton transfer reaction time of flight mass spectrometer (PTR-TOF10k; model PT10-004 manufactured by Ionicon Analytik GmbH, Austria). Details pertaining to the characterization, calibration, and QA/QC of the acquired dataset have been provided in Mishra et al., 2024. It is worth mentioning again that as a significant improvement over other previous PTR-TOF-MS deployments in Delhi, the inlet system of the instrument used in this work was designed for sampling and detection of low-volatility compounds with the extended volatility range technology (Piel et al., 2021). The inlet system of the instrument as well as the ionization chamber is fully built into a heated chamber and the inlet capillary is furthermore fed through a heated hose to ensure there are no "cold" spots for condensation. The entire inlet system is made of inert material (e.g. PEEK or siliconert treated steel capillaries) to keep surface effects minimal. Further, the overall inlet residence time was less than 3 seconds, throughout the campaign. Compared to previous PTR-TOF-MS instruments deployed in Delhi, this instrument also had unprecedented higher mass resolution (greater than 10000 m/$\Delta$m (FWHM) for m/z $\geq$ 79 Th even reaching as high as 15000 at m/z 330) coupled with high detection sensitivity (~ 1 ppt or better for 60 s averaged data), providing unprecedented ability for identification and quantification of new ambient compounds. Mass spectra were acquired over the m/z 15 to 450 amu range at a frequency of 1 Hz. Table S1 lists information pertaining to m/z, compound names, and sources supported by references to previous studies where available, averaged ambient mass concentrations and classification of the species as weak or strong for the PMF model runs. The accuracy error was minimized by conducting a total of 8 span calibrations throughout the study period. The details of these calibrations can be found in Mishra et al., 2024. The precision error for each m/z listed in Table S1, which needs to be included into the PMF model runs, was calculated from the average observed count rate in counts per second (cps) of each m/z with the help of Poisson statistics. The detection limit was determined as 2$\sigma$ of the noise observed in clean zero air. Thermofisher Scientific 48i (IR filter correlation-based spectroscopy), 43i (pulsed UV fluorescence), 49i (UV absorption photometry), and 42i (chemiluminescence) trace level air quality analyzers were used to quantify carbon monoxide (CO), ozone (O$_3$), NO and NO$_2$, respectively. The overall uncertainty of the measurements was less than 6 %. Measurements of PM$_{2.5}$ and PM$_{10}$ were made using Thermofisher Scientific Model 5014i series which is based on the beta-attenuation technique. Technical details pertaining to QA/QC of these instruments have been comprehensively described in our previous works (Chandra and Sinha, 2016; Kumar et al., 2016; Sinha et al., 2014). Carbon dioxide and methane were measured using a cavity ring down spectrometer (Model G2508, Picarro, Santa Clara, USA). The overall uncertainty of these measurements was below 4 % and technical details pertaining to the instrument are available in Chandra et al., 2018.

**2.3 Positive matrix factorization (PMF) model analysis**

The US EPA PMF 5.0 (Paatero et al., 2002, 2014; Paatero & Hopke, 2009; Noris et al., 2014) was applied to a sample matrix of 2496 hourly observations and 111 VOC species. The species with S/N greater than 2.0 were designated as strong species (94) while others were designated as weak species (17). The total VOC mass was included as a weak species and was calculated

as the sum of the mass of the individual 111 VOC species included in the PMF. Overall, the 111 VOC species included in our analysis and their isotopic peaks explained 86 % of the VOC mass detected during our study period. The remaining 119 m/z

that accounted for 14 % of the detected VOC mass could not be included in our PMF analysis mostly because signals were below the detection limit for close to 50 % of the observation period, or because compound identity could not be confirmed via isotopic peaks. $PM_{2.5}$ and $PM_{10}$ were included as additional weak species in the model. The specified uncertainty for weak species is tripled by the PMF model, to limit the influence of such species on the PMF solution. Several authors have recently pioneered the use of VOC tracers in a PMF to source apportion co-emitted greenhouse gasses such as methane, $CO_2$ and $N_2O$

(Guha, et al. 2015; Assan et al. 2018; Schulze et al. 2023). Since the VOCs source-fingerprints of many combustion sources are well constrained and understood, we now extend the use of this promising new technique towards source-apportionment of co-emitted $PM_{2.5}$ and $PM_{10}$. The PMF is a matrix decomposition factor analysis model that decomposes a time series of measured species into a set of factors with fixed source fingerprints whose contributions to the input data set varies with time. This makes the model well suited to accommodate all chemical species co-emitted from the same source.

The EPA PMF 5.0 is a multivariate factor analysis tool and a receptor model that divides the data matrix $X_{ij}$ (time series of measured concentrations of VOCs with i distinct observations and j measured species) into two matrices, $F_{kj}$ (source fingerprint) and $G_{ik}$ (source contribution), along with a residual matrix, $E_{ij}$, using the simultaneous application of the linear least square method in multiple dimensions.

$$X_{ij}= \sum_{k=1}^{p} G_{ik} \times F_{kj} + E_{ij} \qquad (1)$$

The user must provide the number of variables or sources (k). To determine the number of VOC sources the model can resolve in this atmospheric environment, the model was run with 3 to 12 factors. The model was initiated for 20 base runs with the recommended block size of 379, and the run with the lowest $Q_{robust}$ and $Q_{true}$ was chosen for further analysis and displayed in Figure 2. Figure 2 shows how the percentage of total VOCs, $PM_{2.5}$, and $PM_{10}$, attributable to various sources changes when the number of factors increases from 3 to 12, while Fig. S2-S4 illustrates the evolution in the factor contribution time series,

source profile, and percentage of species explained by different sources when the number of factors in the PMF increases. Figure S5 shows how the $Q_{true}/Q_{theoretical}$ ratio and $Q_{robust}/Q_{theoretical}$, and scaled residuals beyond 3 standard deviations drop exponentially when the number of factors increases. It can be seen that initially the $Q_{true}/Q_{theoretical}$ ratio drops faster than $Q_{robust}/Q_{theoretical}$ ratio on account of additional major plumes being better explained with each additional factor. However, with the increase from 11 to 12 factors both drop in a parallel fashion indicating that the point of diminishing returns has been

reached.

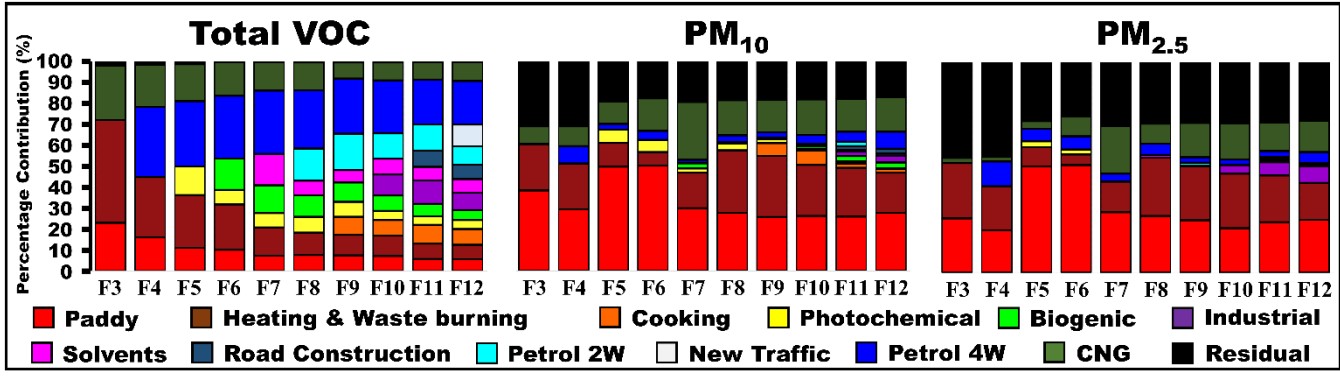

**Figure 2: Percentage of the total VOC, PM$_{10}$ and PM$_{2.5}$ mass explained by each factor in the PMF model output results when the number of PMF factors in the model is increased from 3 to 12. The balance to 100 % shown in black indicates the percentage share of the total mass in the PMF residuals.**

While the three major traffic factors namely; CNG, petrol 4-wheeler, and petrol 2-wheeler are completely resolved with the 8 factors solution, three major biomass-burning related sources namely paddy residue burning, heating and waste burning, and solid fuel-based cooking are separated with a 9-factors solution. Until the PMF opens distinct factors for the OVOC emissions due to industrial solvent usage and stack venting in the 7-factor solution, the partitioning between paddy residue burning and heating and waste burning PM$_{2.5}$ and PM$_{10}$ emissions in the model remains unstable, because these sources with their strong OVOC emissions are most agreeable to accommodating additional OVOC sources in their fingerprint at the expense of explaining the PM$_{2.5}$ and PM$_{10}$ emissions. Once the industrial OVOC emissions have their own factor, this split becomes stable. The amount of PM attributed to residential heating and waste burning stabilizes after a separate factor for cooking emissions opens up in the 9-factor solution. Industrial emissions are separated from solvent usage and other evaporative emissions with a 10-factor solution, and road construction activity emerges as a separate source with an 11-factor solution. While attempting to resolve 12-factors, the model splits transport sector emissions into four separate factors. However, this new transport sector factor shows a time series correlation (R=0.8) with the petrol 4-wheeler factor, and the 12-factor solution was found to be rotationally unstable during bootstrap runs, indicating that the model cannot resolve more than 11 factors with the available VOC tracers. The 12-factor solution also hardly improves the Q$_{robust}$/Q$_{theoretical}$ and Q$_{true}$/Q$_{theoretical}$ ratio (Fig. S5). Therefore, the 11-factor solution was analyzed further. The model was run in the constrained mode elaborately described in Sarkar et al., (2017) and Singh et al., (2023). The rotational ambiguity can be reduced using this option with the aid of prior knowledge by encouraging the model to minimize (pull down) or maximize (pull up) the total mass assigned to specific hourly observations or compounds in source profiles as much as possible within a pre-defined permissible penalty on Q. The primary problem of the base run solutions is that night-time biomass burning plumes contaminate both the biogenic and the photochemical factor. To minimize this in our constrained run, we have pulled down primary emissions (acetonitrile, toluene, C8 aromatics, and C9 aromatics) in the biogenic and photochemical factors. We also pulled down the top-7 strongest nighttime plumes contaminating the biogenic and photochemical factors. In addition, we pulled up the highest plume event for all the anthropogenic emission-related factors as detailed in Table S2. The overall penalty to Q (the object function) was 4.9 %, which is within the

recommended limit of 5 % (Norris et al., 2014; Rizzo & Scheff, 2007). The model uncertainty was assessed using bootstrap

runs. The constrained model was found to be rotationally stable and robust with 100 % of all bootstrap runs for each individual factor mapped onto the base factor with R>0.6 and no unmapped bootstraps.

## 2.4 Calculation of the ozone formation potential, secondary organic aerosol formation and volatility

The contribution of VOCs to ozone production was derived with the maximum incremental reactivity (MIR) (Carter, 2010) method using the following equation

$$OFP = \sum(c_i MIR_i) \tag{2}$$

where $c_i$ is the measured concentration of VOC species i and $MIR_i$ is the maximum incremental reactivity of VOC species i. The secondary organic aerosol production (SOAP) was determined using the following equation

$$SOAP = \sum(c_i SOAP_i) \tag{3}$$

$SOAP_i$ values were calculated with the SOA yields for high $NO_x$ emission environments reported in Table S3 according to the

equation of Derwent et al., (1998; 2010), as Delhi being a megacity is a high $NO_x$ emission environment. This equation evaluates each VOC species' ability to make SOA in relation to the amount of SOA the same mass of toluene would make when introduced to the ambient environment. This is represented by the $SOAP_i$.

The saturation vapour pressure of VOCs was calculated using EPA EPI Suite v4.1 (MPBPWINv.1.43; KOAWIN v.1.00) provided by the US Environmental Protection Agency (US EPA, 2015) according to the method described in Li et al., (2016).

The vapour pressure of liquids and gases is estimated using the average of the Antoine method (Lyman et al., 1990) and the modified Grain method (Lyman 1985). The vapour pressure is then converted to saturation mass concentration $C_0$ in µg m$^{-3}$ using the following equation:

$$C_0 = \frac{M \, 10^6 \, p_0}{760 \, R \, T} \tag{4}$$

wherein M is the molar mass [g mol$^{-1}$], R is the ideal gas constant [8.205 x 10$^{-5}$ atm K$^{-1}$ mol$^{-1}$ m$^3$], $p_0$ is the saturation vapor

pressure [mm Hg], and T is the temperature (K). Organic compounds with $C_0 > 3$ x 10$^6$ µg m$^{-3}$ are classified as VOCs while compounds with $300 < C_0 < 3$ x 10$^6$ µg m$^{-3}$ as Intermediate VOCs (IVOCs).

## 2.5 Comparison of existing emission inventories with PMF derived output

The observational data was grouped according to the predominant airflow into a south-westerly, north-westerly, and south-

easterly group, and the fetch region from which air masses would reach the receptor site within 24 h was determined for each group separately spanning latitude 21–31 ̊N and longitude 72–82 ̊E, latitude 28–32 ̊N and longitude 72–80 ̊E and latitude 25–30 ̊N and longitude 75–88 ̊E, respectively for the three flow regimes. Two gridded emission inventories namely the Emission Database for Global Atmospheric Research (EDGARv6.1) for the year 2018 (Crippa et al., 2022), and the Regional Emission inventory in Asia (REAS v3.2.1) for the year 2015 (Kurokawa & Ohara, 2020) were filtered for these three fetch regions to

compare PMF results with the emission inventory. We compare the relative percentage contribution of sources to the total atmospheric pollution burden in the PMF with the relative percentage contribution of sources to the total emissions for the emission inventories. This approach has been routinely used to evaluate emission inventories with the help of PMF results at different sites around the world (Buzcu-Guven and Fraser, 2008; Morino et al. 2011; Sarkar et al., 2017; Li et. al., 2019; Qin et al., 2022). For the purpose of emission inventory comparison of anthropogenic sources, natural sources such as biogenic

emissions and the photochemistry factor were removed from the PMF output, while the solid fuel-based cooking and residential heating and waste burning emissions were summed up in residential & waste management. In addition, CNG and Petrol 2 & 4-wheeler factors were combined into the consolidated transport sector emissions.

## 3 Results and Discussions

### 3.1 Validation of the PMF output with source fingerprints

Figure 3 shows the source profile of the eleven factors that our PMF analyses resolved. Out of the 111 VOCs only those whose normalized source contribution exceeded 0.1 when divided by the most abundant compound in the same source profile in at least one of the sources, were included in the figure. The source identity of the PMF factors was confirmed by matching the PMF factor profiles with the unit $\mu g\ m^{-3}$ with normalized source fingerprints of grab samples collected from the potential sources. To facilitate the comparison of emission factors and grab samples from different studies with the PMF output, the

source samples were normalized by dividing each species' mass/emission factor by the mass/emission factor of the most abundant species in a given fingerprint. The PMF factor profile matched best against source samples collected from burning paddy fields (R=0.6, Kumar et al., 2020) for the paddy residue burning factor. The cooking factor matched emissions from a cow-dung-fired traditional stove called angithi (R=0.7, Fleming et al., 2018). The residential heating & waste burning factor had a source fingerprint matching emission from leaf litter burning, (R=0.7, Chaudhary et al., 2022), waste burning (R=0.7,

Sharma et al., 2022), and cooking on a chulha fired with a mixture of firewood and cow dung (R=0.9, Fleming et al., 2018). The factors identified as CNG (R=1.0), petrol 4-wheelers (R=0.9), and petrol 2-wheelers (R=0.6) matched tailpipe emissions of the respective vehicle types and fuels (Hakkim et al., 2021). The petrol 4-wheelers (R=0.9), and petrol 2-wheelers (R=0.7) also matched traffic junction grab samples from Delhi (Chandra et al., 2018). The OVOC source fingerprint of the road construction factor matched the source fingerprint of asphalt mixture plants and asphalt paving (R=0.9, Li et al., 2020), while

the hydrocarbon source fingerprint matched diesel-fuelled road construction vehicles (R=0.6, Che et al., 2023). The factors identified as solvent usage and evaporative emissions matched ambient air grab samples collected from an industrial area at Jahangirpuri (R=0.7), and Dhobighat at Akshar Dham (R=0.5) in this study. The factor identified as industrial emissions showed the greatest similarity to ambient air grab samples from the vicinity of the Okhla waste-to-energy plant (R=0.8), Gurugram (R=0.7) and Faridabad (R=0.8) industrial area. The biogenic factor showed the greatest similarity to leaf wounding

compounds released from Populus tremula (R=0.8, Portillo-Estrada et al., 2015) as well as BVOC fluxes from Mangifera indica (R=0.4, Datta et al., 2021).

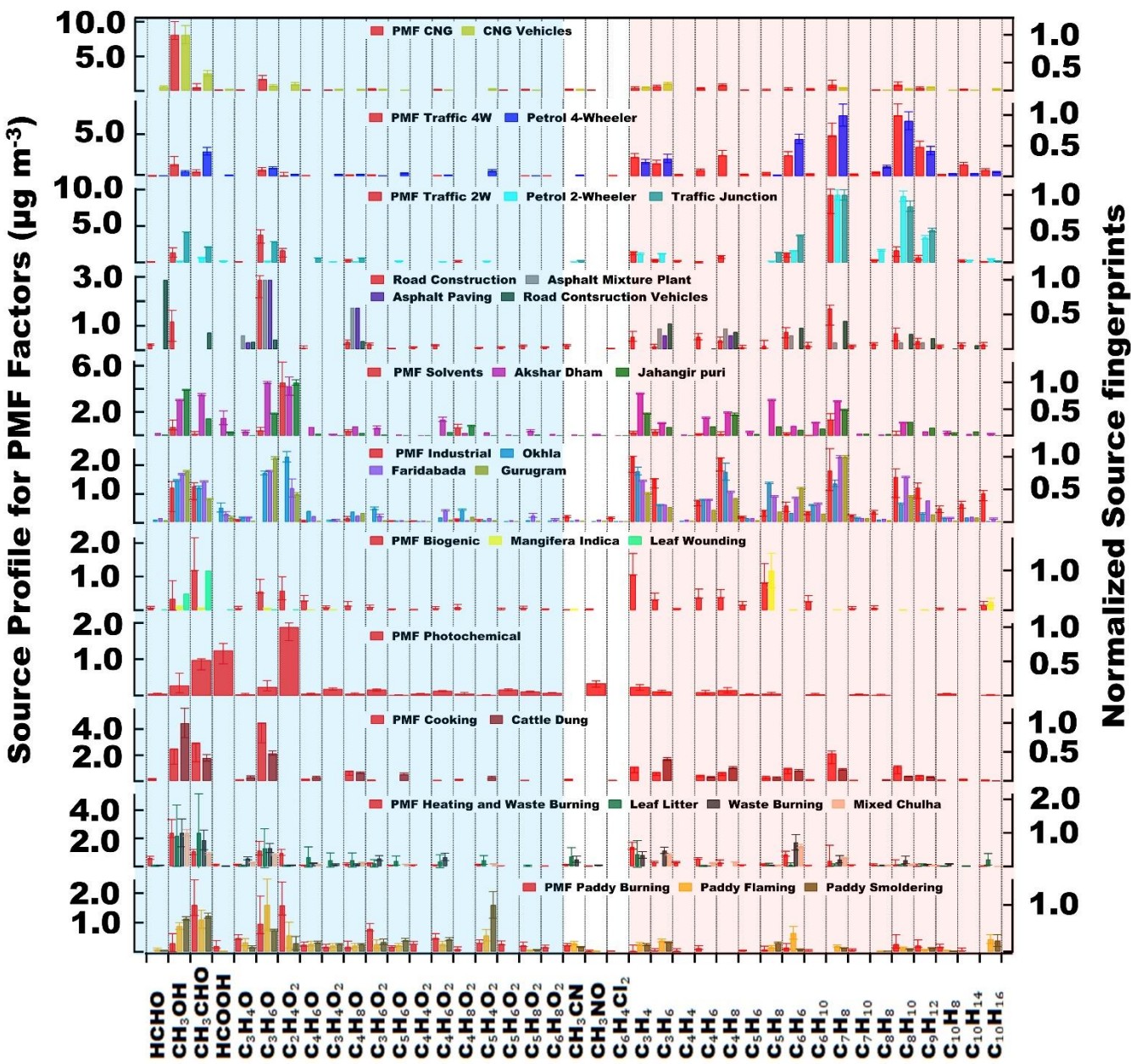

**Figure 3: PMF factor profile of the 11 factors identified.** The source profile in µg m$^{-3}$ (left in red) and the normalized source fingerprint of grab samples collected at the source (right in various colours). The Error bars indicate the 2σ uncertainty range from the bootstrap runs for PMF factor profiles and the 1σ error of the mean of the emission factors for source samples.

Figure 4 shows the relative contribution of different sources to the total pollution burden of VOCs, PM$_{2.5}$ and PM$_{10}$ at the receptor site. In the megacity of Delhi, transport sector sources contributed most (42±4 %) to the total VOC burden, while it

contributed much less (only 24 %) to the total VOC burden in Mohali a suburban site 250 km north of Delhi during the same

season (Singh et al., 2023). On the other hand, the contribution of paddy residue burning (6±2 %) and the summed residential sector emissions (17±3 % in Delhi and 18 % in Mohali) to the total VOC burden during post-monsoon season were similar at both sites. The contribution of the different factors to the SOA formation potential (Fig. 4e) stands in stark contrast to their contribution to primary particulate matter emissions. SOA formation potential was dominated by the transport sector (54 %) while direct $PM_{10}$ (52±8 %) and $PM_{2.5}$ (48±12 %) emissions were dominated by different biomass burning sources (Fig. 4 b &

c). CNG-fuelled vehicles also contribute significantly to the $PM_{10}$ (15±3 %) and $PM_{2.5}$ (11±3 %) burden. A significant share of the $PM_{10}$ (18 %) and $PM_{2.5}$ (28 %) burden is associated with the residual and not directly linked to combustion tracers. This share can likely be attributed to windblown dust arriving at the site through long-range transport (Pawar et al., 2015) and to secondary organic, and secondary inorganic aerosols such as ammonium sulphate and ammonium nitrate. Due to the complex relationship of secondary aerosol with gas-phase precursors and emission tracers, VOC tracers are not a suitable tool to source-

apportion this aerosol component. Meteorological conditions, homogeneous, heterogeneous, and multiphase chemistry control how fast primary emissions are converted to secondary aerosol. To explain the source of those species, one also needs to invoke the physicochemical and thermodynamical properties of the aerosol (Acharja et al., 2022).

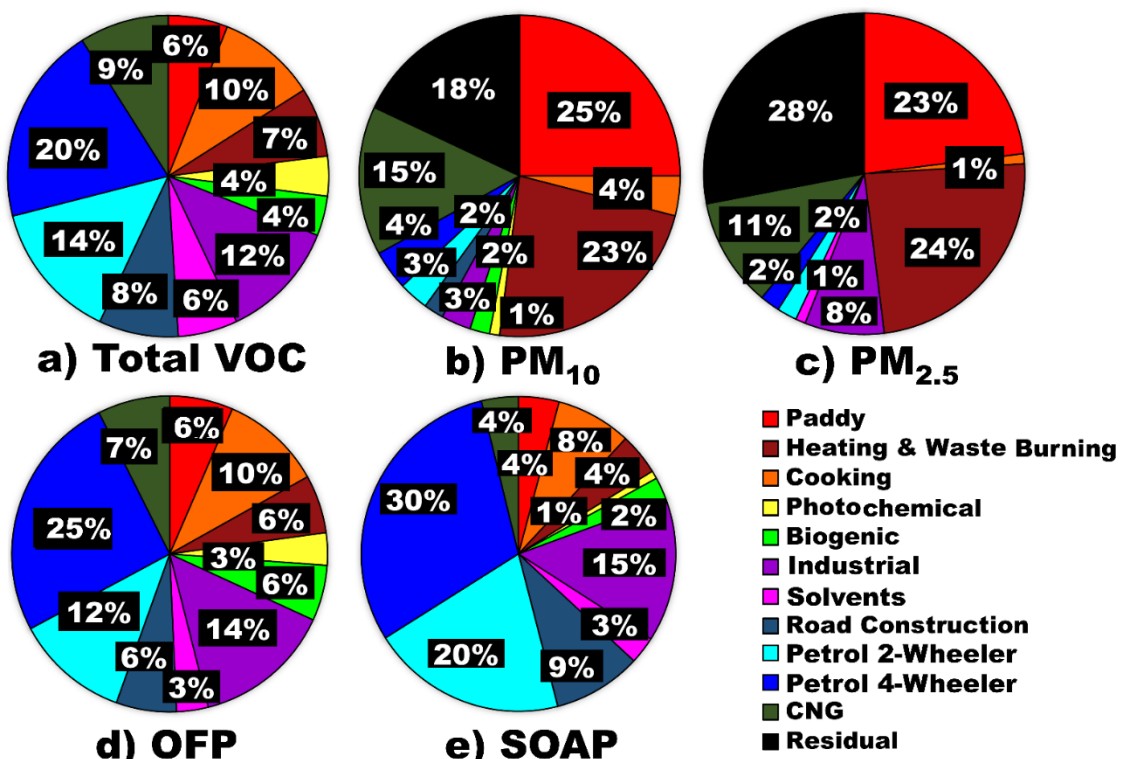

**Figure 4:** Source contribution of the 11 sources to the (a) total ambient VOC mass loading, (b) $PM_{10}$ mass loading (c) $PM_{2.5}$ mass

loading (d) ozone formation potential and (e) SOA formation potential.

### 3.2 Detailed discussion of individual emission sources

### 3.2.1 Factor 1: Paddy residue burning

Paddy residue burning was one of the largest contributors to the total observed $PM_{10}$ (25 %) and $PM_{2.5}$ (23 %) (Fig.4b,4c) mass concentrations in Delhi. An earlier WRF-Chem-based study with the FINNv1.5 inventory had attributed 20 % of the $PM_{2.5}$ burden to this source for the year 2018 (Kulkarni et al., 2020). Its importance as a PM source stands in stark contrast to its minor contribution to the overall VOC mass loading in Delhi (6 %). In Mohali, Punjab, this source was also found to only contribute 6 % to the VOC burden in October and November (Singh et al., 2023). In descending rank of mass contribution, acetaldehyde ($CH_3CHO$), acetic acid ($C_2H_4O_2$), acetone + propanal ($C_3H_6O$), hydroxyacetone ($C_3H_6O_2$), acrolein ($C_3H_4O$), diketone ($C_4H_6O_2$) and furfural ($C_5H_4O_2$) contributed most to the total VOC mass of this factor. Figure 5 shows that the 24-h averaged factor contribution time series has the highest cross correlation with same day fire counts (R=0.8), while hourly average source contributions correlate most with $PM_{2.5}$ (0.7) and $PM_{10}$ (0.7) (Table S4). The high correlation with same-day fire counts points towards nearby fire activity as the dominant source of paddy burning-related pollution in the Delhi NCR. A recent study from Punjab indicated that the largest PM enhancements at a receptor are caused by fire occurring within 50 km radius around the receptor site (Pawar & Sinha, 2022). Figure S6 shows that the $PM_{2.5}$ and $PM_{10}$ mass loadings at the receptor site increased by $0.027\pm0.006$ and $0.047\pm0.01$ µg m$^{-3}$, respectively for each additional fire count within the 24-hour fetch region whenever the trajectories are arriving through north-west and south-west region. It is very interesting to note that the incremental increase in $PM_{2.5}$ and $PM_{10}$ mass loadings for each additional fire count were almost four times higher than the former regions when the trajectory fetch region was south-east with $0.11\pm0.01$ and $0.19\pm0.02$ µg m$^{-3}$, respectively, likely because the complete burns of entire fields (Figure S7) that are prominent in Punjab can be more easily identified as a fire activity with satellite-based detection (Liu et al., 2019; 2020), while the partial burns (Figure S8) that are more prevalent in the eastern IGP and in Haryana have larger omission errors (Liu et al., 2019; 2020). Regional gradients in fire detection efficiency can complicate attempts to model air quality with the help of fire-count-based emission inventories (Kulkarni et al., 2020).

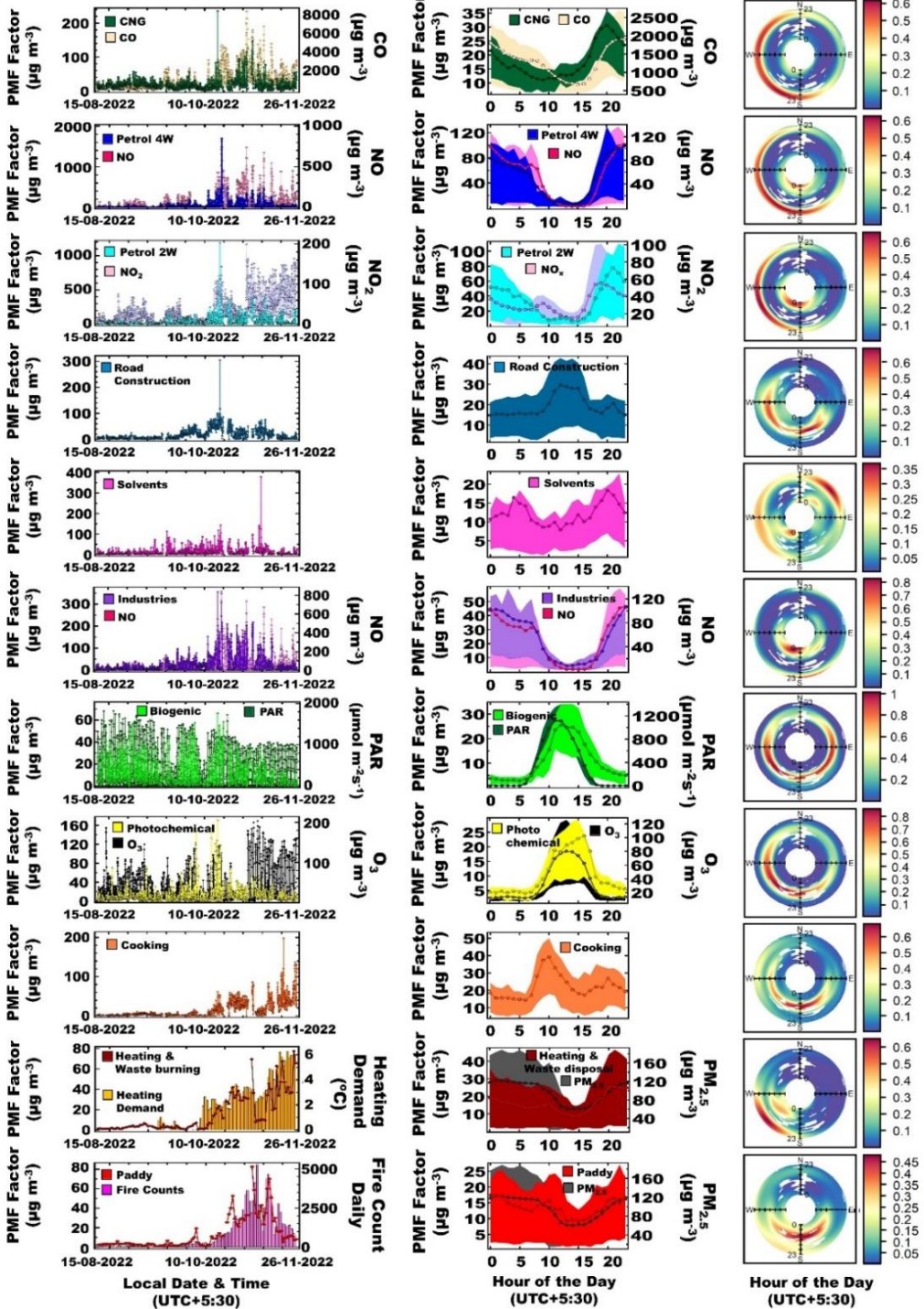

**Figure 5: Time series of each factor in μg m⁻³ (left column) with respective normalized diurnal profiles (centre column). The shaded region in the diurnal profiles depicts the area between the 25th and 75th percentile while the median of the dataset is marked as the line. The polar plots (right column) depict the conditional probability of a factor having a mass contribution above the 75th percentile**



of the dataset during a certain hour of the day between midnight (centre of rose) and 23:00 local time (outside of rose) from a certain wind direction. This probability is determined by dividing the number of observations above the 75$^{th}$ percentile by the total number of measurements in each bin.

Figure 6 shows that this factor explained the largest percentage share of O-heteroarene compounds such as furfural ($C_5H_4O_2$), methyl furfural ($C_6H_6O_2$), hydroxy methyl furfural ($C_6H_6O_3$), furanone ($C_4H_4O_2$), hydroxymethyl furanone ($C_5H_6O_3$), furfuryl alcohol ($C_5H_6O_2$), furan ($C_4H_4O$), methyl furans ($C_5H_6O$), C2-substituted furans ($C_6H_8O$), and C3-substituted furans ($C_7H_{10}O$),

which are produced by the pyrolysis of cellulose and hemicellulose, and have previously been detected in biomass burning samples (Coggon et al., 2019; Hatch et al., 2015; 2017; Koss et al., 2018; Stockwell et al., 2015). Figure 6 also shows that this factor explains the largest share of the most abundant oxidation products that result from the nitrate radical-initiated oxidation of toluene as well as from OH-initiated oxidation of aromatic compounds under high $NO_x$ conditions, namely nitrotoluene ($C_7H_7NO_2$) and nitrocresols ($C_7H_7NO_3$) (Ramasamy et al., 2019), which indicates a certain degree of aging of the plumes.

These nitroaromatic compounds are significant contributors to SOA and BrC, (Palm et al., 2020, Harrison et al., 2005). It also explains several other nitrogen containing VOCs such as nitroethane ($C_2H_6NO_2$), the biomass burning tracer acetonitrile ($CH_3CN$) and pentanenitrile ($C_5H_9N$). The presence of pentanenitrile isomers in biomass burning smoke has previously been confirmed using gas chromatography-based studies (Hatch et al., 2015, Hatch et al., 2017). In addition the factor explains the largest percentage share of acrolein ($C_3H_4O$), hydroxyacetone ($C_3H_6O_2$), cyclopentadienone ($C_5H_4O$), cyclopentanone

($C_5H_8O$), diketone ($C_4H_6O_2$), pentanedione ($C_5H_8O_2$), hydroxybenzaldehyde ($C_7H_6O_2$), guaiacol ($C_7H_8O_2$), and the levoglucosan fragment ($C_6H_8O_4$), many of these compounds are known to form during lignin pyrolysis (Hatch et al., 2015, Koss et al., 2018; Nowakowska et al., 2018), while dimethylbutenedial ($C_6H_8O_2$), trimethylbutenedial ($C_7H_{10}O_2$) are ring opening oxidation products of aromatic compounds (Zaytsev et al., 2019). Figure S9 shows the volatility oxidation state plot for all 111 VOCs in which the marker size represents the percentage share of each compound explained by the paddy residue

burning factor and markers are colour coded by the number of carbon atoms. The plot shows evidence of the first- and second-generation oxidation products of C5 and C6 hydrocarbon transitioning from the VOC to the IVOC range along trajectories expected for the addition of =O functionality to the molecule (Jimenez, et al. 2009), while C7 hydrocarbons progress along trajectories expected for both the addition of -OH and =O functionality. This indicates that paddy residue burning contributes significantly to the SOA burden. However, the fact that the $PM_{10}$ mass associated with this factor (36.5 µg m$^{-3}$) is 1.8 times

larger than the $PM_{2.5}$ mass (20.7 µg m$^{-3}$) and 3 times larger than the VOC mass (11.6 µg m$^{-3}$) released during the same combustion process, points towards the relatively coarse ash formed from the phytolith skeleton of rice straw (Figure S10) as the dominant aerosol source.

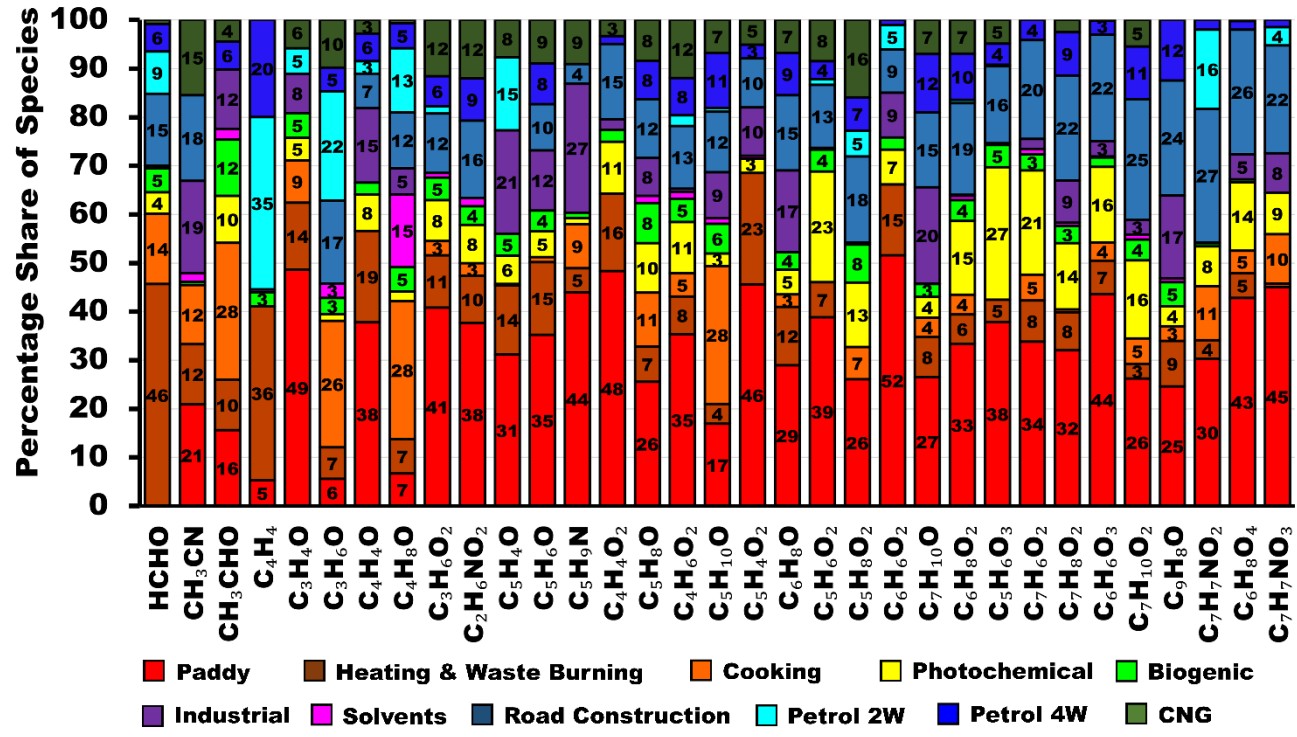

**Figure 6: Contribution of PMF factors to VOC species for which different forms of biomass burning contribute the highest percentage share of the atmospheric burden in Delhi.**

### 3.2.2 Factor 2: Residential heating and waste burning

The residential heating and waste burning factor is the second largest particulate matter source at the receptor site and contributes 23 % and 24 % to the total $PM_{10}$ and $PM_{2.5}$ mass loadings, respectively (Fig. 4), while it contributed only 7 % to the total VOC mass loading, 6 % and 4 % to the ozone and SOA formation potential, respectively (Fig.4). Emissions peak at nighttime (Fig. 5) and the factor contribution time series displays the largest cross-correlation with the 24 h averaged heating demand (R=0.8) (Fig. S6), $PM_{10}$ (R=0.7), $PM_{2.5}$ (R=0.6) and $NO_2$ (R=0.7) and CO (R=0.5) (Table S4). The lower correlation with NO (R=0.4) (Table S4), indicated that emissions are combustion-related but not always fresh. Occasionally, fresh plumes reach the receptor within minutes, however the majority of plumes have a higher atmospheric age, as NO is a short-lived species and oxidized to $NO_2$ on the timescale of minutes in the presence of ozone (Sinha et al., 2014). The factor contribution time series is anti-correlated with temperature (R=-0.6) and has its strong correlation with the 24 h averaged heating demand (R=0.8) indicating that this combustion activity is primarily triggered by the need to keep warm. Figure S11 shows that the $PM_{2.5}$ and $PM_{10}$ mass loadings at the receptor site increase by 13.9 µg m$^{-3}$ and 22.3 µg m$^{-3}$, respectively for each degree increase in the 24-h average heating demand. Earlier studies have documented the strong seasonality of open waste burning emissions over Delhi, as well as the diversity of fuel used in wintertime heating-related fires (Nagpure et al., 2015). This factor explains 7 % of the total VOC mass loading. The top contributors to the VOC mass of this factor are in descending rank of contribution:

methanol ($CH_3OH$), propyne ($C_3H_4$), acetone + propanal ($C_3H_6O$), acetaldehyde ($CH_3CHO$), acetic acid ($C_2H_4O_2$) and benzene ($C_6H_6$). Figure 6 shows that this factor explains the largest percentage share of the total mass for formaldehyde (HCHO) and vinylacetylene + 1-buten-3-yne ($C_4H_4$), and the second largest percentage share of furfural ($C_5H_4O_2$), methylfurfural ($C_6H_6O_2$), furan ($C_4H_4O$), methyl furan ($C_5H_6O$), furanone ($C_4H_4O_2$) and acrolein ($C_3H_4O$). All these compounds are characteristic of biomass burning smoke (Hatch et al., 2015; Stockwell et al., 2015; Koss et al., 2018).

### 3.2.3 Factor 3: Solid fuel-based cooking

The cooking factor is a daytime factor and explains 10 % of the total VOC mass loading, 10 % and 8 % of the ozone and SOA formation potential (Fig. 4), but only a negligible share of the total $PM_{10}$ ($\leq$4 %) burden. The volatility oxidation space plot (Figure S9) also shows very little evidence of IVOC oxidation products that could partition into the aerosol phase. The activity peaks from 8 am to noon time, with a secondary peak in the early evening hours and persists throughout monsoon and post-monsoon season. Emissions reaching the receptor site show no correlation with NO (R=0.1) indicating plumes are not fresh. In descending rank of mass contribution ($C_3H_6O$), acetaldehyde ($CH_3CHO$), methanol ($CH_3OH$), toluene ($C_7H_8$), the sum of C8 aromatics ($C_8H_{10}$), propyne ($C_3H_4$) and benzene ($C_6H_6$) contribute most to this factor. These aromatic compounds have been reported to originate from cooking emissions (Crippa et al., 2013). Figure 6 shows that factor explains the largest percentage share of butanone ($C_4H_8O$), pentanone ($C_5H_{10}O$), acetaldehyde ($CH_3CHO$), acetone ($C_3H_6O$), and benzaldehyde ($C_7H_6O$). All these compounds are characteristic of biomass burning smoke (Hatch et al., 2015; Stockwell et al., 2015; Koss et al., 2018).

### 3.2.4 Factor 4: CNG

CNG-fuelled vehicles are identified as the third largest source of $PM_{10}$ (15 %) and $PM_{2.5}$ (11 %) and contribute 9 % to the total VOC burden (Fig. 4). The much higher contribution of this source to the coarse mode particulate matter burden (22.5 $\mu gm^{-3}$ $PM_{10}$) when compared to the fine mode particular matter burden (10.4 $\mu gm^{-3}$ $PM_{2.5}$), confirms earlier emission-inventory-based estimates which flagged that non-tailpipe emissions such as brake and tire wear and road dust resuspension have become the dominant transport sector related particulate matter sources in the Delhi-NCR region (Nagpure et al., 2016). Non-tailpipe emissions such as brake and tire wear and road dust resuspension contribute most to the $PM_{10}$ burden, although they have also become the largest source of transport sector fine mode aerosol and VOC emissions in some countries that have transitioned to Euro-6 norms (Harrison et al., 2021). This study attributes a large share of these non-tailpipe emissions to trucks, buses and other commercial vehicles that are typically fuelled by CNG, because commercial diesel vehicles of <10 years age face severe entry restrictions, that limit their use within the Delhi NCR while older diesel vehicles have been completely banned from plying within City limits. Policy interventions in favour of CNG use (Krelling & Badami, 2022) have resulted in a halving of diesel sales, a rapid conversion of Delhi's HDV fleet to CNG (Figure S12), and a significant reduction in tailpipe exhaust emissions. In descending order methanol ($CH_3OH$), acetone + propanal ($C_3H_6O$), toluene ($C_7H_8$), C8 aromatic compounds ($C_8H_{10}$), butene ($C_4H_8$), propene ($C_3H_6$), and acetaldehyde ($CH_3CHO$) contribute most to the VOC mass in this source. Figure 7 shows that the factor explains the largest percentage share of methanol ($CH_3OH$) and the second largest percentage share of

ethanol ($C_2H_6O$). These compounds are formed by the incomplete combustion of CNG that is catalytically converted to methanol and ethanol (Singh et al., 2016).

### 3.2.5 Factor 5: Petrol 4-wheeler factor

Figure 4 shows petrol 4-wheeler contributed 20 %, 25 %, and 30 % to the VOC mass loading, OFP, and SOAP, respectively. The source fingerprint of this matched tailpipe emissions of petrol-fuelled 4-wheelers (Hakkim et al., 2021) and is characterized, in descending rank of contribution, by C8-aromatics, toluene, C9-aromatics ($C_9H_{12}$), benzene, butene + methyl tert-butyl ether (MTBE) fragment, propyne, propene, methanol and C2-substituted xylenes + C4-substituted benzenes ($C_{10}H_{14}$).

Figure 5 shows that emissions peak in the evening between 7 pm and midnight with average VOC mass loadings >70 µg m$^{-3}$ and reach the receptor site from most wind directions. Emissions are strongly correlated with NO (R=0.8), CO (R=0.7), and $CO_2$ (R=0.7) indicating the receptor site is impacted by fresh combustion emissions from this source and the atmospheric age of most plumes is on the timescale of minutes. Figure 7 shows that the factor explains the largest percentage share of most aromatic compounds, namely C8-aromatics, toluene, C9-aromatics ($C_8H_{12}$), C4-substituted benzene + C2-substituted xylene, benzene, styrene ($C_8H_8$), methylstyrenes + indane ($C_9H_{10}$), and C2-substituted styrenes ($C_{10}H_{12}$) and a few oxygenated aromatic hydrocarbons such as methyl phenol isomers ($C_7H_8O$) and methyl chavicol ($C_{10}H_{12}O$). The fact that the factor explains the largest percentage share of ethanol and the MTBE fragment ($C_4H_8$) can likely be attributed to ethanol blending and the use of MTBE in petrol (Achten et al., 2001). This factor also explains the largest percentage share of several other hydrocarbons such as propyne ($C_3H_4$), propene ($C_3H_6$), cyclopentadiene($C_5H_6$), hexane ($C_6H_{13}$), $C_7H_6$, $C_7H_{10}$, and cycloheptene ($C_7H_{12}$).

Figure S9 shows that this factor contributes significantly to the burden of C6 to C10 hydrocarbons, and hence SOA formation potential. However, due to freshly emitted plumes, it hardly contributes to the burden of the first and second-generation oxidation products of these hydrocarbons at the receptor site. Instead, this factor is likely to contribute to secondary pollution formation downwind of the Delhi NCR.

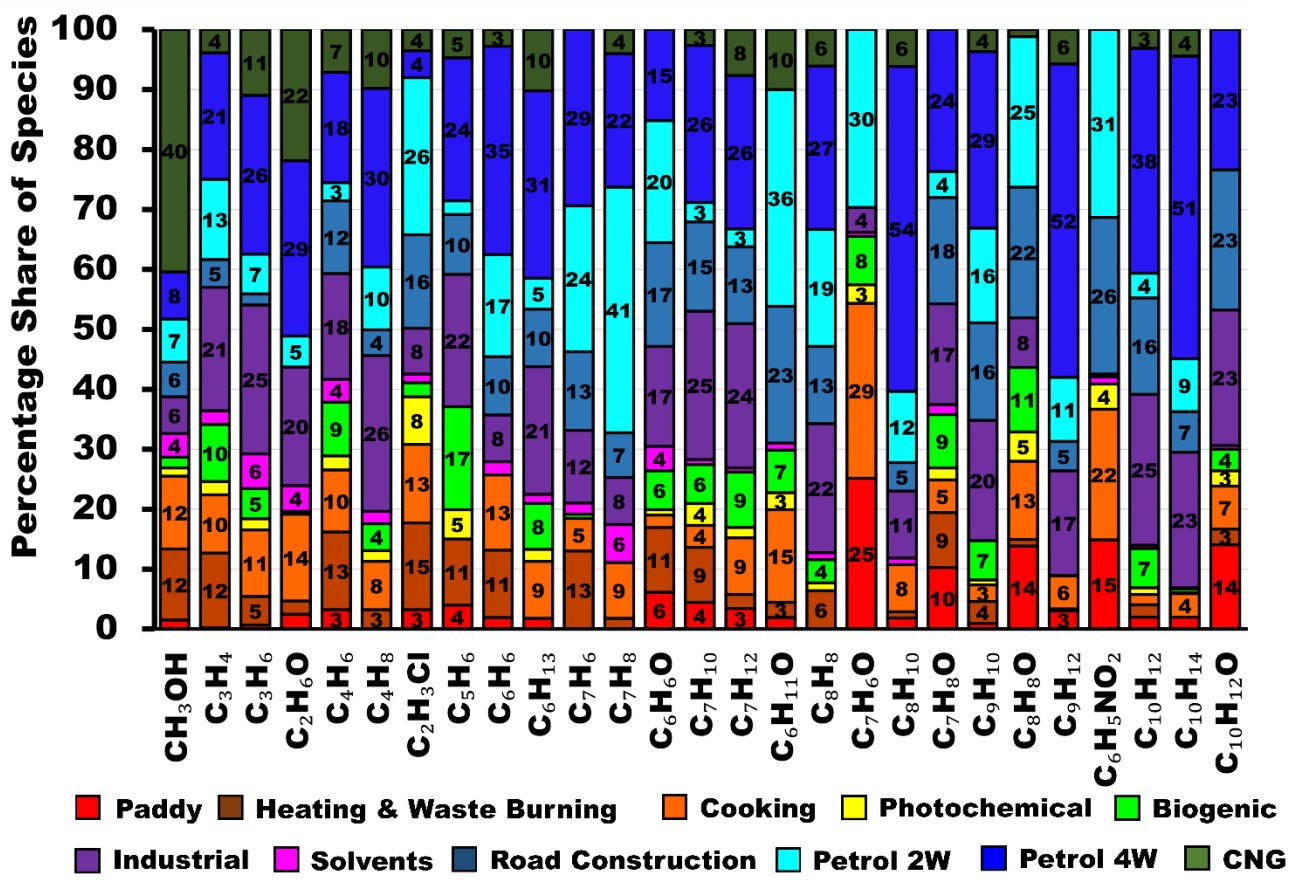

Figure 7: Contribution of PMF factors to VOC species for which the transport sector contributes the highest percentage share of the atmospheric burden in Delhi.

### 3.2.6 Factor 6: Petrol 2-wheeler factor

Figure 4 shows petrol 2-wheeler contributed 14 %, 12 %, and 20 % to the VOC mass loading, OFP, and SOAP respectively. The source fingerprint of this source matched tailpipe emissions of petrol-based 2-wheelers (Hakkim et al., 2021) and are characterized, in descending rank of contribution, by toluene, acetone + propanal, C-8 aromatic compounds, acetic acid ($C_2H_4O_2$), propyne ($C_3H_4$), methanol ($CH_3OH$), benzene ($C_6H_6$), the MTBE fragment and C-9 aromatics ($C_9H_{12}$). A key difference of the petrol 2-wheeler source profile in comparison to the petrol 4-wheeler source profile is the lower benzene to toluene ratio, which is supported by the GC-FID analysis of tailpipe exhaust (Kumar et al., 2020). Figure 5 shows that emissions peak in the evening between 8 pm and 10 pm with average VOC mass loadings >50 µg m$^{-3}$ and reach the receptor site from most wind directions. Emissions are strongly correlated with $NO_x$ (R=0.6), CO (R=0.6), and $CO_2$ (R=0.7), but have a lower correlation with NO (R=0.5) (Table S4), and a larger contribution of oxygenated compounds to the source profile,

indicating that the emissions have been photochemically aged. This suggests that contrary to 4-wheeler plumes which originate from the immediate vicinity of the receptor site in central Delhi (Figure S1), 2-wheeler plumes reach the receptor after prolonged transport from more distant rural and suburban areas on the outskirts of the city. In such areas, people often favour two-wheelers over four-wheelers. Figure 7 shows that this factor explains the largest percentage share of toluene, and a number of oxygenated aromatic compounds such as benzaldehyde ($C_7H_6O$), tolualdehyde ($C_8H_8O$), and phenol ($C_6H_6O$). It also explains the largest percentage share of nitrobenzene ($C_6H_5NO_2$), cyclohexanone ($C_6H_{11}O$), and vinyl chloride ($C_2H_3Cl$). It also explains the second largest percentage share of benzene, vinylacetylene ($C_4H_4$), acetone + propanal, methoxyamine ($CH_5NO$), and butanoic acid/ethyl acetate ($C_4H_9O_2$).

### 3.2.7 Factor 7: Industrial

This factor contributes 12 %, 14 %, 15 %, 8 % and 3 % to the VOC mass loading, OFP, SOAP, $PM_{2.5}$ and $PM_{10}$ mass loading. On average more than 30$\mu$g m$^{-3}$ of the VOC burden throughout the night from 9 pm to 7 am (Fig. 5) is from this factor. This factor is identified as an industrial point sources located in the wind sector S to SW of the receptor site. Emissions are most strongly correlated with CO (R=0.7), NO (R=0.7), CH4 (R=0.8), and $CO_2$ (R=0.8) indicating that the emissions are fresh and originate from combustion processes. The main contributors towards the VOC mass in the industrial factor, are in descending order of contribution propyne ($C_3H_4$), butene + MTBE fragment ($C_4H_8$), toluene ($C_7H_8$), C-8 aromatic compounds ($C_8H_{10}$), propene ($C_3H_6$), acetaldehyde ($CH_3CHO$), methanol ($CH_3OH$), C-9 aromatics and the sum of monoterpenes ($C_{10}H_{16}$). The source fingerprint is most similar to ambient air grab samples collected near the Okhla waste to energy plant and industrial area in Faridabad.

Figure 8 shows that the factor explains the largest percentage share of methanethiol ($CH_5S$), a chemical used in the manufacturing of the essential amino acid methionine, in the plastic industry and the manufacturing of pesticides dichlorobenzenes ($C_6H_4Cl_2$), a chemical used in the synthesis of dyes, pesticides, and other industrial products and methoxyamine ($CH_5NO$). Analyses of the primary dataset by Mishra et al. (2024) also qualitatively inferred an industrial source for methanethiol and dichlorobenzene. It also explains the largest percentage share of the sum of monoterpenes, camphor/pinene oxide ($C_{10}H_{16}O$), santene ($C_9H_{14}$), the terpene fragment ($C_8H_{12}$), $C_8H_{14}$, $C_9H_{16}$, cyclohexene ($C_6H_{10}$) and cyclopentylbenzene ($C_{11}H_{14}$). Terpenes are used in the food and beverages, cosmetics, pharmaceutical, and rubber industry. In addition, this factor also explains the largest percentage share of a large suite of volatile and IVOC aromatic hydrocarbons including naphthalene ($C_{10}H_8$), methyl naphthalene ($C_{11}H_{10}$), $C_{12}H_{16}$, $C_{13}H_{18}$, $C_{13}H_{20}$, $C_{13}H_{22}$, $C_{14}H_{20}$, and $C_{14}H_{22}$. Ambient observations for most of these IVOCs have not been reported in the literature so far. Only, $C_9H_{14}$, $C_{12}H_{12}$, and $C_{12}H_{16}$ have been reported from aircraft engine emissions (Kılıç et al., 2018) while terpenes, $C_9H_{16}$, cyclopentylbenzene, naphthalene and methyl naphthalene have been reported from a wide range of combustion sources (Hatch et al., 2015, Bruns et al., 2017). Most other compounds have so far only been reported to degas from heated asphalt (Khare et al., 2020). Due to the high abundance of IVOCs in this factor, it contributes 15 % to the total SOA formation potential. Figure S9 shows the volatility oxidation state plot for all 111 VOCs in which the marker size represents the percentage share of each compound explained by the industrial

factor and markers are colour coded by the number of carbon atoms. The plot shows evidence of the first- and second-generation oxidation products of C6 to C10 hydrocarbon transitioning from the VOC to the IVOC range along trajectories expected for the addition of =O functionality to the molecule (Jimenez, et al. 2009). This and the fact that the entire aerosol associated with this factor is PM$_{2.5}$, indicates that most of the aerosol associated with this factor is likely SOA.


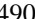

**Figure 8: Contribution of PMF factors to VOC species for which the industries, solvent usage, photochemistry, or biogenic sources contribute the highest percentage share of the atmospheric burden in Delhi.**

### 3.2.8 Factor 8: Solvents and Evaporative Emissions

Solvent usage and evaporative emissions reach the site from several point sources and wind directions often in the form of short and intense plumes that show no correlation with combustion tracers. This source contributes most to the VOC burden at night and explains 6 % of the total VOC but ≤1 % of the total PM$_{2.5}$ and PM$_{10}$ mass (Fig. 4). The source fingerprint of the solvents factor (Fig. 3) is characterized in descending rank of mass contribution by acetic acid + glycolaldehyde (C$_2$H$_4$O$_2$), toluene (C$_7$H$_8$), methanol (CH$_3$OH), butanoic acid/ethyl acetate (C$_4$H$_9$O$_2$), acetone + propanal (C$_3$H$_6$O) and butanal + butanone 500 + MEK (C$_4$H$_8$O). Figure 8 shows that the factor explains the largest share of organic acids namely butanoic acid, acetic acid,

and isocyanic acid (HNCO), and the second largest share of butanal + butanone + MEK ($C_4H_8O$). These compounds point towards stack venting of VOCs from chemical, food, or pharmaceutical industries or polymer manufacturing as likely sources of these emissions (Hodgson et al., 2000, Villberg et al., 2001, Jankowski et al., 2017, Gao et al., 2019). This assessment is broadly confirmed by the fact that the best match (R=0.7) for this source was collected from a plot situated opposite a polymer

manufacturing unit and next to a pet food manufacturer in an industrial area at Jahangirpuri, North of the receptor site.

### 3.2.9 Factor 9: Road construction

The road construction factor contributed 8 % to the total VOC mass loading and 2 % to the total $PM_{10}$. This factor is almost absent during monsoon season, as road repair work is mostly avoided during this period due to water logging risks, and emissions from this source generally peak during the day as degassing of compounds from asphalt is temperature-driven and

continues for days after the initial paving (Khare et al., 2020). The source fingerprint of the road construction factor is characterized in descending order of the mass concentrations by acetone + propionaldehyde, toluene, methanol, benzene, and C8-aromatics. Acetone and propionaldehyde were found to be the most abundant oxygenated volatile organic compounds emitted during asphalt paving (Li et al., 2020). The source profile had the greatest similarity with the mix of emissions that would originate from asphalt paving (Li et al., 2020) and the tailpipe of road construction vehicles (Che et al., 2023). As

represented by Fig. 9, this factor explains the largest percentage share of a large suite of volatile and IVOC hydrocarbons namely, heptene ($C_7H_{14}$), $C_{11}H_{12}$, $C_{12}H_{12}$, $C_{14}H_{14}$, $C_{14}H_{18}$, $C_{16}H_{24}$, $C_{17}H_{28}$, and $C_{18}H_{30}$. In addition, it explains the second largest percentage share of many other IVOC hydrocarbons namely $C_9H_{14}$, $C_9H_{16}$, $C_{11}H_{14}$, $C_{12}H_{16}$, $C_{13}H_{18}$, $C_{13}H_{20}$, $C_{13}H_{22}$, $C_{14}H_{20}$, $C_{14}H_{22}$. Except for the four hydrocarbons $C_7H_{14}$, $C_9H_{14}$, $C_9H_{16}$, and $C_{11}H_{12}$, all of these IVOCs have been reported to degas at 60°C from asphalt pavement (Khare et al., 2020). So far only $C_{14}H_{18}$ has been reported as fresh gas phase emissions (transport

time <2.5 min) from a farm (Loubet et al., 2022) in ambient air, while $C_{17}H_{28}$ has been reported in the aerosol phase (Xu et al., 2022). The road construction factor also explains the largest percentage share of a long list of OVOCs namely, C6 diketone isomers ($C_6H_{10}O_2$), C2-substituted phenol ($C_8H_{10}O$), $C_7H_{12}O_2$, $C_8H_{14}O_2$, $C_8H_{16}O_2$, phthalic anhydride ($C_8H_4O_3$), which is a naphthalene oxidation product (Bruns et al., 2017), $C_9H_{10}O$, $C_9H_{12}O_2$, $C_9H_{14}O_2$, $C_9H_{16}O_2$, $C_9H_{18}O_2$, $C_{10}H_{12}O$, $C_{10}H_{18}O$, $C_{10}H_8O_3$, $C_{10}H_{16}O_3$, and $C_{12}H_{18}O_2$. However, out of these only $C_{10}H_{12}O$ and $C_{10}H_{18}O$ have been detected as direct emissions

from heated asphalt pavement (Khare et al., 2020) indicating that most OVOCs in this factor are possibly oxidation products of short-lived IVOCs hydrocarbons emitted by this source. This assessment is supported by the volatility oxidation state plot for the road construction factor (Figure S9) which demonstrates that both precursors and oxidation products are present in this factor and that C6 to C10 hydrocarbons appear to be progressing from the VOC to the IVOC range along trajectories expected for the addition of =O functionality to the molecule (Jimenez, et al. 2009).


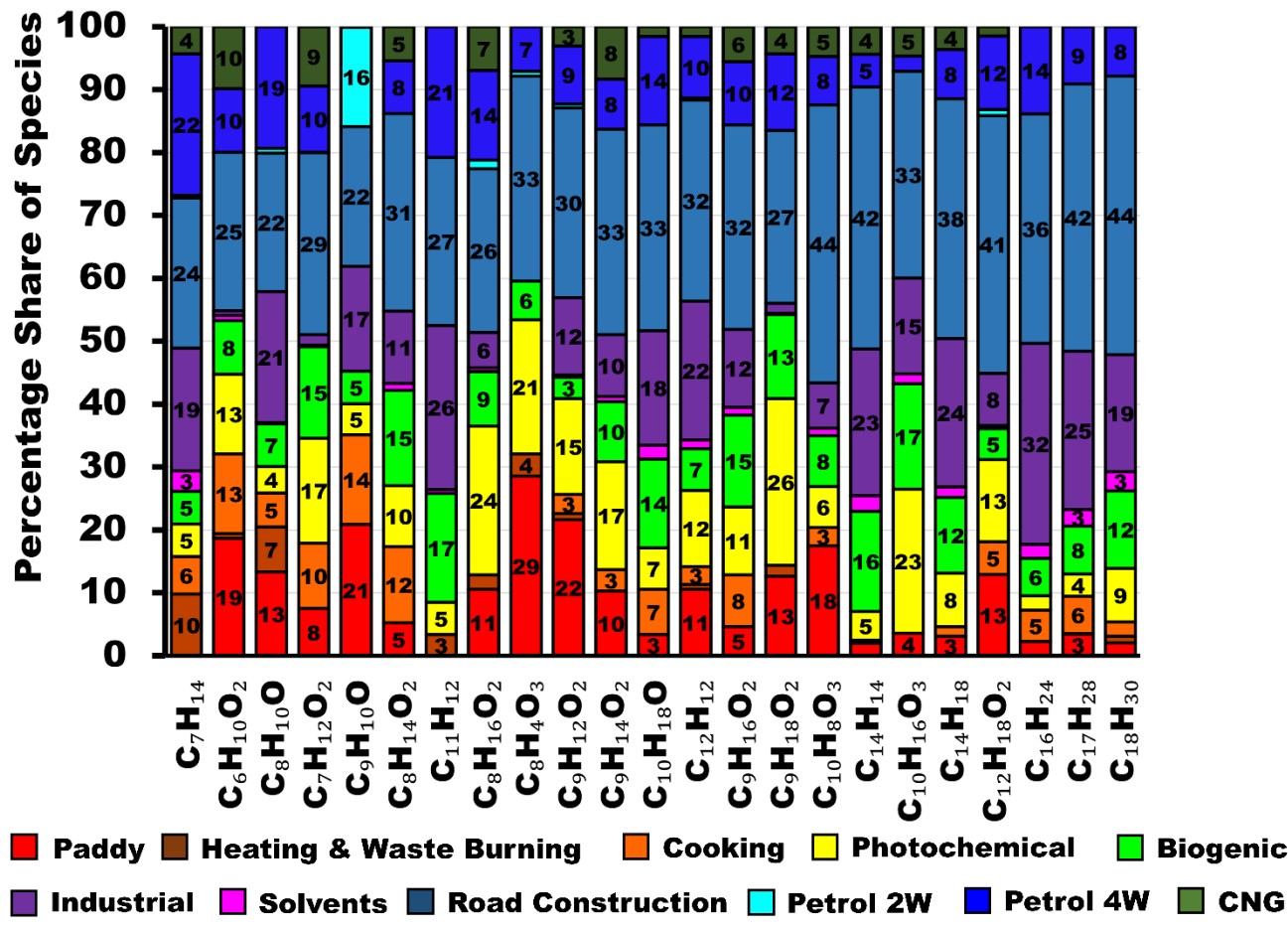

**Figure 9: Contribution of PMF factors to VOC species for which the road construction contributes the highest percentage share of the atmospheric burden in Delhi.**

### 3.2.10 Factor 10: Photochemistry

The photochemical factor has a diurnal profile that follows the diurnal profile of ozone (R=0.4). The factor profile is dominated by OVOCs such as acetic acid ($C_2H_4O_2$), formic acid ($CH_3O_2$), acetaldehyde ($CH_3CHO$), formamide ($CH_4NO$), and methanol ($CH_3OH$). Figure 8 shows that the factor explains the largest percentage share of formic acid, formamide, and methyl glyoxal ($C_3H_5O_2$). It also explains the second largest percentage share of isocyanic acid (HNCO) and hexanamide ($C_6H_{13}NO$), which are formed by the photooxidation of amines (Yao et al., 2016; Wang et al., 2022). Some compounds point towards a significant contribution of photochemically aged biomass burning emissions to this factor for example furfuryl alcohol ($C_5H_6O_2$), hydroxymethyl furanone ($C_5H_6O_3$), and hydroxybenzaldehyde ($C_7H_6O_2$). While this factor explained ≤4 % of the total VOC share and negligible share of PM$_{2.5}$ and PM$_{10}$ mass in Delhi, photochemically aged biomass burning emissions were a significant source of VOCs at a suburban site in Punjab during the post-monsoon season of 2017 (Singh et al., 2023). The

difference is likely due to the fact that the great smog episode of 2017 was primarily driven by low wind speeds a shallow boundary layer and regional-scale build-up of emissions over a prolonged period (Dekker et al., 2019, Roozitalab, et al., 2021), while the post-monsoon season of 2022 experienced western disturbances and higher ventilation coefficients. The factor also explains the largest percentage share of the total mass for organic acids such as nonanoic acid ($C_9H_{18}O_2$), n-octanoic acid ($C_8H_{16}O_2$) which have been detected in biomass-burning impacted environments in China (Mochizuki et al., 2019), $C_{12}H_{18}O_2$ which has been found in aged wildfire plumes in the US (Haeri, 2023), and the terpene ozonolysis products norpinonaldehyde ($C_9H_{14}O_2$) and cis-Pinonic acid ($C_{10}H_{16}O_3$) (Camredon et al., 2010) and $C_7H_{12}O_2$. Pinonic acid was found to be an important aerosol phase tracer of biogenic SOA formation in India (Mahilang et al., 2021) and $C_7H_{12}O_2$ has been reported as a pinonic acid aqueous-phase photolysis product (Lignell et al., 2013) Fig. 8.

### 3.2.11 Factor 11: Biogenic

Biogenic VOC emissions at the receptor site show the highest cross-correlation with photosynthetic active radiation (PAR, R=0.7) and temperature (R=0.7) (Table S4) and explain 4 % of the total VOC burden and 2 % of the $PM_{10}$ burden in the PMF. The BVOC emission in this factor is relatively fresh as the ratio of isoprene to its first-generation oxidation products MEK ($C_4H_8O$) and MVK+MACR ($C_4H_6O$) is 5.9 and 3.0, respectively. At the site, the top of the tree canopy of roadside trees is located approximately 20 m below the inlet height. Figure 3 shows that in descending rank of mass contribution, acetaldehyde ($CH_3CHO$), $C_3H_4$, isoprene ($C_5H_8$), acetic acid + glycolaldehyde ($C_2H_4O_2$), and acetone + propanal ($C_3H_6O$) are the major contributors for biogenic factor indicating that leaf wounding compounds contribute significantly to the BVOC burden in Delhi (Portillo-Estrada et al., 2015). The signal at m/z 41.035 can potentially be attributed to $C_3H_4$ the 2-methyl-3-butene-2-ol fragment (Kim et al., 2010; Park et al., 2013) a known fragment of isoprene (Yuan et al., 2017). Figure 8 shows that this factor explains the largest percentage share of two BVOCs namely isoprene + 2-methyl-3-butene-2-ol fragment, and its oxidation product, methyl vinyl ketone, methacrolein, and 2-butenal. It also explains the largest percentage share of C6 amides ($C_6H_{13}NO$) which are produced by the photo-oxidation of amines (Yao et al., 2016). The potential precursor, C6-amines has previously been detected in forested environments (You et al., 2014). However, it is also possible that C-6 amides are only attributed to the biogenic factor because their diurnal concentration profile matches that of first-generation oxidation products, and the source strength is high during both monsoon and post-monsoon season. This type of time series would also be expected if the precursors of this oxidation product are emitted from agricultural activities.

### 3.3 Comparison with emission inventories

Figure 10 shows a comparison of different anthropogenic emission inventories with the PMF output data from this study for three overlapping fetch regions corresponding to the fetch region from which air masses will reach the receptor site within 24 hours for different airflow patterns (Figure 1).

One feature that stands out in this comparison is that all inventories appear to significantly overestimate the relative contribution of residential fuel usage to the VOC and particulate matter emissions for all fetch regions. In absolute terms, the

Regional Emission Inventory in Asia (REAS v3.2.1) for the year 2015 (Kurokawa & Ohara, 2020) and the Emission Database for Global Atmospheric Research (EDGARv6.1) for the year 2018 (Crippa et al., 2022), agree on the residential sector $PM_{2.5}$ emissions for the NW fetch region (Table S5). According to the latest estimates (Pandey et al., 2021), the NW-IGP region has the lowest prevalence of solid fuel usage in the entire IGP and the inventories appear to overestimate the $PM_{2.5}$ emissions from this fetch region only by a factor of 1.5-1.9. For the SW and SE fetch region, respectively, REAS v3.2.1 estimates much larger residential sector $PM_{2.5}$ emissions than EDGARv6.1 and overestimates the PMF estimates by a factor of 3.7 and 4.6. In contrast, EDGARv6.1 only overestimates PMF estimates by a factor of 1.8 and 3.2, for the SW and SE fetch region respectively. Solid fuel-based cooking is more prevalent in both Central and Western India and the Eastern IGP than in the NW-IGP (Pandey et al., 2021). The overestimation in both inventories may be caused by a gradual adoption of cleaner technology. Sharma et al., (2022) calculated a 13 % drop in residential sector $PM_{2.5}$ emissions between 2015 and 2020 due to higher LPG sales and a continuation of that trend to 2022 could explain the overestimation of residential fuel usage in the present emission inventory data. For $PM_{10}$, the EDGARv6.1 emission estimates for the NW, SW, and SE fetch region are greater than the REASv3.2.1 emission inventory. The EDGARv6.1 and REASv3.2.1 inventory both overestimate our PMF $PM_{10}$ results by a factor of 1.5 to 3.0. However, while the REASv3.2.1 inventory appears to assume that most of the residential sector aerosol emissions occur in the fine mode, our PMF results (Fig. 10) clearly agree with the EDGARv6.1 inventory on the fact that there are significant coarse aerosol emissions associated with solid-fuel based cooking and heating. Table S5 shows that for residential sector VOC emissions, the absolute emissions in the EDGARv6.1 inventory are almost twice as large as those in the REASv3.2.1 inventory, even though the percentage contribution of this sector to the VOC emissions in the inventory in Figure 10 appears to be similar for both, because of larger VOC emissions from solvent use and industries in the EDGARv6.1 inventory. Both inventories overestimate the relative importance of residential sector emissions in relation to VOC emissions from other sectors by more than a factor of two when compared to our PMF estimate, most likely because they have not been updated with recent fuel shifts towards LPG in the relatively prosperous Delhi NCR region (Sharma et al., 2022).

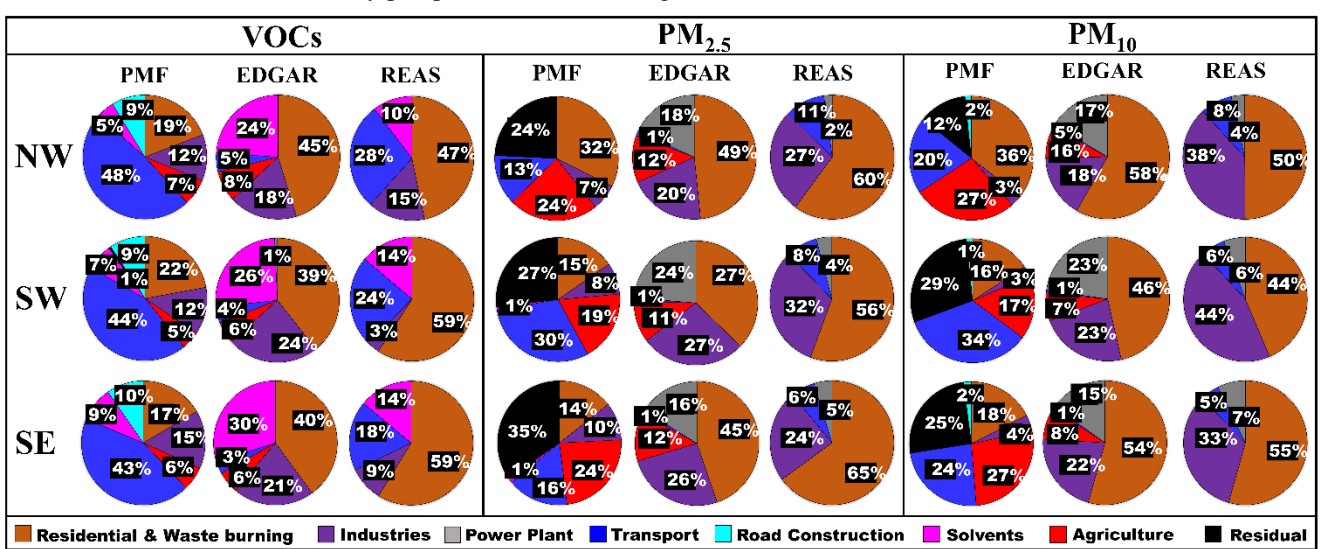

**Figure 10: Comparison of different anthropogenic emission inventories with the PMF output from this study for three overlapping fetch regions corresponding to different airflow patterns.**

With respect to industrial emissions of VOCs for the NW fetch region, our PMF results indicate that the actual emissions are slightly smaller than those in the REASv3.2.1 inventory, while the EDGARv6.1 inventory overestimates emissions. For the SW and SE fetch region, our PMF estimates fall in between those of the EDGARv6.1 inventory and the REASv3.2.1 inventory. For industrial $PM_{2.5}$ emissions, both EDGARv6.1 & REASv3.2.1 are close and agree on the magnitude of emissions for the NW, SW, and SE fetch region, respectively, and both inventories appear to overestimate emissions when compared to our PMF results. Our findings seem to suggest that the pollution boards have been somewhat successful in clamping down on industrial emissions and the technology employed is better than what is currently reflected in emission inventories. Industrial fly ash ($PM_{10}$) emissions are larger in the REASv3.2.1 inventory for all the fetch regions compared to EDGARv6.1 inventory. Yet both inventories appear to significantly overestimate industrial emissions when compared to our PMF results. These findings also indicate the pollution boards have been somewhat successful in clamping down on large and visible fly ash sources and that the EDGARv6.1 inventory has captured this clean-technology transition better.

The REASv3.2.1 inventory completely misses direct VOC and PM emissions from the agricultural sector. The EDGARv6.1 inventory significantly underestimates $PM_{2.5}$ & $PM_{10}$ emissions from agricultural activities, which include, but are not limited to crop residue burning, in comparison to our PMF results, particularly over NW-India (Table S5). Over this fetch region EDGARv6.1 attributes as much $PM_{2.5}$ to all agricultural activities combined for the full year as the FINNv2.5 inventory (Wiedinmyer et al., 2023) attributes just to agricultural residue burning activities taking place between 15th August and 26th November 2021 (a time period comparable to the period in our model run), without including the emissions from rabi crop residue burning in summer (Kumar et al., 2016) and other agricultural activities such as harvest and ploughing. For $PM_{10}$ the fire count based FINNv2.5 estimate is twice as high as the emission estimate of EDGARv6.1 for this fetch region, and more likely to be correct, because the phytoliths present in rice straw form coarse mode ash during the combustion process (Figure S10). The fact that EDGAR appears to underestimate residue-burning emissions over this fetch region has been flagged earlier (Pallavi et al., 2019; Kumar et al., 2021; Singh et al., 2023). Our PMF analysis also reveals that the relative contribution of agricultural residue burning to the PM burden over the North-Western IGP (24 % and 27 % of $PM_{2.5}$ and $PM_{10}$, respectively) and South-Eastern IGP (24 % and 27 % of $PM_{2.5}$ and $PM_{10}$, respectively) is comparable, despite the much lower fire counts over the South-Eastern IGP (17,810), when compared to the North Western IGP (61,334). This indicates that either fires to the SE are burning closer to the receptor site or the fire detection efficiency in this fetch region is lower. Table S5 reveals that the relative importance of agricultural emissions over the SE fetch region is even more severely underestimated in the FINNv2.5 inventory than in the EDGARv6.1 inventory due to poorer fire detection (close to 100 % omission error) for the partial burns prevalent over this region (Lui et al. 2019; 2020, Figure S8) when compared to the complete burns prevalent over the NW IGP (Lui et al. 2019; 2020, Figure S7).

Transport sector VOC emissions appear to be severely underestimated in the EDGARv6.1 inventory for the NW, SW, and SE fetch region, which has been previously flagged for earlier versions of the same inventory (Sarkar et al., 2017; Pallavi et al.,

2019; Singh et al., 2023). The REASv3.2.1 inventory also underestimates our PMF results. This indicates that the contribution of the transport sector to ambient VOC pollution levels in a megacity like Delhi may not be adequately reflected in both the emission inventories. Our PMF suggests that the overall contribution of the transport sector to the total $PM_{2.5}$ and $PM_{10}$ pollution levels occurs primarily due to non-exhaust emissions from the CNG-fuelled public transport fleet. These non-exhaust emissions are much larger than what is accounted for both in the EDGARv6.1and REASv3.2.1 inventories for $PM_{2.5}$ & $PM_{10}$ emissions from the NW, SW, and SE fetch region. The transport sector-related findings of this PMF source apportionment study are in agreement with earlier source apportionment studies that often attributed a quarter or more of the total PM emissions to the transport sector. Some prior studies used metals like Pb and/or OC/EC as transport sector activity tracers (Jain et al., 2017, 2020; Sharma et al., 2016, Jaiprakash et al., 2016; Sharma & Mandal, 2017), while others attributed almost the entire HOA component of organic aerosol to transport sector emissions (Reyes-Villega et al., 2021; Cash et al., 2021; Kumar et al., 2022, Shukla et al., 2023) or used a Chemical Mass Balance (CMB) model with source fingerprints from the EPA database (Nagar et al., 2017). Our PMF results differ to emission-inventory-based assessments, which only attribute a minor share of the total PM burden to this activity (Guo et al., 2017). Our findings also add insights to the reasons why the transport sector targeted air quality interventions yielded such poor results (Chandra et al., 2018). Public transport availability was ramped up during the periods when road-rationing schemes restricted the use of private 4-wheelers. Our results suggest that moving forward only investments into the road infrastructure that reduce resuspension, modal shifts from buses towards metro-based public transport and electric vehicles with >50 % regenerative braking (Liu et al., 2021) that limit brake wear can yield meaningful reductions in the transport sector-related PM emissions.

Our PMF results indicate that solvent usage results in VOC emissions that are more in line with the REASv3.2.1 inventory while the EDGARv6.1 inventory overestimates emissions by a factor of 4 for all the fetch regions.

Power generation is not considered to be a significant VOC source in both emission inventories (<1 % of the total VOC mass), and fails to show up as a separate sector in our PMF results, as our model runs rely on VOC tracers to track pollution sources. The contribution of energy generation towards the PM burden particularly in the EDGARv6.1 emission inventory, however, is significant. It is however, striking to note that the PMF features a residual that is of similar magnitude as the $PM_{2.5}$ and $PM_{10}$ emissions attributed to power generation in the EDGARv6.1 inventory. Power generation is believed to be the dominant source of secondary sulfate aerosol (Atabakhsh et al., 2023), which is the largest contributor to the secondary inorganic aerosol burden in monsoon season (Catch et al., 2021).. It is hence likely, that much of our PMF residual can be attributed primarily to this source. While a portion of this residual, particularly during post monsoon season, may also be secondary ammonium nitrate, to which power generation, transport sector and industrial NOx and NH3 emissions contribute (Alanen et al., 2017; Link et al., 2017), ammonium nitrate formation is not thermodynamically favoured during the warm months of the year. The amount of emissions attributed to power generation in the REASv3.2.1 inventory is much smaller than those reflected in EDGARv6.1, likely because the inventory misses several coal generation units that were commissioned between 2015-2018.

Our PMF results identify road construction and asphalt pavements as an additional VOC source that is at present not reflected in emission inventories.

**4 Conclusions**

This study presents source-apportionment results derived from application of the positive matrix factorization model to a recently acquired high-quality dataset of $PM_{2.5}$ & $PM_{10}$, and 111 VOCs measured using the new PTR-TOF-MS10K enhanced volatility instrument, during monsoon and post-monsoon seasons of 2022, from one of the world's most polluted megacities: Delhi. We found that the top ranked major emission source of gas phase and aerosol phase differed from each other, highlighting the complexity of air pollution sources in such atmospheric environments. While fresh paddy burning was a negligible source of VOCs (6 %), it was the largest source of $PM_{2.5}$ & $PM_{10}$ (23 % & 25 %) in the Delhi NCR regions during our study period, likely because combustion of phytolith containing rice straw triggers the formation of coarse mode ash (Figure S10) that contributes significantly to the PM burden. $PM_{2.5}$ & $PM_{10}$ are the two main criteria air pollutants regulated under the national ambient air quality standard that are thought to be the leading cause of the air pollution emergency in November in Delhi annually (Khan et. al., 2023). The strong correlation of $PM_{2.5}$ & $PM_{10}$ with same-day fire counts, and VOC emission signatures of fresh paddy burning plumes showed that fires burning in and within the vicinity of Delhi-NCR and plumes that reached the receptor on the same day were the strongest contributory source of the high pollution levels, compared to plumes from more distant states such as Punjab and Pakistan Punjab. Both are located north-west of Delhi-NCR and were thought to be the stronger contributors to the pollution levels because the detected fire activity is more prevalent there. Furthermore, $PM_{2.5}$ & $PM_{10}$ emissions from residential heating and waste burning (24 % & 23 %) rival those from crop residue burning, and unlike paddy residue burning emissions, which are episodic, this activity persists into winter. While popular perception generally blames burning in Punjab for the high particulate matter burden due to paddy stubble burning, our PMF reveals that despite the much lower fire counts over the Eastern IGP (17,810) when compared to the North Western IGP (61,334) both are a significant source of paddy stubble burning PM in the NCR region. Also, sources that are generally targeted by most clean air action plans such as tailpipe exhaust emissions of private vehicles and industries are responsible for less than one-quarter of the particulate matter mass loading that can be traced with the help of gas-phase organic molecular tracers. Instead, the transport sector's PM emissions are dominated by the non-exhaust emissions such as road dust suspension, break wear, and tire wear of the CNG-fueled commercial vehicle fleet, which according to a recent emission inventory for Delhi are one order of magnitude larger than the transport sector tailpipe exhaust emissions (Nagpure et al., 2016).

The PMF results based on primary in-situ data indicate that the EDGARv6.1 inventory provides a better representation of emissions than the REASv3.2.1 inventory for most sectors, with the exception of transport sector emissions and VOC emissions from solvent use. Agricultural burning emissions over the NW-IGP are best represented in FINNv2.5 (Wiedinmyer et al., 2023) while agricultural emissions over the SE-IGP are better captured by EDGARv6.1. At present none of the residential sector inventories appears to have incorporated the change in the magnitude and spatial patterns due to the recent adoption of cleaner cooking technology interventions since 2018. Transport sector non-exhaust emissions are still absent (REASv3.2) or

underestimated (EDGARv6.1) in all inventories. For VOC emissions from solvent usage, REASv3.2 provides better emissions than EDGARv6.1. There is also a road construction sector in our PMF results which has a significant (9-10 %) contribution to the VOC burden but hasn't been addressed in any of the emission inventories so far, and our study by including measurements of specific molecular markers of this activity has been able to shed new strategic insights concerning this missing source.

A considerable portion of the $PM_{10}$ (18 %) and $PM_{2.5}$ (28 %) load is connected to residual sources, not directly related to combustion tracers. This contribution is likely due to windblown dust transported over long distances (Pawar et al., 2015) as well as secondary inorganic aerosols like ammonium sulfate and ammonium nitrate whose precursors are primarily emitted from power plants. Despite including the most comprehensive set of organic species measured in Delhi to date, our study does not include similar information about these other species.

Residential heating and waste burning was identified as one of the largest contributors to PM pollution, and this source is active year-round with strengths varying depending on seasonality. The total contribution of residential sector solid fuel usage and waste burning (17 % in Delhi and 18 % in Mohali) to the VOC burden during post-monsoon season was similar at both sites. So, targeting these through improved access to cleaner energy sources for heating and cooking would likely improve air quality significantly in other seasons. Future similarly designed quantitative studies would be needed to confirm this hypothesis.

The findings and insights from this study emphasize the necessity for a comprehensive, multi-sectoral approach to reduce primary emissions. While several recent efforts in some sectors (e.g. residential biofuel and cooking) appear to have yielded emission reduction benefits, the narrative to blame the post-monsoon pollution exclusively on the more visible sources (e.g. paddy residue burning), needs to be corrected so other sources are also mitigated. Our findings support the assertions of (Ganguly et al., 2020), who have pointed out previously that, rather than solely focusing on specific sources like agricultural residue burning or transport emissions, it's crucial to address the disparity between the primary targets of clean air action plans and the actual dominant sources of particulate matter. Future action plans need to account for more targeted and impactful pollution control measures and also a more comprehensive approach to address the diverse urban mixed sources highlighted in this study, such as industries and residential solid fuel/waste burning, non-exhaust road emissions, and emissions from road construction.

This new approach of combining VOC tracers with PM measurements provides great potential for improved source apportionment in complex emission environments, at a level of detail that is more meaningful than just attributing emissions to biomass burning or fossil-fuel burning, which has been the case in all previous studies from the region to date. Previously in Delhi-NCR region, Kumar et al. 2022 identified "cooking-related OA" using EESI-TOF analysis but due to analytical limitations, the paper only reported quantitative data for three primary factors, namely HOA, BBOA-1, and BBOA-2, without naming the activities responsible for the formation of BBOA-1 and BBOA-2. One of the more comprehensive AMS-based studies (Cash et al., 2021) spanning pre-monsoon, monsoon and post monsoon season of the year 2018 only identified three different primary biomass burning factors, namely cooking organic aerosol (6 % of $PM_1$), solid fuel organic aerosol (≤11 % of $PM_1$), and semi-volatility biomass burning organic aerosol (≤13 % of $PM_1$), that broadly appear to correspond to our solid

fuel-based cooking (4 % of $PM_{10}$), residential heating and waste burning (23 % of $PM_{10}$), and paddy residue burning (25 % of $PM_{10}$) factors. However, the study failed to name and attribute two of these three factors in policy relevant ways, could not

identify the significant contribution of coarse mode fly ash to the total aerosol burden, and also was unable to distinguish between different fossil-fuel related sources. Our study design which captured contrasts between clean-monsoon and polluted-post-monsoon air, and included measured VOC source fingerprints and molecular tracers enabled us to distinguish paddy-residue burning from other biomass burning sources, and resolve similar traffic emission sources (e.g. 2-wheelers from 4-wheelers and CNG vehicles). This provides a significant advance over existing source-apportionment studies and its

application would be of great relevance in other complex emission environments suffering from high air pollution where quantitative knowledge of sources can lead to evidence-based emission reduction prioritization efforts and a better understanding of the atmospheric chemistry of polluted environments around the world.

**Data availability**

PMF model simulations and input data can be obtained by contacting Baerbel Sinha.

**Author Contribution**

Arpit Awasthi: Data curation, Formal analysis, Investigation, Software, Visualization, Writing – original draft preparation. Baerbel Sinha: Conceptualization, Data curation, Formal analysis, Methodology, Project administration, Resources, Software, Supervision, Validation,review & editing, Writing – review & editing. Haseeb Hakkim: Data curation, Formal analysis, Investigation, Writing – review & editing. Sachin Mishra: Data curation, Formal analysis, Investigation. Varkrishna M.:

Investigation. Gurmanjot Singh: Investigation. Sachin D. Ghude: Resources. Vijay Kumar Soni: Resources. N. Nigam: Resources. Vinayak Sinha: Conceptualization, Data curation, Project administration, Methodology, Supervision, Writing – review & editing. M. Rajeevan: Resources.

**Competing Interests**

The authors declare that they have no conflict of interest.

**Acknowledgements:**

We acknowledge the financial support given by the Ministry of Earth Sciences (MOEs), the government of India, to support the RASAGAM (Realtime Ambient Source Apportionment of Gases and Aerosol for Mitigation) project at IISER Mohali vide grant MOES/16/06/2018-RDEAS Dt. 22.6.2021. S.M acknowledges IISER Mohali for Institute PhD fellowship. AA acknowledges MoE for PMRF PhD fellowship. We thank Dr. R. Mahesh, Dr. Gopal Iyengar, Dr. R. Krishnan (Director, IITM

Pune), Prof. Gowrishankar (Director, IISER Mohali), Dr. M. Mohapatra (DG, IMD), Dr. M. Ravichandran (Secretary Ministry of Earth Science) for their encouragement and support. We thank student members of the Atmospheric Chemistry and Emissions (ACE) research group and Aerosol Research Group (ARG) of IISER Mohali and IITM Pune in particular AkashVispute and PrassanaLonkarand local scientists of IMD for their local logistics support. The authors gratefully acknowledge the NASA/ NOAA Suomi National Polar-orbiting Partnership (Suomi NPP) and NOAA-20 satellites VIIRS fire count data used in this publication. The authors gratefully acknowledge the NOAA Air Resources Laboratory (ARL) for the provision of the HYSPLIT transport and dispersion model used in this publication.

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
