# Peer review of "Biomass burning sources control ambient particulate matter but traffic and industrial sources control volatile organic compound emissions and secondary pollutant formation during extreme pollution events in Delhi"

_EGUsphere, 2024_

## Author Comment (AC1)

**Referee 1:**
The manuscript presents a positive matrix factorization analysis of a consequent PTR-ToF-MS dataset, to which $PM_{2.5}$ and $PM_{10}$ data were added. The general outline, scope and main conclusions are very clear. The results are interesting, and each of the 11 obtained factors is thoroughly described, backed up with external data and source profiles, and well explained.
We appreciate the referee for acknowledging the significance and content of the work, and for considering it of great interest to ACP readers.

However, I do feel that the methodology is not described enough. There should be more details on how the uncertainties were calculated. What were the uncertainties for each compound and their range?
We thank the referee for the suggestion. The overall uncertainty of each compound comprises of two components, the accuracy error and the precision error. The accuracy error was minimized with the help of 8 span calibrations using a certified calibration gas standard (Societa Italiana Acetilene E Derviati; S.I.A.D. S.p.A., Italy) that had 11 hydrocarbons at ~100 ppb, namely methanol, acetonitrile, acetone, isoprene, benzene, toluene, xylene, trimethylbenzene, and dichlorobenzene and trichlorobenzene. Additionally, to ensure accurate mass axis calibration for every acquired spectra, an internal standard namely 1,3-di-iodobenzene ($C_6H_5I_2^+$) detected at m/z 330.848 and its fragment ion [$C_6H_5I^+$]) detected at m/z 204.943 were co-injected with ambient air. The technical details of these calibrations have been discussed in greater detail in the companion paper egusphere-2024-500 and calibration plots and transmission curve can be found there. However, for the purpose of PMF runs only the random uncertainty, that is the precision error, should be included as uncertainty in the PMF. If the systematic error is accidentally included the $Q_{true}/Q_{theoretical}$ ratio can drop below 1 even for a 3-factor solution. The precision error for each m/z was calculated from the observed count rate in counts per second (cps) using Poisson statistics. This is a routine way to report the precision error of measurements recorded by systems such as electron multipliers or multichannel plates. For entering the precision error into the PMF we used the average signal in cps of the m/z for the study period to calculate the average precision error in %. We have added the following text to lines 146ff in the main manuscript:
"The accuracy error was minimized by conducting a total of 8 span calibrations throughout the study period. The details of these calibrations can be found in Mishra et al., 2024. The precision error for each m/z listed in table S1, which needs to be included into the PMF model runs, was calculated from the average observed count rate in counts per second (cps) of each m/z with the help of Poisson statistics. The detection limit was determined as 2σ of the noise observed in clean zero air. "
We also have included the precision error and detection limit used in our model runs in the supplementary Table S1.

Also, more information is needed on the PMF approach of adding $PM_{2.5}$ and $PM_{10}$ data to the VOC dataset.
While PMF has routinely been used to source apportion non-methane volatile organic compounds (NMVOCs) in the literature for a quite some time, several authors have recently pioneered the use of NM-VOC tracers in a PMF to source apportion greenhouse gases such as methane, $CO_2$ and $N_2O$ (Guha, et al. 2015, Assan et al. 2018, Schulze et al. 2018) by making use of the fact that the VOCs source-fingerprints of many combustion sources are well constrained and understood. We now extend the use of this promising new technique towards source-apportionment of $PM_{2.5}$ and $PM_{10}$. The PMF is a matrix decomposition factor analysis model that deconvolves a time series of measured species into a set of factors with fixed source fingerprints whose contributions to the input data set varies with time. This makes the model well suited to accommodate all chemical species co-emitted from the same combustion source as long as the emissions impacting the receptor site are fresh enough for the VOC fingerprint to be preserved. We have included the following text and new references in our method section and list of references, respectively:
"Several authors have recently pioneered the use of VOC tracers in a PMF to source apportion co-emitted greenhouse gasses such as methane, $CO_2$ and $N_2O$ (Guha, et al. 2015, Assan et al. 2018, Schulze et al. 2023). Since the VOCs source-fingerprints of many combustion sources are well constrained and understood, we now extend the use of this promising new technique towards source-apportionment of co-emitted $PM_{2.5}$ and $PM_{10}$. The PMF is a matrix decomposition factor analysis model that decomposes a time series of measured species into a set of factors with fixed source fingerprints whose contributions to the input data set varies with time. This makes the model well suited to accommodate all chemical species co-emitted from the same source."

Assan, S., Vogel, F.R., Gros, V., Baudic, A., Staufer, J., Ciais, P.: Can we separate industrial CH4 emission sources from atmospheric observations? - A test case for carbon isotopes, PMF and enhanced APCA, Atmospheric Environment, 187, 317-327, https://doi.org/10.1016/j.atmosenv.2018.05.004, 2018.
Guha, A., Gentner, D. R., Weber, R. J., Provencal, R., and Goldstein, A. H.: Source apportionment of methane and nitrous oxide in California's San Joaquin Valley at CalNex 2010 via positive matrix factorization, Atmos. Chem. Phys., 15, 12043–12063, https://doi.org/10.5194/acp-15-12043-2015, 2015.

Schulze, B., Ward, R. X., Pfannerstill, E. Y, Zhu, Q., Arata, C., Place, B., Nussbaumer, C., Wooldridge, P., Woods, R., Bucholtz, A., Cohen, R. C., Goldstein, A. H., Wennberg, P. O., Seinfeld, J. H.: Methane Emissions from Dairy Operations in California's San Joaquin Valley Evaluated Using Airborne Flux Measurements, *Environ. Sci. Technol.* 2023, 57, 48, 19519–19531, https://doi.org/10.1021/acs.est.3c03940, 2023.

What were the steps leading to the solution (how many runs, how was the base case chosen if these runs gave different solutions.

The following text was added to the manuscript:

"The model was initiated for 20 base runs with the recommended block size of 379, and the run with the lowest $Q_{robust}$ and $Q_{true}$ was chosen for further analysis and display in Figure 2."

Were there any challenges with this approach?

No there are no challenges with this approach. Once the number of factors in the PMF approaches the true number of major sources, the PMF output becomes very stable with minimal differences between the different base runs and even minimal differences in the factor time series and percentage of VOCs explained by individual factors.

In the abstract you mention "our novel source apportionment method", but it is not very clear in the paper how novel or different it is.

The approach of source-apportioning PM sources with the help of high-time resolution measurements and better understood VOC tracers instead of highly fragmented AMS mass spectra or low time resolution offline aerosol samples is novel. We have revised following text in line 45ff to highlight the novelty:

"Several authors have recently pioneered the use of VOC tracers in a PMF to source apportion co-emitted greenhouse gasses such as methane, $CO_2$ and $N_2O$ (Guha, et al. 2015, Assan et al. 2018, Schulze et al. 2023). Since the VOCs source-fingerprints of many combustion sources are well constrained and understood, we now extend the use of this promising new technique towards source-apportionment of co-emitted $PM_{2.5}$ and $PM_{10}$. The PMF is a matrix decomposition factor analysis model that decomposes a time series of measured species into a set of factors with fixed source fingerprints whose contributions to the input data set varies with time. This makes the model well suited to accommodate all chemical species co-emitted from the same source."

Also, you mention that the factors are stable in the bootstrap repetitions; however, the uncertainties of the model in Figure 3 seem quite important.

As noted in the figure caption, we have plotted the $2\sigma$ uncertainty of the model in Figure 3. Hence the error bars may look-worse than they are because many authors typically report $1\sigma$ error bars or fail to include any error bars in their factor profiles. When the PMF model actually behaves in an unstable manner, it is typical to see uncertainties in excess of 100% of the mass assigned, due to factor swapping during bootstrap runs. The average uncertainty of VOCs that are present with a loading of $>1~\mu gm^{-3}$ in factor profile in our PMF runs is only 25%. The largest uncertainty bars belonged to source fingerprints which in some cases have a large vehicle-to-vehicle and fire-to-fire variability of the emission factor for certain compounds. However, upon introspection in response to the reviewer's comment, we note that it may be more appropriate to report the error of the mean of the emission factor, rather than the vehicle-to-vehicle and fire-to-fire variability as uncertainty in this figure, and we have updated the figure accordingly:

[Figure]

Figure 1: PMF factor profile of the 11 factors identified. The source profile in μg m⁻³ (left in red) and the normalized source fingerprint of grab samples collected at the source (right in various colours). The Error bars indicate the 2σ uncertainty range from the bootstrap runs for PMF factor profiles and the 1σ error of the mean of the emission factors for source samples.

Also, the contribution of factors (i.e. paddy, residential) for $PM_{2.5}$ and $PM_{10}$ changes a lot when the number of factors varies, suggesting they may not be very stable.

It is important to understand that deductive reasoning models like the PMF suffer from large artefacts when their basic assumptions (in the case of the PMF the assumption on the minimum number of sources affecting the receptor) are heavily violated. Until the PMF opens distinct factors for the industrial OVOC emissions in the 7-factor solution, the PMF compromises between accommodating the industrial OVOC emissions in these two source profiles and explaining the biomass burning PM emissions in the model. The root cause is that certain OVOCs such as organic acids, methanol, acetone and acetaldehyde, which are a very characteristic part of the source fingerprint of different biomass burning sources, originate from diverse sources. Apart from being BVOCs, these compounds can also be photo chemically formed, used as solvents, and are emitted by industrial sources. Till the PMF opens distinct factors for the industrial emissions of these compounds in the 7-factor solution, the partitioning between paddy residue burning PM emissions and heating and waste burning PM emissions in the model remains unstable. Once the industrial OVOC emissions have their own factor, this split becomes stable. Biomass burning sources are major sources of organic acids, methanol, acetone and acetaldehyde sources and these two factors are most "agreeable" towards accommodating the additional industrial OVOCs emissions (and BVOC emissions and the photochemical source) in their source profiles, till a separate factor for each of the above sources is opened up in the PMF. The shift in the VOC source fingerprints that occur as and when each of the above gets its own factor are most visible in Figure S4. Once all of the above

including the industrial OVOC emissions have their own independent factor profile in the PMF, the amount of PM attributed to paddy residue burning and the VOC source fingerprint of the source become stable in the PMF solution. The amount attributed to residential heating and waste burning stabilizes after a separate factor for cooking emissions opens up in the 9-factor solution. The following text was inserted into line 174 to make this clearer:

"Until the PMF opens distinct factors for the industrial OVOC emissions in the 7-factor solution, the partitioning between paddy residue burning and heating and waste burning $PM_{2.5}$ and $PM_{10}$ emissions in the model remains unstable, because these sources with their strong OVOC emissions are most agreeable to accommodating additional OVOC sources in their fingerprint at the expense of explaining the $PM_{2.5}$ and $PM_{10}$ emissions. Once the industrial OVOC emissions have their own factor, this split becomes stable. The amount of PM attributed to residential heating and waste burning stabilizes after a separate factor for cooking emissions opens up in the 9-factor solution."

Do you have other information to back up the factors' stability (i.e., low time-series correlations between the factors)?

Yes. Out of 55 possible factor pairs 51 factor pairs have an R<0.5 and 49 R<0.4 while 4 have an R between 0.5 and 0.6. No pair displays an R>0.6. We also have additional support to back up that these two factors are distinct and real. The intensity of the paddy burning factor correlates with the same day fire counts in the 24-h fetch region (R=0.8) and the burning decreases by the time wheat sowing is almost complete. The heating and waste disposal factor keeps on increasing proportional to the increase in heating demand towards the onset of winters and it shows an R=0.8 with the 24-h averaged heating demand. Their time series correlation displays an R=0.5 on account of both activities being negligible in monsoon and active in post monsoon season and both being most active in the early evening hours. The time series correlation of hourly averaged data is not necessarily a highly diagnostic tool that can be used in isolation to identify whether or not factors are genuine, as R values in the range of 0.5-0.6 can be accomplished merely because two sources share the same diurnal patter such as high concentration values at night when emissions mix into a shallow nocturnal boundary layer and lower values during the day when the boundary layer is well mixed. This is particularly true if one of the two sources is as ubiquitous as traffic. The highest R values for any factor pairs in our 11-factor solution occur for the correlation between the industrial source and two traffic sources. It displays R=0.59 and 0.56 with 4-wheelers and 2-wheelers, respectively. This happens despite the fact that the industrial source emissions primarily reach the receptor from what appears to be point sources located in the wind sector SE to SW of the site, while both 4-wheeler and 2-wheeler emissions reach the site every night and from all wind directions. 4-wheeler and 2-wheeler also show R=0.51 with each other because both type of emission occur simultaneously on the same roads. The following text was modified in line 180:

"Therefore, the 11-factor solution, which showed R<0.6 for all possible factor pairs, was analyzed further."

How do the scaled residuals change when increasing the number of factors or between different runs?

The scaled residual outside the -3 σ to + 3 σ range decreases in an exponential pattern with the increase in the number of factors. Since this is a large dataset, their number is still large (10^3 observations) in the 11-factor solution. This information has been added to Fig. S5 and the figure is referenced as in the text as follows:

"Figure S5 shows how the $Q_{true}/Q_{theoretical}$ ratio and $Q_{robust}/Q_{theoretical}$, and scaled residuals beyond 3 standard deviations drop exponentially when the number of factors increases. It can be seen that initially the $Q_{true}/Q_{theoretical}$ ratio drops faster than $Q_{robust}/Q_{theoretical}$ ratio on account of additional major plumes being better explained with each additional factor. However, with the increase from 11 to 12 factors both drop in a parallel fashion indicating that the point of diminishing returns has been reached."

The comparison of the PMF output with emission inventory results needs more justification. If I understand correctly, PMF results are concentrations and seem to be directly compared to emissions, which are in different quantities and on different scales.

While evaluating the percentage contribution of different sources to the burden of specific pollutants such as $PM_{2.5}$ over a fetch region that is reasonable and related to the atmospheric lifetime of the pollutant in question, the comparison can be considered valid. After all, the lifetime for any given VOC such as benzene is independent of its source. Hence the percentage share each source contributes to the measured burden at a site should be proportional to the percentage share the different sources within the fetch region contribute to emissions, provided that the emissions are correctly represented in the emission inventory and the fetch region is chosen suitably small to ensure that emissions from a source within the fetch region can reach the receptor without significant loss. In this study, we are not comparing the absolute concentrations of the PMF and emission inventories, but rather a relative percentage contribution of sources to the total burden. This approach has been routinely used at many other sites of the world (e.g. Buzcu-Guven and Fraser, 2008 https://doi.org/10.1016/j.atmosenv.2008.02.025, Morino et al. 2011, https://doi.org/10.1029/2010JD014762 Li

et. al., 2019 https://doi.org/10.5194/acp-19-5905-2019; Qin et al,. 2022 https://doi.org/10.1007/s11356-022-19145-7, ) to compare PMF outputs with emission inventories. The reason why absolute concentrations are also sometimes brought into the discussion is that at times the look of pie charts can be deceptive, as is the case e.g. for industrial $PM_{2.5}$ emissions. Both the EDGAR and the REAS inventory have almost identical industrial $PM_{2.5}$ emissions in the inventory, yet the pie charts look visibly different, because the larger energy sector emissions and the presence of agricultural burning emissions in the EDGAR inventory visually shrink the size of that "pie slice" compared to how it looks like for the REAS inventory. Looking at the absolute numbers helps to resolve which inventory is more likely to be wrong and more importantly for which source. To address the reviewer's concerns, we have now reworded section 2.5 in the Materials and Methods section and have added more justification and references. It now reads as follows:

"The observational data was grouped according to the predominant airflow into a south-westerly, north-westerly, and south-easterly group, and the fetch region from which air masses would reach the receptor site within 24 h was determined for each group separately spanning latitude 21–31˚N and longitude 72–82˚E, latitude 28–32˚N and longitude 72–80˚E and latitude 25–30˚N and longitude 75–88˚E, respectively, for the three flow regimes. Two gridded emission inventories namely the Emission Database for Global Atmospheric Research (EDGARv6.1) for the year 2018 (Crippa et al., 2022), and the Regional Emission inventory in Asia (REAS v3.2.1) for the year 2015 (Kurokawa & Ohara, 2020) were filtered for these three fetch regions to compare PMF results with the emission inventory. We compare the relative percentage contribution of sources to the total atmospheric pollution burden in the PMF with the relative percentage contribution of sources to the total emissions for the emission inventories. This approach has been routinely used to evaluate emission inventories with the help of PMF results at different sites around the world (Buzcu-Guven and Fraser, 2008, Morino et al. 2011, Sarkar et al., 2017; Li et. al., 2019, Qin et al., 2022). For the purpose of emission inventory comparison of anthropogenic sources, natural sources such as biogenic emissions and the photochemistry factor were removed from the PMF output, while the solid fuel-based cooking and residential heating and waste burning emissions were summed up in residential & waste management. In addition, CNG and Petrol 2 & 4-wheeler factors were combined into the consolidated transport sector emissions."

The conclusions drawn here seem too strong (i.e. lines 536-539). Also, please justify why the PMF results are more correct than model outputs? (i.e. when you state that sources are under-/over-estimated in the models).
The PMF results are based on the primary data acquired at the airshed site. On the other hand, emission inventories rely on activity data and emission factors that is often lags behind in terms of updation by few years and therefore less well constrained. E.g. residential emissions are at times simply scaled with the increase in the population without adjusting for fuel shifts, while transport sector emissions may be scaled with fuel sales without accounting for the shift to lower emission control technologies (e.g. Euro-6=BS-VI). Due to this, routine updates often fail to encompass the technological advancements as well as measures effected by the policy change in a particular region. Hence, we assume that the results arising from the direct ambient measurements are closer to the reality of 2022 than emission inventories that have last been updated in 2015. However, we appreciate the reviewer's valid comment and now support each of the points we are making with more references to supporting literature with similar findings. In lines 536-539 the text now reads as follows
"Table S6 shows that for residential sector VOCs emissions, the absolute emissions in the EDGARv6.1 inventory are almost twice as large as those in the REASv3.2.1 inventory, even though the percentage contribution of this sector to the VOC emissions in the inventory in Figure 10 appears to be similar for both, because of larger VOC emissions from solvent use and industries in the EDGARv6.1 inventory. Both inventories overestimate the relative importance of residential sector emissions in relation to VOC emissions from other sectors by more than a factor of two when compared to our PMF estimate, most likely because they have not been updated with recent fuel shifts towards LPG in the relatively prosperous Delhi NCR region (Sharma et al., 2022)."
Lines 550-560 containing statements about the agricultural sector emissions in various inventories have also been revised as follows including by addition of new Figure S10 showing coarse mode aerosol from the use of paddy in an industrial burner:
"The EDGARv6.1 inventory significantly underestimates $PM_{2.5}$ & $PM_{10}$ emissions from agricultural activities, which include, but are not limited to crop residue burning, in comparison to our PMF results, particularly over NW-India (Table S6). Over this fetch region EDGARv6.1 attributes as much $PM_{2.5}$ to all agricultural activities combined for the full year as the FINNv2.5 inventory (Wiedinmyer et al., 2023) attributes just to agricultural residue burning activities taking place between 15th August and 26th November 2021 (a time period comparable to the period in our model run), without including the emissions from rabi crop residue burning in summer (Kumar et al., 2016) and other agricultural activities such as harvest and ploughing. For $PM_{10}$ the fire count based FINNv2.5 estimate is twice as high as the emission estimate of EDGARv6.1 for this fetch region, and more likely to be correct, because the phytoliths present in rice straw form coarse mode ash during the combustion process (Figure S10). The fact that EDGAR appears to underestimate residue-burning emissions

over this fetch region has been flagged earlier (Pallavi et al., 2019; Kumar et al., 2021; Singh et al., 2023). Our PMF analyses also reveals that the relative contribution of agricultural residue burning to the PM burden over the North-Western IGP (24 % and 27 % of $PM_{2.5}$ and $PM_{10}$, respectively) and South-Eastern IGP (24 % and 27 % of $PM_{2.5}$ and $PM_{10}$, respectively) is comparable, despite the much lower fire counts over the South-Eastern IGP (17,810), when compared to the North Western IGP (61,334). This indicates that either fires to the SE are burning closer to the receptor site or the fire detection efficiency in this fetch region is lower. Table S6 reveals that the relative importance of agricultural emissions over the SE fetch region is even more severely underestimated in the FINNv2.5 inventory than in the EDGARv6.1 inventory due to poorer fire detection (close to 100% omission error) for the partial burns prevalent over this region (Lui et al. 2019, 2020, Figure S8) when compared to the complete burns prevalent over the NW IGP (Lui et al. 2019, 2020, Figure S7)."

[Figure]

**Figure S10 SEM image of rice ash from the electrostatic precipitator of an industrial boiler fired with rice husk and straw illustrating the coarse mode nature of the ash generated during the combustion of phytolith containing biomass.**

Be more concise when you present the description of the factors, the fact that all the values and VOC m/z are written in the main text makes it tedious to read. Use only VOC names (or formula if unclear what the compound is, but the m/z are already listed in Table S1). Also, delete all the concentration and % values in the main text if they are already on the figures, except if it is useful to emphasize the point (example in line 632: "a considerable portion of the $PM_{10}$ (18%) and $PM_{2.5}$ (28%)"). Same for $log_{10}C_0$, find a clearer way to present them. Another option would be to put the extensive description of factors in SI and a summary and interpretation in the main text.

We appreciate the referee's suggestion and have made the requested changes to the manuscript. Since the number of changes is large we are not listing each individual one into the response file. However, we chose to retain the factor descriptions in the main text. The $log_{10}C_0$ values are now presented in a new supplementary figure (Figure S9) that also helps to address the reviewer's comment regarding SOA formation from these factors.

[Figure]

**Figure S9: Volatility oxidation state plots for all factors that individually contribute more than 3% to the total SOA formation potential.**

**Specific comments/questions**

Line 87-88: I would suggest adding a map of the receptor site with the surroundings (i.e. roads, industries, agriculture…), and referencing it when needed.

We have added a detailed map of the receptor site with its surroundings to the supplement as Figure S1 and have referenced it in the text as follows:

Ambient air was sampled into the instruments from the roof-top of a tall building (28.5896°N-77.2210°E) at ~35 m above ground, located within the premises of the Indian Meteorological Department (IMD) at Lodhi Road, New Delhi situated in Central Delhi. The sampling site is a typical urban area surrounded by green spaces, government offices, and residential areas, but not in the direct vicinity of any major industries (Fig. S1)

[Figure]

**Figure S1: Map of the immediate surroundings of the IMD (28.5896°N-77.2210°E) sampling site in Central Delhi. (Google Earth Imagery ©Google Earth)**

Line 108: I think it would be worth summarizing the main differences between the 3 wind sectors (in terms of typology, specificity, and later on results).

We appreciate the referee's suggestions, hence a map has been added to Figure 1 and we have shifted the description from section 3.3 to section 2.1. The relevant sentences now read as follows:

"Figure 1 shows the location of the site and the 120 h back trajectories of air masses arriving at the site that were grouped according to the dominant synoptic regional scale transport into a) south-westerly (orange and yellow) flows carrying emissions from southern Punjab, Haryana, Uttar Pradesh, Madhya Pradesh, Rajasthan and Gujarat towards the receptor, b) north-westerly (light and dark blue) flows carrying emissions from Pakistan Punjab, Indian Punjab, Haryana, Western Uttar Pradesh, Himachal Pradesh, and Uttarakhand towards the receptor, and c) south-easterly flows (light and dark red) carrying emissions from Haryana, Southern Uttarakhand, Uttar Pradesh, Bihar and Nepal towards the receptor. Figure 1d shows a Google Earth image with a spatial map of the daily fire counts in the region for the post-monsoon season alongside with the maximum 24-h fetch region for each of these synoptic flow situations marked by coloured square. Figure 1e-h shows the e) photosynthetic active radiation, f) daily fire counts in the fetch region (21-32°N, 72-88°E), g) temperature and relative humidity, and h) the ventilation coefficient and the sum of the daily rainfall during the study period (15th August 2022– 26th November 2022)."

Section 3.1 & Figure 3: I would suggest putting Figure 3 in supplementary and replacing it with only this study's factors profiles in concentration (instead of normalized).

We are now displaying the factor profiles in concentration units on the left-hand axis of Figure 3 in the main text. However, we have chosen to retain the visual comparison with the source profiles. To do so we have shifted towards showing the source profiles on a secondary axis which continues to be normalized, because many individual panels have mixed units (e.g. samples from a traffic junction in the units µg/m3 and tailpipe exhaust with the units g/kg). It is important to note normalization does not alter the fingerprint of the PMF output and does not affect the R of the cross-correlation analysis between source samples and PMF output either. It just permits us to easily combine things in different units and sources with different absolute emission intensity into one plot.

However, we assume that the spirit of the reviewers' suggestion is related to the fact that the figure is a little congested. Hence, we reduced the number of source profiles shown in addition to the PMF fingerprint to at most 3 per panel to reduce the congestion in this figure. The revised Figure 3 looks as follows:

[Figure]

**Figure 2: PMF factor profile of the 11 factors identified. The source profile in µg m⁻³ (left in red) and the normalized source fingerprint of grab samples collected at the source (right in various colours). The Error bars indicate the 2σ uncertainty range from the bootstrap runs for PMF factor profiles and the 1σ error of the mean of the emission factors for source samples.**

In text 3.1, I would add the R correlation (of profile and/or diurnal cycle) of this study's factors with the mentioned reference factors to justify the factors' interpretation.

We thank the reviewer for this suggestion and have added the R of the correlation of the source profiles of PMF output with the source fingerprints of the source samples in the text in this section. The revised text now reads as follows.

"Figure 3 shows the source profile of the eleven factors that our PMF analyses resolved. Out of the 111 VOCs only those whose normalized source contribution exceeded 0.1 when divided by the most abundant compound in the same source profile in at least one of the sources, were included in the figure. The source identity of the PMF factors was confirmed by matching the PMF factor profiles with the unit $\mu g\ m^{-3}$ with normalized source fingerprints of grab samples collected from the potential sources. To facilitate the comparison of emission factors and grab samples from different studies with the PMF output, the source samples were normalized by dividing each species' mass/emission factor by the mass/emission factor of the most abundant species in a given fingerprint. The PMF factor profile matched best against source samples collected from burning paddy fields (R=0.6, Kumar et al., 2020) for the paddy residue burning factor. The cooking factor matched emissions from a cow-dung-fired traditional stove called angithi (R=0.7, Fleming et al., 2018). The residential heating & waste burning factor had a source fingerprint matching emission from leaf litter burning, (R=0.7, Chaudhary et al., 2022), waste burning (R=0.7, Sharma et al., 2022), and cooking on a chulha fired with a mixture of firewood and cow dung (R=0.9, Fleming et al., 2018). The factors identified as CNG (R=1.0), petrol 4-wheelers (R=0.9), and petrol 2-wheelers (R=0.6) matched tailpipe emissions of the respective vehicle types and fuels (Hakkim et al., 2021). The petrol 4-wheelers (R=0.9), and petrol 2-wheelers (R=0.7) also matched traffic junction grab samples from Delhi (Chandra et al., 2018). The OVOC source fingerprint of the road construction factor matched the source fingerprint of asphalt mixture plants and asphalt paving (R=0.9, Li et al., 2020), while the hydrocarbon source fingerprint matched diesel-fuelled road construction vehicles (R=0.6, Che et al., 2023). The factors identified as solvent usage and evaporative emissions matched ambient air grab samples collected from an industrial area at Jahangirpuri (R=0.7), and Dhobighat at Akshar Dham (R=0.5) in this study. The factor identified as industrial emissions showed the greatest similarity to ambient air grab samples from the vicinity of the Okhla waste-to-energy plant (R=0.8), Gurugram (R=0.7) and Faridabad (R=0.8) industrial area. The biogenic factor showed the greatest similarity to leaf wounding compounds released from Populus tremula (R=0.8, Portillo-Estrada et al., 2015) as well as BVOC fluxes from Mangifera indica (R=0.4, Datta et al., 2021)."

Sections 3.1 & 3.2: Since you have a dedicated subsection for the comparison of the sources with references, you don't have to repeat them when describing each factor.

Thank you for the helpful suggestion. We have significantly shortened the text of section 3.1 lines 246-254 and where appropriate the description of individual factors and are now avoiding repetition of text and numbers between section 3.1 and 3.2.:

Figure 4 shows the relative contribution of different sources to the total pollution burden of VOCs, $PM_{2.5}$ and $PM_{10}$ at the receptor site. In the megacity of Delhi, transport sector sources contributed most (42±4 %) to the total VOC burden, while it contributed much less (only 24 %) to the total VOC burden in Mohali a suburban site 250 km North of Delhi during the same season (Singh et al., 2023). On the other hand, the contribution of paddy residue burning (6±2 %) and the summed residential sector emissions (17±3 % in Delhi and 18 % in Mohali) to the total VOC burden during post-monsoon season were similar at both sites. The contribution of the different factors to the SOA formation potential (Fig. 4e), stands in stark contrast to their contribution to primary particulate matter emissions. SOA formation potential was dominated by the transport sector (54 %) while direct $PM_{10}$ (52%) and $PM_{2.5}$ (48%) emissions were dominated by different biomass burning sources (Fig. 4 b & c). CNG-fuelled vehicles also contribute significantly to the $PM_{10}$ (15±3 %) and $PM_{2.5}$ (11±3 %) burden. A significant share of the $PM_{10}$ (18 %) and $PM_{2.5}$ (28 %) burden is associated with the residual and not directly linked to combustion tracers. This share can likely be attributed to windblown dust arriving at the site through long-range transport (Pawar et al., 2015) and to secondary organic, and secondary inorganic aerosols such as ammonium sulphate and ammonium nitrate. Due to the complex relationship of secondary aerosol with gas-phase precursors and emission tracers, VOC tracers are not a suitable tool to source-apportion this aerosol component. Meteorological conditions, homogeneous, heterogeneous, and multiphase chemistry control how fast primary emissions are converted to secondary aerosol. To explain the source of those species, one also needs to invoke the physicochemical and thermodynamical properties of the aerosol. (Acharja et al., 2022).

Figure 3: How were the displayed compounds chosen for this graph? And please use the compounds' names so that it is clearer.

We display all compounds whose normalized mass is at least 0.1 in at least one of the factor profiles to limit the number of species displayed and keep the figure legible. We have now included this information in the text. We prefer not to name compounds, since particularly at the higher m/z there can be many different chemical compounds with the same monoisotopic mass. Hence, we felt it better to consistently use the chemical formula in the figures. We are discussing names alongside the chemical formula where appropriate in the text of section

3.2, but a figure x-axis is not the appropriate place accommodate a differentiated discussion of possible names. Hence, we retain the chemical formula instead. The revised text now reads:

"Figure 3 shows the source profile of the eleven factors that our PMF analyses resolved. Out of the 111 VOCs only those whose normalized source contribution exceeded 0.1 when divided by the most abundant compound in the same source profile in at least one of the sources, were included in the figure."

I would suggest adding Figure S3 in the main text as it is referenced a lot, and that way you don't need to put the % in the main text.

Thank you for the kind suggestion. In accordance with the referee's suggestion, we have removed the percentages from the text, and added Figure S3 to the main text. In response to the editor's comments on our manuscript, we have converted each panel to a separate Figure which is now being referenced as Figure 6 to Figure 9. The revised text segments read as follows:

Section 3.2.1

[revised manuscript text omitted]

Line 224: "The source identity of the PMF factors was confirmed by matching the normalized PMF factor profiles with normalized source fingerprints". Could you add more detail about this, did you check the R correlations? Or was it just by visually comparing them?
Yes, we have done an R correlation of the source profiles with the PMF factors. We have Now included R values in the manuscript. The revised text reads as follows
"Figure 3 shows the source profile of the eleven factors that our PMF analyses resolved. Out of the 111 VOCs only those whose normalized source contribution exceeded 0.1 when divided by the most abundant compound

in the same source profile in at least one of the sources, were included in the figure. The source identity of the PMF factors was confirmed by matching the PMF factor profiles with the unit µg m$^{-3}$ with normalized source fingerprints of grab samples collected from the potential sources. To facilitate the comparison of emission factors and grab samples from different studies with the PMF output, the source samples were normalized by dividing each species' mass/emission factor by the mass/emission factor of the most abundant species in a given fingerprint. The PMF factor profile matched best against source samples collected from burning paddy fields (R=0.6, Kumar et al., 2020) for the paddy residue burning factor. The cooking factor matched emissions from a cow-dung-fired traditional stove called angithi (R=0.7, Fleming et al., 2018). The residential heating & waste burning factor had a source fingerprint matching emission from leaf litter burning, (R=0.7, Chaudhary et al., 2022), waste burning (R=0.7, Sharma et al., 2022), and cooking on a chulha fired with a mixture of firewood and cow dung (R=0.9, Fleming et al., 2018). The factors identified as CNG (R=1.0), petrol 4-wheelers (R=0.9), and petrol 2-wheelers (R=0.6) matched tailpipe emissions of the respective vehicle types and fuels (Hakkim et al., 2021). The petrol 4-wheelers (R=0.9), and petrol 2-wheelers (R=0.7) also matched traffic junction grab samples from Delhi (Chandra et al., 2018). The OVOC source fingerprint of the road construction factor matched the source fingerprint of asphalt mixture plants and asphalt paving (R=0.9, Li et al., 2020), while the hydrocarbon source fingerprint matched diesel-fuelled road construction vehicles (R=0.6, Che et al., 2023). The factors identified as solvent usage and evaporative emissions matched ambient air grab samples collected from an industrial area at Jahangirpuri (R=0.7), and Dhobighat at Akshar Dham (R=0.5) in this study. The factor identified as industrial emissions showed the greatest similarity to ambient air grab samples from the vicinity of the Okhla waste-to-energy plant (R=0.8), Gurugram (R=0.7) and Faridabad (R=0.8) industrial area. The biogenic factor showed the greatest similarity to leaf wounding compounds released from Populus tremula (R=0.8, Portillo-Estrada et al., 2015) as well as BVOC fluxes from Mangifera indica (R=0.4, Datta et al., 2021)."

Line 235-236: Did you measure the Munirka furniture market and Dhobighat at Akshar Dham samples? If not, could you add their reference?
Yes, all source-samples not referenced were collected by us. Meanwhile we have collected more samples and found a more relevant match for the solvent factor and on a plot situated opposite a polymer manufacture and next to a pet food manufacturer have updated the figure with better matches. We now also clearly state that samples were collected by us:
The factors identified as solvent usage and evaporative emissions matched ambient air grab samples collected in an industrial area at Jahangirpuri (R=0.7), and Dhobighat at Akshar Dham (R=0.5) in this study.

Figure5: I would suggest enlarging (by the x axis) the timeseries plot, to make them easier to read. You should keep the same order of the factors as in description (& throughout the paper). What do the lines/shaded areas for the diurnal cycles represent (mean, median…)?
We have modified the plot and figure caption as per the suggestions:

[Figure]

**Figure 3: Time series of each factor in µg m⁻³ (left column) with respective normalized diurnal profiles (centre column). The shaded region in the diurnal profiles depicts the area between the 25th and 75thand percentile while the median of the dataset is marked as the line. The polar plots (right column) depict the conditional probability of a factor having a mass contribution above the 75th percentile of the dataset during a certain hour of the day between midnight (centre of rose) and 23:00 local time (outside of rose) from a certain wind direction. This probability is determined by dividing the number of observations above the 75th percentile by the total number of measurements in each bin.**

3.2.2. There is a mention that this factor may not be always fresh, which I found interesting, you could add a few words at the end of the paragraph about the fresh/aged nature of the factors based on all the information.

We appreciate the suggestions. This assessment was primarily based on the fact that this night time factor shows a lower R with NO than with NO₂. Our comment was primarily meant to contrast this factor with some of the transport sector and industrial emissions that have a much higher R with NO than NO₂ indicating the night time plumes of these factors are so fresh that their atmospheric lifetime is more likely on the scale of minutes rather than in hours, while heating and waste disposal plumes are occasionally fresh but often also aged. We have modified the text in line 330 to clarify as follows:

"The lower correlation with NO (R=0.4) (Table S5), indicated that emissions are combustion-related but not always fresh. Occasionally, fresh plumes reach the receptor within minutes, however the majority of plumes have a higher atmospheric age, as NO is a short-lived species and oxidized to $NO_2$ on the timescale of minutes in the presence of ozone"

3.2.3. Some of these compounds (i.e. aromatics) can also be associated with cooking activities (e.g. Crippa et al (2013), doi.org/10.5194/acp-13-8411-2013).
Thank you for the suggestion. The reference has been added to the manuscript.
These aromatic compounds have been reported to originate from cooking emissions (Crippa et al., 2013).

Crippa, M., Canonaco, F., Slowik, J. G., El Haddad, I., DeCarlo, P. F., Mohr, C., Heringa, M. F., Chirico, R., Marchand, N., Temime-Roussel, B., Abidi, E., Poulain, L., Wiedensohler, A., Baltensperger, U., and Prévôt, A. S. H.: Primary and secondary organic aerosol origin by combined gas-particle phase source apportionment, Atmos. Chem. Phys., 13, 8411–8426, https://doi.org/10.5194/acp-13-8411-2013, 2013.

3.2.4. You could add one sentence about the interpretation of VOCs (i.e. $CH_3OH$ $CH_3OH$ methanol and ethanol) for this factor.
We appreciate the suggestions. We have modified the text in the manuscript as follows:
"Figure 7 shows that the factor explains the largest percentage share of $CH_3OH$ methanol and the second largest percentage share of $C_2H_6O$ ethanol. These compounds are formed by the incomplete combustion of CNG that is catalytically converted to methanol and ethanol (Singh et al., 2016)."

Singh, S., Mishra, S., Mathai, R., Sehgal, A. K., & Suresh, R.: Comparative study of unregulated emissions on a heavy duty CNG engine using CNG & hydrogen blended CNG as fuels. SAE Int. J. Engines, 9(4), 2292-2300, http://dx.doi.org/10.4271/2016-01-8090, 2016.

3.2.5. & 3.2.6. Add a sentence (or change existing text) to highlight the differences between 2-wheeler & 4-wheeler factors.
We appreciate the suggestions. We have revised the manuscript to better contrast the two. Firstly petrol 4-wheeler emissions are on average much fresher as central Delhi is a prosperous neighbourhood dominated by private cars. Petrol 2-wheeler plumes are on average more aged. Section 3.2.5 now starts as follows:
"Figure 4 shows petrol 4-wheeler contributed to 20 %, 25 %, and 30 % to the VOC mass loading, OFP, and SOAP, respectively. The source fingerprint of this source matched tailpipe emissions of petrol-fuelled 4-wheelers (Hakkim et al., 2021) and is characterized, in descending rank of contribution, by C8-aromatics, toluene, C9-aromatics ($C_9H_{12}$), benzene, butene + methyl tert-butyl ether (MTBE) fragment, propyne, propene, methanol and C2-substituted xylenes + C4-substituted benzenes ($C_{10}H_{14}$). Figure 5 shows that emissions peak in the evening between 7 pm and midnight with average VOC mass loadings >70 µg m$^{-3}$ and reach the receptor site from most wind directions. Emissions are strongly correlated with NO (R=0.8), CO (R=0.7), and $CO_2$ (R=0.7) indicating the receptor site is impacted by fresh combustion emissions from this source and the atmospheric age of most plumes is on the timescale of minutes."
Section 3.2.6 now starts as follows:
"Figure 4 shows petrol 2-wheeler contributed to 14 %, 12 %, and 20 % to the VOC mass loading, OFP, and SOAP respectively. The source fingerprint of this source matched tailpipe emissions of petrol-based 2-wheelers (Hakkim et al., 2021) and are characterized, in descending rank of contribution, by toluene, acetone + propanal, C-8 aromatic compounds, acetic acid ($C_2H_4O_2$), propyne ($C_3H_4$), methanol ($CH_3OH$), benzene ($C_6H_6$), the MTBE fragment and C-9 aromatics ($C_9H_{12}$). A key difference of the petrol 2-wheeler source profile in comparison to the petrol 4-wheeler source profile is the lower benzene to toluene ratio, which is supported by the GC-FID analysis of tailpipe exhaust (Kumar et al., 2020). Figure 5 shows that emissions peak in the evening between 8 pm and 10 pm with average VOC mass loadings >50 µg m$^{-3}$ and reach the receptor site from most wind directions. Emissions are strongly correlated with $NO_x$ (R=0.6), CO (R=0.6) and $CO_2$ (R=0.7), but have a lower correlation with NO (R=0.5) (Table S5), and a larger contribution of oxygenated compounds to the source profile, indicating that the emissions have been photochemically aged. This suggests that contrary to 4-wheeler plumes which originate from the immediate vicinity of the site in central Delhi (Figure S1), 2-wheeler plumes reach the receptor after prolonged transport from more distant rural and suburban areas on the outskirts of the city. In such areas, people often favour two-wheelers over four-wheelers."

Line 422: Interesting! Could this last sentence mean that part of $PM_{2.5}$ for this factor would be SOA?
Yes, likely most of it because PM10=PM2.5 for this factor. We have added a sentence to this effect along with a volatility oxidation state plot for this factor.
"Figure S9 shows the volatility oxidation state plot for all 111 VOCs in which the marker size represents the

percentage share of each compound explained by the industrial factor and markers are colour coded by the number of carbon atoms. The plot shows evidence of the first- and second-generation oxidation products of C6 to C10 hydrocarbon transitioning from the VOC to the IVOC range along trajectories expected for the addition of =O functionality to the molecule (Jimenez, et al. 2009). This and the fact that the entire aerosol associated with this factor is $PM_{2.5}$, indicates that most of the aerosol associated with this factor is likely SOA."

[Figure]

**Figure S9: Volatility oxidation state plots for all factors that individually contribute more than 3% to the total SOA formation potential.**

Jimenez, J. L., Canagaratna, M. R., Donahue, N. M., Prevot, A. S. H., Zhang, Q., Kroll, J. H., DeCarlo, P. F., Allan, J. D., Coe, H., Ng, N. L., Aiken, A. C., Docherty, K. S., Ulbrich, I. M., Grieshop, A. P., Robinson, A. L., Duplissy, J., Smith, J. D., Wilson, K. R., Lanz, V. A., Hueglin, C., Sun, Y. L., Tian, J., Laaksonen, A., Raatikainen, T., Rautiainen, J., Vaattovaara, P., Ehn, M., Kulmala, M., Tomlinson, J. M., Collins, D. R., Cubison, M. J., Dunlea, E. J., Huffman, J. A., Onasch, T. B., Alfarra, M. R., Williams, P. I., Bower, K., Kondo, Y., Schneider, J., Drewnick, F., Borrmann, S., Weimer, S., Demerjian, K., Salcedo, D., Cottrell, L., Griffin, R., Takami, A., Miyoshi, T., Hatakeyama, S., Shimono, A., Sun, J. Y., Zhang, Y. M., Dzepina, K., Kimmel, J. R., Sueper, D., Jayne, J. T., Herndon, S. C., Trimborn, A. M., Williams, L. R., Wood, E. C., Middlebrook, A. M., Kolb, C. E., Baltensperger, U., and Worsnop, D. R.: Evolution of Organic Aerosols in the Atmosphere, Science, 326, 1525-1529, https://doi.org/10.1126/science.1180353, 2009.

Line 432-433: Do you have references for this last statement?
Yes, we have added references to this statement.
"Figure 8 shows that the factor explains the largest share of organic acids namely butanoic acid, acetic acid and isocyanic acid (HNCO) and the second largest share of butanal + butanone + MEK ($C_4H_8O$). These compounds point towards stack venting of VOCs from chemical-, food-, or pharmaceutical industries or polymer manufacturing as likely sources of these emissions (Hodgson et al., 2000, Villberg et al., 2001, Jankowski et al., 2017, Gao et al., 2019). This assessment is broadly confirmed by the fact that the best source match for this source was collected from a plot situated opposite a polymer manufacture and next to a pet food manufacturer in an industrial area at Jahangirpuri (R=0.7) N of the receptor site."

Gao, Z., Hu, G., Wang, H., Zhu, B.: Characterization and assessment of volatile organic compounds (VOCs) emissions from the typical food manufactures in Jiangsu province, China, Atmos. Pollut. Res. 10(2), 571-579, https://doi.org/10.1016/j.apr.2018.10.010, 2019.
Hodgson, S. C., Casey, R. J., Bigger, S. W., & Scheirs, J.: Review of volatile organic compounds derived from polyethylene. Polym-Plast Technol, 39(5), 845-874. https://doi.org/10.1081/PPT-100101409, 2000.
Jankowski, M. J., Olsen, R., Thomassen, Y., & Molander, P.: Comparison of air samplers for determination of isocyanic acid and applicability for work environment exposure assessment. Environm. Sci-Proc. Imp., 19(8), 1075-1085, https://doi.org/10.1039/C7EM00174F, 2017.
Villberg, K., & Veijanen, A.: Analysis of a GC/MS thermal desorption system with simultaneous sniffing for determination of off-odor compounds and VOCs in fumes formed during extrusion coating of low-density polyethylene. Anal. Chem. 73(5), 971-977.https://doi.org/10.1021/ac001114w, 2001.

3.2.9. Interesting, the last sentence suggests a possible link of the OVOCs with SOA?
Yes. We have added a statement to this effect.
This assessment is supported by the volatility oxidation state plot for the road transport factor (Figure S10) which demonstrates that both precursors and oxidation products are present in this factor and that C6 to C10 hydrocarbons appear to be progressing from the VOC to the IVOC range along trajectories expected for the addition of =O functionality to the molecule (Jimenez, et al. 2009).

Jimenez, J. L., Canagaratna, M. R., Donahue, N. M., Prevot, A. S. H., Zhang, Q., Kroll, J. H., DeCarlo, P. F., Allan, J. D., Coe, H., Ng, N. L., Aiken, A. C., Docherty, K. S., Ulbrich, I. M., Grieshop, A. P., Robinson, A. L., Duplissy, J., Smith, J. D., Wilson, K. R., Lanz, V. A., Hueglin, C., Sun, Y. L., Tian, J., Laaksonen, A., Raatikainen, T., Rautiainen, J., Vaattovaara, P., Ehn, M., Kulmala, M., Tomlinson, J. M., Collins, D. R., Cubison, M. J., Dunlea, E. J., Huffman, J. A., Onasch, T. B., Alfarra, M. R., Williams, P. I., Bower, K., Kondo, Y., Schneider, J., Drewnick, F., Borrmann, S., Weimer, S., Demerjian, K., Salcedo, D., Cottrell, L., Griffin, R., Takami, A., Miyoshi, T., Hatakeyama, S., Shimono, A., Sun, J. Y., Zhang, Y. M., Dzepina, K., Kimmel, J. R., Sueper, D., Jayne, J. T., Herndon, S. C., Trimborn, A. M., Williams, L. R., Wood, E. C., Middlebrook, A. M., Kolb, C. E., Baltensperger, U., and Worsnop, D. R.: Evolution of Organic Aerosols in the Atmosphere, Science, 326, 1525-1529, https://doi.org/10.1126/science.1180353, 2009.

3.3. It's a little tedious to read with all the emission values, please select when it is truly important to have them.
We appreciate the referee's suggestions and added the numbers as Table S5 to the supplement. We have revised the text to reduce the numbers. It now reads as follows:

[revised manuscript text omitted]

Lines 500-505: A map could be useful here as well.

We appreciate the referee's suggestions and added the map to Figure 1.

[Figure]

**Figure 4: 120 h back trajectory air mass reaching receptor site at Mausam Bhawan building (28.5896°N-77.2210°E, 50 m above ground level) grouped according to the dominant synoptic scale transport into a) South-Westerly, b) North-Westerly, and c) South-Easterly flow. Square boxes indicate the fetch region from which air masses typically reach the receptor site within 24 hrs for a given flow situation with the d) spatial map of the daily fire counts in the region for the post-monsoon season. The bottom panels show the e) photosynthetically active radiation, f) daily fire counts in the fetch region, g) temperature and relative humidity, and h) the ventilation coefficient and the sum of the daily rainfall for the study period.**

Line 536-539: "our PMF results indicate that the actual emissions are slightly smaller than those" "our PMF estimates fall in between those of the EDGARv6.1 inventory and the REASv3.2.1 inventory" I don't understand how you come to these conclusions, did you calculate emissions out of the PMF concentrations? If yes, please state. If not, I don't think you can directly compare PMF results and emissions, only in terms of contributions to the total "measured" compounds for each method.

While evaluating the percentage contribution of different sources to the burden of specific pollutants such as PM2.5 over a fetch region that is reasonable and related to the atmospheric lifetime of the pollutant in question,

the comparison can be considered valid. After all, the lifetime of a given VOC (e.g. benzene) is independent of their source. Hence the percentage share each source contributes to the measured burden at a site should be proportional to the percentage share the different sources within the fetch region contribute to emissions, provided that the emissions are correctly represented in the emission inventory and the fetch region is chosen suitably small to ensure that emissions from a source within the fetch region can reach the receptor without significant loss. In this study, we are not comparing the absolute concentrations of the PMF and emission inventories, but rather a relative percentage contribution of sources to the total burden. This approach has been routinely used at many other sites of the world (e.g. Buzcu-Guven and Fraser, 2008 https://doi.org/10.1016/j.atmosenv.2008.02.025, Morino et al. 2011, https://doi.org/10.1029/2010JD014762 Li et. al., 2019 https://doi.org/10.5194/acp-19-5905-2019; Qin et al,. 2022 https://doi.org/10.1007/s11356-022-19145-7, ) to compare PMF outputs with emission inventories. The reason why absolute concentrations are also brought into the discussion is, that at times the look of pie charts can be deceptive as is the case e.g. for industrial $PM_{2.5}$ emissions. Both the EDGAR and the REAS inventory have almost identical industrial $PM_{2.5}$ emissions in the inventory, yet the pie charts look visibly different, because the larger energy sector emissions and the presence of agricultural burning emissions in the EDGAR inventory visually shrink the size of that "pie slice" compared to how it looks like for the REAS inventory. Looking at the absolute numbers helps to resolve which inventory is more likely to be wrong and for which source. We have reworded this paragraph to make it clearer that we are comparing the relative contribution to the total VOC burden with the relative contribution to the total emissions for the inventory.

Table S6 shows that for residential sector VOCs emissions the absolute emissions in the EDGARv6.1 inventory are almost twice as large as those in the REASv3.2.1 inventory, even though the percentage contribution of this sector to the VOC emissions in the inventory in Figure 10 appears to be similar for both, because of larger VOC emissions from solvent use and industries in the EDGARv6.1 inventory. Both inventories overestimate the relative importance of residential sector emissions in relation to VOC emissions from other sectors by more than a factor of two when compared to our PMF estimate, most likely because they have not been updated with recent fuel shifts to LPG in the relatively prosperous Delhi NCR region.

Line 551: "The EDGARv6.1 inventory significantly underestimates $PM_{2.5}$ & $PM_{10}$ from agricultural activities" Please backup this statement with a map for example to justify that agricultural emissions should be high.
We have already backed up this statement with numbers and a comparison to the FINNv2.5 inventory. Now we also included fire counts in a map in Figure 1, have added images of ash from paddy burning and have simplified the text to make it clearer as follows:
"The REASv3.2.1 inventory completely misses direct VOC and PM emissions from the agricultural sector. The EDGARv6.1 inventory significantly underestimates $PM_{2.5}$ & $PM_{10}$ emissions from agricultural activities, which include, but are not limited to crop residue burning, in comparison to our PMF results, particularly over NW-India (Table S6). Over this fetch region EDGARv6.1 attributes as much $PM_{2.5}$ to all agricultural activities combined for the full year as the FINNv2.5 inventory (Wiedinmyer et al., 2023) attributes just to agricultural residue burning activities taking place between 15th August and 26th November 2021 (a time period comparable to the period in our model run), without including the emissions from rabi crop residue burning in summer (Kumar et al., 2016) and other agricultural activities such as harvest and ploughing. For $PM_{10}$ the fire count based FINNv2.5 estimate is twice as high as the emission estimate of EDGARv6.1 for this fetch region, and more likely to be correct, because the phytoliths present in rice straw form coarse mode ash during the combustion process (Figure S10). The fact that EDGAR appears to underestimate residue-burning emissions over this fetch region has been flagged earlier (Pallavi et al., 2019; Kumar et al., 2021; Singh et al., 2023). Our PMF analyses also reveals that the relative contribution of agricultural residue burning to the PM burden over the North-Western IGP (24 % and 27 % of $PM_{2.5}$ and $PM_{10}$, respectively) and South-Eastern IGP (24 % and 27 % of $PM_{2.5}$ and $PM_{10}$, respectively) is comparable, despite the much lower fire counts over the South-Eastern IGP (17,810), when compared to the North Western IGP (61,334). This indicates that either fires to the SE are burning closer to the receptor site or the fire detection efficiency in this fetch region is lower. Table S6 reveals that the relative importance of agricultural emissions over the SE fetch region is even more severely underestimated in the FINNv2.5 inventory than in the EDGARv6.1 inventory due to poorer fire detection (close to 100% omission error) for the partial burns prevalent over this region (Lui et al. 2019, 2020, Figure S8) when compared to the complete burns prevalent over the NW IGP (Lui et al. 2019, 2020, Figure S7)."

Line 554-556: There were any more results available from FINNv2.5? "between 15th and August and 26th November 2021 alone" please clarify, was it 15/08-26/11? Then it's the same length as the current dataset…
The time period matches our observational period just that the data is for the previous year. Unfortunately, 2022 data is not yet available for download, hence we cannot match it with same year data. However, the fact that there are two main crop residue burning seasons of which only one is included in the FINN estimate but both of

which should be included in the annual EDGAR number doesn't change from year to year as can be seen in the figure below.

[Figure]

We have now clarified in the text why this period was selected.
Over this fetch region EDGARv6.1 attributes as much PM$_{2.5}$ to all agricultural activities combined for the full year as the FINNv2.5 inventory (Wiedinmyer et al., 2023) attributes just to agricultural residue burning activities taking place between 15th August and 26th November 2021 (a time period comparable to the period in our model run), without including the emissions from rabi crop residue burning in summer (Kumar et al., 2016) and other agricultural activities such as harvest and ploughing.

Table S1: You could add calculated uncertainties and detection limits here. Also, if the "Sr. No" numbers are not used, you can delete them from the table. Are the "Mean" and range values here the detection limits or the averaged concentrations throughout the campaign?

We appreciate the suggestions. We have included the precision error and detection limit used to initiate the model in the supplement Table S1. The mean value is the campaign averaged value and the range represents the minimum and maximum observed throughout the campaign. We now clarify this in the Table caption.

**Table S1: 111 NMVOCs species used in the PMF model, the table lists the major compound identifications and the references supporting such assignments from previous works, along with average of the observational period reported in this study (with range min-max), detection limits, precision error.**

Table S2 & S3: Same comment about the "Sr. No".
Deleted

Figure S1: Are these figures referenced in the paper?
Yes, these Figures now numbered as S2-S4 in response to an editors comment are referenced as follows:
Figure 2 shows how the percentage of total VOC, PM$_{2.5}$, and PM$_{10}$, attributable to various sources changes when the number of factors increases from 3 to 12, while Fig. S2-S4 illustrates the evolution in the factor contribution time series, source profile, and percentage of species explained by different sources when the number of factors in the PMF increases.

**Technical corrections**
Throughout the paper, add · in units (ex µg·m$^{-3}$) done
Title: There shouldn't be an abbreviation in the title, please use volatile organic compounds instead of VOC.
done the revised title is:
Biomass burning sources control ambient particulate matter but traffic and industrial sources control volatile organic compound emissions and secondary pollutant formation during extreme pollution events in Delhi

Line 16: There is a repetition of the word "using", please change. Revised to:
Here, we source-apportioned VOCs and PM, using a high-quality recent (2022) dataset of 111 VOCs, $PM_{2.5,}$ and $PM_{10}$ in a positive matrix factorization (PMF) model.

Line 23: Replace "(<2)" by "at least by a factor of 2".done

Line 36: Please reformulate "continues to add".Revised to:
Delhi with a population of 31.7 million people (UN World Population Prospects 2022), sees an addition of over six hundred thousand vehicles per year (2022 VAHAN-Ministry of Road Transport and Highways (MoRTH), Government of India).

Line 70: Delete the first "source" in "quantify the source contribution of the different sources".(done)

Line 80: Delete ":" in the title and check all the titles.(done)

Line 113: "in blue" aren't there other colours used on the graph too? Yes, the revised figure caption reads as follows:
Figure 5: 120 h back trajectory air mass reaching receptor site at Mausam Bhawan building (28.5896°N-77.2210°E, 50 m above ground level) grouped according to the dominant synoptic scale transport into a) South-Westerly, b) North-Westerly, and c) South-Easterly flow. d) spatial map of the daily fire counts in the region for the post-monsoon season with square boxes indicate the fetch region from which air masses typically reach the receptor site within 24 hrs for a given flow situation with the. The bottom panels show the e) photosynthetically active radiation (PAR), f) daily fire counts in the fetch region, g) temperature and relative humidity, and h) the ventilation coefficient and the sum of the daily rainfall for the study period.

Line 116: Correct to "solar radiation as photosynthetically active radiation (PAR)".(done)

Line 119: Please add the dates of monsoon and post-monsoon seasons. Revised to:
During the monsoon season (15.08-30.09.2022), the air masses from the south-west direction (western arm of the monsoon) were more prevalent than air masses reaching the site form the south-east (Bay of Bengal arm of the monsoon). During the post-monsoon season (01.10-26.11.2022) air masses remain confined over the NW-IGP for prolonged periods and primarily reach the site from the north-west (Fig. 1b), except during the passage of western disturbances (05.10-10.10.2022 and 04.11-10.11.2022), which result in brief periods with south westerly and south-easterly flow and rain (Fig. 1h).

Line 151-152: The structure of the sentence seems wrong, please correct. The sentence has been split into 2 sentences.
The US EPA PMF 5.0 (Paatero et al., 2002, 2014; Paatero & Hopke, 2009; Noris et al., 2014) was applied to a sample matrix of 2496 hourly observations and 111 VOC species. The species with S/N greater than 2.0 were designated as strong species (94) while others were designated as weak species (17).

Line 180-181: There is a repetition of the word "model", please change. This was a typo. The sentence now reads as follows:
The model was run in the constrained mode elaborately described in Sarkar et al., (2017) and Singh et al., (2023).

Line 190: "T" to delete at the beginning of the paragraph. (done)

Line 180-181190-191: There is a repetition of the word "using", please change. Revised the sentence now reads as follows:
The contribution of VOCs to ozone production was derived with the maximum incremental reactivity (MIR) (Carter, 2010) method using the following equation

Line 194: Change to "The secondary organic aerosol production (SOAP)" in small case.(done)

Line 196 & 197: Replace NOx with $NO_X$ and check this throughout the paper .(done)

Line 197-199: This sentence is a bit unclear. The sentence was split and now reads as follows:
This equation evaluates each VOC species' ability to make SOA in relation to the amount of SOA the same mass of toluene would make when introduced to the ambient environment. This is represented by the $SOAP_i$.

Line 219: Replace "while" by starting a new sentence with "In addition,".(done)

Line 220: Replace "are" with "were" and check that it is the right tense throughout the paper.(done)

Line 247: Delete "," in "(Fig. 4 a & d) were petrol". This section was shortened in response to a previous comment and now reads as follows

"Figure 4 shows the relative contribution of different sources to the total pollution burden of VOCs, $PM_{2.5}$ and $PM_{10}$ at the receptor site. In the megacity of Delhi, transport sector sources contributed most (42±4 %) to the total VOC burden, while it contributed much less (only 24 %) to the total VOC burden in Mohali a suburban site 250 km North of Delhi during the same season (Singh et al., 2023). On the other hand, the contribution of paddy residue burning (6±2 %) and the summed residential sector emissions (17±3 % in Delhi and 18 % in Mohali) to the total VOC burden during post-monsoon season were similar at both sites. The contribution of the different factors to the SOA formation potential (Fig. 4e), stands in stark contrast to their contribution to primary particulate matter emissions. SOA formation potential was dominated by the transport sector (54 %) while direct $PM_{10}$ (52%) and $PM_{2.5}$ (48%) emissions were dominated by different biomass burning sources (Fig. 4 b & c)."

Figure 4: "Photo", "P2W" & "P4W" could be written in the full name. (done)

Line 252: Delete "," between "both" & "paddy".(done)

Line 293: Put "-3" in superscript.(done)

Line 286 & l288: Delete "," in "A recent study in Punjab indicated that" and "increased by 0.027 and 0.047 $\mu g \cdot m^{-3}$ respectively".(done)

Line 357: There is a repetition of the word "identified", please change. Deleted

Line 362: I would suggest deleting the sentence "this is consistent with our results", as "confirms" in line 358 already suggests this.(done)

Line 368-369: Keep "$\mu g \cdot m^{-3}$)" on the same line.(done)

Line 383: Delete the first "source" in "The source fingerprint of this source".(done)

Line 397: Correct the start of the sentence to "This factor contributes on average more than 30 $\mu g \cdot m^{-3}$"(done)

Line 397-398: The second part of the sentence, "due to…", to reformulate and you could reference the added map of surroundings. Rephrased, the sentence now reads as follows:
"This suggests that contrary to 4-wheeler plumes which originate from the immediate vicinity of the site in central Delhi (Figure S1), 2-wheeler plumes reach the receptor after prolonged transport from more distant rural and suburban areas on the outskirts of the city. In such areas, people often favour two-wheelers over four-wheelers."

Line 399: Add space in "NO (R=0.7)" and correct "$CH_4$".done

Line 402 & 404: Once you have written full MTBE and MT, abbreviation is fine. For monoterpenes, you can also write only full name.done

Line 403: There are 2 "," after "acetaldehyde (1.2 $\mu g \cdot m^{-3}$)".This has been rephrased in response to other comments
"The main contributors towards the VOC mass in the industrial factor, are in descending order of contribution propyne ($C_3H_4$), methyl tert-butyl ether ($C_4H_8$), toluene ($C_7H_8$), C-8 aromatic compounds ($C_8H_{10}$), propene ($C_3H_6$), acetaldehyde ($CH_3CHO$), methanol ($CH_3OH$), C-9 aromatics and the sum of monoterpenes ($C_{10}H_{16}$)."

Line 415-418: This part is a little difficult to read, cf general comment about writing all the values.
We deleted the values and have instead created a figure for the supplement. The revised text reads as follows

"In addition, this factor also explains the largest percentage share of a large suite of volatile and IVOC aromatic hydrocarbons including naphthalene ($C_{10}H_8$), methyl naphthalene ($C_{11}H_{10}$), $C_{12}H_{16}$, $C_{13}H_{18}$, $C_{13}H_{20}$, $C_{13}H_{22}$, $C_{14}H_{20}$, and $C_{14}H_{22}$."

Line 438: Use "acetone + propanal" as before. Changed

Line 452-460: This part is quite difficult to read and understand, cf general comment about writing all the values.
It has been revised as follows:
As represented by Fig. 9, this factor explains the largest percentage share of a large suite of volatile and IVOC hydrocarbons namely, heptene ($C_7H_{14}$), $C_{11}H_{12}$, $C_{12}H_{12}$, $C_{14}H_{14}$, $C_{14}H_{18}$, $C_{16}H_{24}$, $C_{17}H_{28}$, and $C_{18}H_{30}$. In addition, it explains the second largest percentage share of many other IVOC hydrocarbons namely $C_9H_{14}$, $C_9H_{16}$, $C_{11}H_{14}$, $C_{12}H_{16}$, $C_{13}H_{18}$, $C_{13}H_{20}$, $C_{13}H_{22}$, $C_{14}H_{20}$, $C_{14}H_{22}$. Except for the four hydrocarbons $C_7H_{14}$, $C_9H_{14}$, $C_9H_{16}$, and $C_{11}H_{12}$, all of these IVOCs have been reported to degas at 60°C from asphalt pavement (Khare et al., 2020). So far only $C_{14}H_{18}$ has been reported as fresh gas phase emissions (transport time <2.5 min) from a farm (Loubet et al., 2022) in ambient air, while $C_{17}H_{28}$ has been reported in the aerosol phase (Xu et al., 2022). The road construction factor also explains the largest percentage share of a long list of OVOCs namely, C6 diketone isomers ($C_6H_{10}O_2$), C2-substituted phenol($C_8H_{10}O$), $C_7H_{12}O_2$, $C_8H_{14}O_2$, $C_8H_{16}O_2$, phthalic anhydride ($C_8H_4O_3$) , which is a naphthalene oxidation product (Bruns et al., 2017), $C_9H_{10}O$, $C_9H_{12}O_2$, $C_9H_{14}O_2$, $C_9H_{16}O_2$, $C_9H_{18}O_2$, $C_{10}H_{12}O$, $C_{10}H_{18}O$, $C_{10}H_8O_3$, $C_{10}H_{16}O_3$, and $C_{12}H_{18}O_2$. However, out of these only $C_{10}H_{12}O$ and $C_{10}H_{18}O$ have been detected as direct emissions from heated asphalt pavement (Khare et al., 2020) indicating that most OVOCs in this factor are possibly oxidation products of short-lived IVOCs hydrocarbons emitted by this source. This assessment is supported by the volatility oxidation state plot for the road transport factor (Figure S10) which demonstrates that both precursors and oxidation products are present in this factor and that C6 to C10 hydrocarbons appear to be progressing from the VOC to the IVOC range along trajectories expected for the addition of =O functionality to the molecule (Jimenez, et al. 2009).

Line 531-532: Keep "$y^{-1}$" on the same line. done

Line 558: Delete "to" in "Our PMF results reveal that to agricultural".done

Line 608: "two criteria air pollutants" do you mean "critical"?
No. India has a National Ambient Air Quality Standards (NAAQS) for six commonly found air pollutants known as criteria air pollutants. $PM_{10}$ and $PM_{2.5}$ are two of the six criteria for air pollutants regulated under this law. The text has been revised as follows
"While fresh paddy burning was a negligible source of VOCs (6 %), it was the largest source of $PM_{2.5}$ & $PM_{10}$ (23 % & 25 %) in the Delhi NCR regions during our study period, likely because combustion of phytolite containing rice straw triggers the formation of coarse mode ash (Figure S10) that contributes significantly to the PM burden. $PM_{2.5}$& $PM_{10}$ are the two main criteria air pollutants regulated under the national ambient air quality standard that are thought to be the leading cause of the air pollution emergency in November in Delhi annually (Khan et. al., 2023).

Line 622: What is EDGARv6.1 better than in this sentence?
Revised to:
"The PMF results based on primary in-situ data indicate that the EDGARv6.1 inventory provides a better representation of emissions than the REASv3.2.1 inventory for most sectors, with the exception of transport sector emissions and VOC emissions from solvent use. Agricultural burning emissions over the NW-IGP are best represented in FINNv2.5, while agricultural emissions over the SE-IGP are better captured by EDGARv6.1."

Line 635: Add "in Delhi": "Despite including the most comprehensive set of organic species in Delhi to date"
Revised to
"Despite including the most comprehensive set of organic species measured in Delhi to date, our study does not include similar information about these other species."

Line 644: Add "," after "that" done

Line 651: Replace "till date" by "to date" done

---

## Author Comment (AC2)

The paper *"Biomass burning sources control ambient particulate matter but traffic and industrial sources control VOCs and secondary pollutant formation during extreme pollution events in Delhi"* discusses the sources responsible for air pollution problems in Delhi. For this, they made stationary ambient gas-phase measurements at a prominent location in urban New Delhi and performed source apportionment analysis on the collected data. The chemical profiles of the factors were compared with previous measurements and tracers to identify sources. The work is quite timely since New Delhi is one of the most polluted cities in the world, and regulatory policies are currently being restricted by our limited understanding of the sources in the region.

We thank the referee for recognizing and highlighting the importance of this research work.

Yet I have significant concerns, which I think should be resolved prior to proceeding with publication. Some of my biggest concerns are with the conclusions drawn and stated quite imposingly in the conclusion section. Hence, I'll discuss those first before moving to the next major ones.

Line 606-607: fresh paddy burning is shown to be a negligible source of VOCs but the largest sources of PM2.5 and PM10. This is highly confusing to me. PM2.5 would be formed from the secondary oxidation of a lot of gas-phase organic molecules emitted from paddy burning. As such it should be emitting precursors of SOA. Or are the authors suggesting that paddy-burning directly emits particulate matter into the atmosphere but no VOCs?

Thank you for seeking this important clarification. Yes, we are suggesting that paddy straw burning is a source of primary aerosol. It is important to note that paddy straw contains a rigid, microscopic structures made of silica known as phytolite. Upon burning, this structure is converted into a glassy ash. The mass of this type of aerosol emitted during the combustion process appears to be quite high when compared to the mass of VOCs emitted in the same combustion process. The high ash formation is a well known fact in engineering circles. Co-combustion of more than 10% of paddy straw alongside with other fuels in power generation units causes severe equipment fouling, due to the potassium (K) rich glassy ash formed (Lui et al. 2022 https://doi.org/10.1016/j.energy.2022.123950, Madhiyanon et al. 2020 https://doi.org/10.1016/j.joei.2020.04.001). We have inserted a scanning electron microscopy image of ash collected from the electrostatic precipitator of an industrial boiler fired with rice husk and straw to our supplement to illustrating the coarse mode nature of the ash generated during the combustion of phytolite containing biomass.

The reality appears to be that much of the aerosol emitted during paddy residue burning is 1) primary and 2) relatively coarse. The root cause of the discrepancy between the contribution to the VOC mass and the contribution to the PM mass appears to be that the glassy ash particles are bigger and have a higher density than organic aerosol and contribute more to the total aerosol mass, that secondary aerosol particles with smaller size and lower density. Just like dust, this ash cannot be detected by AMS and since the chemical composition is >96% $SiO_2$ with minor amounts of K, any routine CMB analysis would likely attribute this type of aerosol to the natural dust fraction. This explains why earlier studies may have failed to recognize the importance of ash aerosol. We have added a figure (Figure S10) and revised the text to reflect this more clearly as follows:

[Figure]

Figure S10 SEM image of rice ash from the electrostatic precipitator of an industrial boiler fired with rice husk and straw, illustrating the coarse mode nature of the ash generated during the combustion of phytolith containing

biomass.

"While fresh paddy burning was a negligible source of VOCs (6 %), it was the largest source of $PM_{2.5}$ & $PM_{10}$ (23 % & 25 %) in the Delhi NCR regions during our study period, likely because combustion of phytolite containing rice straw triggers the formation of coarse mode ash (Figure S10) that contributes significantly to the PM burden. $PM_{2.5}$ & $PM_{10}$ are the two main criteria air pollutants regulated under the national ambient air quality standard that are thought to be the leading cause of the air pollution emergency in November in Delhi annually (Khan et. al., 2023)."

It is, however, important to note that we do not claim that paddy burning is not a VOC source. In Section 3.2., we clearly state that it is the largest source of a relatively long list of VOCs.
The main point that we are making is that in terms of its relative contribution to the overall pollution levels of certain pollutants when compared to other sectors such as e.g. road transport, this activity is far more important as a $PM_{2.5}$ and $PM_{10}$ source than it is as a VOC source/SOA precursor source. We have shifted the supplementary figure to the main text and revised the text to make this clearer. The revised text to clarify this aspect reads as follows:

"Figure 6 shows that this factor explained the largest percentage share of O-heteroarene compounds such as furfural ($C_5H_4O_2$), methyl furfural ($C_6H_6O_2$), hydroxy methyl furfural ($C_6H_6O_3$), furanone ($C_4H_4O_2$), hydroxymethyl furanone ($C_5H_6O_3$), furfuryl alcohol ($C_5H_6O_2$), furan ($C_4H_4O$), methyl furans ($C_5H_6O$), C2-substituted furans ($C_6H_8O$), and C3-substituted furans ($C_7H_{10}O$), which are produced by the pyrolysis of cellulose and hemicellulose, and have previously been detected in biomass burning samples (Coggon et al., 2019; Hatch et al., 2015; 2017; Koss et al., 2018; Stockwell et al., 2015). Figure 6 also shows that this factor explains the largest share of the most abundant oxidation products that result from the nitrate radical-initiated oxidation of toluene as well as from OH-imitated oxidation of aromatic compounds under high $NO_x$ conditions, namely nitrotoluene ($C_7H_7NO_2$) and nitrocresols ($C_7H_7NO_3$) (Ramasamy et al., 2019), which indicates a certain degree of aging of the plumes. These nitroaromatic compounds are significant contributors to SOA and BrC, (Palm et al., 2020, Harrison et al., 2005). It also explains several other nitrogen containing VOCs such as nitroethane ($C_2H_6NO_2$), the biomass burning tracer acetonitrile ($CH_3CN$) and pentanenitrile ($C_5H_9N$). The presence of pentanenitrile isomers in biomass burning smoke has previously been confirmed using gas chromatography-based studies (Hatch et al., 2015, Hatch et al., 2017). In addition the factor explains the largest percentage share of acrolein ($C_3H_4O$ ), hydroxyacetone ($C_3H_6O_2$), cyclopentadienone ($C_5H_4O$), cyclopentanone ($C_5H_8O$), diketone ($C_4H_6O_2$), pentanedione ($C_5H_8O_2$), hydroxybenzaldehyde ($C_7H_6O_2$), guaiacol ($C_7H_8O_2$), and the levoglucosan fragment ($C_6H_8O_4$), many of these compounds are known to form during lignin pyrolysis (Hatch et al., 2015, Koss et al., 2018; Nowakowska et al., 2018), while dimethylbutenedial ($C_6H_8O_2$), trimethylbutenedial ($C_7H_{10}O_2$) are ring opening oxidation products of aromatic compounds (Zaytsev et al., 2019)."

[Figure]

**Figure 6: VOC species to which different forms of biomass burning contribute the highest percentage share of the atmospheric burden in Delhi**

Is it possible that the PTR-TOF did not measure or fragment a lot of precursor species emitted from paddy burning?
 The proton transfer reaction technology is a soft ionization technique and the operating conditions of 120 Td during the deployment further facilitate negligible fragmentation. In addition, the instrument deployed in this work was equipped with extended volatility range technology which has been missing from previous PTR-TOF studies conducted in Delhi. This has been explained in detail in the companion paper (Mishra et al., 2024) and such a system enabled us to detect and measure an unprecedented long list of IVOCs emitted from other sources (industries and asphalt paving), hitherto undetected in ambient gas phase observations without fragmentation. The PMF VOC source signature further matched observational data obtained via source samples collected directly on burning paddy fields. Hence, there is no evidence of loss of VOCs. It is also important to note that we could measure a lot of SOA precursor species and some of their first-generation oxidation products e.g. nitrotoluene ($C_7H_7NO_2$), nitrocresols and ring opening oxidation products of aromatic compounds such as dimethylbutenedial ($C_6H_8O_2$), trimethylbutenedial ($C_7H_{10}O_2$) and could successfully attribute them to paddy

burning factor. However, the total VOC mass attributed to this factor 11.6 µg m$^{-3}$ is less than the PM$_{2.5}$ (20.7 µg m$^{-3}$) and PM$_{10}$ (36.5 µg m$^{-3}$) mass attributed to this factor. Since the factor has a photochemical age of less than 24 hours, and the SOA yields (in terms for % of mass converted to aerosol) for many compounds on such timescale are <20%, the overall SOA contribution to the PM$_{2.5}$ mass is smaller than the mass contribution of primary ash particles. Thus, it is highly unlikely that the PTR-TOF-MS missed measuring many precursors due to inlet losses or that the compounds fragmented massively in the system used in this work. We have added the following text and supplementary figure S9 and S10 to the end of this section to make this clearer:

"Figure S9 shows the volatility oxidation state plot for all 111 VOCs in which the marker size represents the percentage share of each compound explained by the paddy residue burning factor and markers are colour coded by the number of carbon atoms. The plot shows evidence of the first- and second-generation oxidation products of C5 and C6 hydrocarbon transitioning from the VOC to the IVOC range along trajectories expected for the addition of =O functionality to the molecule (Jimenez, et al. 2009), while C7 hydrocarbons progress along trajectories expected for both the addition of -OH and =O functionality. This indicates that paddy residue burning contributes significantly to the SOA burden. However, the fact that the PM$_{10}$ mass associated with this factor (36.5 µg m$^{-3}$) is 1.8 times larger than the PM$_{2.5}$ mass (20.7 µg m$^{-3}$) and 3 times larger than the VOC mass (11.6 µg m$^{-3}$) released during the same combustion process, points towards the relatively coarse ash formed from the phytolith skeleton of rice straw (Figure S10) as the dominant aerosol source."

[Figure]

**Figure S9: Volatility oxidation state plots for all factors that individually contribute more than 3% to the total SOA formation potential.**

Line 620 (also 566-568): "The transport sector's PM emissions are dominated by the non-exhaust emissions of the CNG-fuelled commercial vehicle fleet." This sounds somewhat unlikely. Which non-exhaust emissions are the authors referring to emitting from CNG vehicles? I can think of break/tyre-wear as a possible source but that contributes primarily to coarse PM, not so much to fine. Are there evaporative emissions of some kind? I imagine CNG itself would have negligible potential to form ambient PM given its small molecular size.

Yes, tire wear, break wear and dust re-suspension are precisely the sources we are implicating and those sources are well supported by the PMF output, because the $PM_{10}$ emissions attributed to this source in the PMF (22.5 $\mu gm^{-3}$) are indeed are twice as large as the $PM_{2.5}$ emissions attributed to this source (10.4 $\mu gm^{-3}$). According to a recent emission inventory for Delhi (Nagpure et al., 2016 https://doi.org/10.1016/j.atmosenv.2015.12.026) their contribution to transport sector PM is one order of magnitude larger than that of the tailpipe emissions and two orders of magnitude larger than the VOC mass. The SOA formation potential of the dominant VOC in the tailpipe exhaust of this vehicle class, methanol, and ethanol is very small hence SOA is not a significant contributor to the PM mass associated with the CNG factor. We have now expressed this more clearly.

"Also, sources that are generally targeted by most clean air action plans such as tailpipe exhaust emissions of private vehicles and industries are responsible for less than one-quarter of the particulate matter mass loading that can be traced with the help of gas-phase organic molecular tracers. Instead, the transport sector's PM emissions are dominated by the non-exhaust emissions such as road dust suspension, break wear and tire wear of the CNG-fueled commercial vehicle fleet, which according to a recent emission inventory for Delhi are one order of magnitude larger than the transport sector tailpipe exhaust emissions (Nagpure et al., 2016). "

On the other hand, the transport sector in Delhi would have diesel trucks which are known to be large emitters of SOA precursors.

The majority of heavy-duty vehicles in Delhi, have transitioned to CNG fuel. So have internal delivery vehicles and most taxis. There are strict restrictions on the entry of diesel trucks. This shift aligns with Delhi's strict adoption of Euro-6 norms in 2018, and restrictions that completely ban the use of more than 10-year-old diesel vehicles within city limits, which forces owners to sell these into the second hand market of less restrictive states or convert them to CNG with a conversion kit. To incentivize cleaner technologies like CNG kit (Krelling et al., 2022 https://doi.org/10.1016/j.tranpol.2021.10.019), the administration heavily subsidizes the price of CNG which was 1.5 times lower per km than that of diesel during the study period. Commercial transport is price sensitive, hence, the number of diesel vehicles on the roads is very low. Diesel-fuelled trucks typically circumvent city borders when they pass by Delhi. This practice is influenced by heavy fines on entry of old vehicles and also stringent time regulations imposed on the entry of diesel trucks. The recent changes in both regulations and their enforcement have resulted in halving the diesel sales in the Delhi NCR over the past 5 years. A random selection of pictures clicked in the timespan of less than 10 minutes while driving around Delhi pasted below supports the fact that diesel trucks are hardly plying across the city and hence not important enough to get their own PMF factor. The CNG cylinders mounted in the place where the diesel tank used to be are easily visible on most trucks. We have now inserted the following supplementary figures and text into Section 3.2.4. to clarify:

"This study attributes a large share of these non-tailpipe emissions to trucks, buses and other commercial vehicles that are typically fuelled by CNG, because commercial diesel vehicles of <10 years age face severe entry restrictions, that limit their use within the Delhi NCR while older diesel vehicles have been completely banned from plying within City limits. Policy interventions in favour of CNG use (Krelling & Badami, 2022) have resulted in a halving of diesel sales, a rapid conversion of Delhi's HDV fleet to CNG (Figure S12), and a significant reduction in tailpipe exhaust emissions."

[Figure]

**Figure S12: Random selection of photographs clicked while driving around Delhi. One can clearly see the white CNG cylinders mounted in the place where the fuel tank used to be during vehicle conversion. Photo credits: Kriti Annika Sinha**

Dust resuspension has been attributed to non-exhaust emissions, but I am not sure if I agree with that classification. Dust is not a vehicular source. Hence, I would like the authors to extensively elaborate on what forms PM from non-exhaust emissions from CNG vehicles.

Dust can be natural and windblown but it can also be anthropogenic. When dust is suspended from the road or more importantly by off-road usage of heavy vehicles e.g. during construction activity it is classified as an anthropogenic transport sector emission (see e.g. the recent review by Harrison et al. 2021, https://doi.org/10.1016/j.atmosenv.2021.118592). Particularly when it comes to avoidable road transport, these emissions would not be present if the same transport demand had been satisfied via freight train/passenger train/metro rail/tram. The road transport share of these anthropogenic emission can be reduced by 1) modal shifts of the transport demand towards rail and 2) wet road cleaning to reduce the silt burden on the road in case of road dust suspension and 3) regenerative braking in case of EVs. However, certain types of activity by HDVs that suspend a very large amount of dust e.g. during transport of construction material to construction sites or during the movement of waste are hard to eliminate. A picture of a waste disposal truck (marked with a red arrow) moving on one of the Delhi's "garbage mountains" with a dust plume (encircled with a red squre) trailing behind the vehicle is pasted below. We hope that the reviewer agrees such a "dust plume" should not be labelled as a "natural" dust plume, even though in the case of trucks moving on construction sites the suspended dust aerosol will be chemically indistinguishable from natural soil minerals suspended by wind alone. Since the anthropogenic dust is transported in the air alongside with the tailpipe emission plume, our PMF is capable of tracking this type of "dust plumes" to their primary source, namely, HDV movement.

[Figure]

We also modified the text in Section 2.4 to make things clearer:

"CNG-fuelled vehicles are identified as the third largest source of $PM_{10}$ (15 %) and $PM_{2.5}$ (11 %) and contribute 9 % to the total VOC burden (Fig. 4). The much higher contribution of this source to the coarse mode particulate matter burden (22.5 $\mu gm^{-3}$ $PM_{10}$) when compared to the fine mode particular matter burden (10.4 $\mu gm^{-3}$ $PM_{2.5}$), confirms earlier emission-inventory-based estimates which flagged that non-tailpipe emissions such as brake and tire wear and road dust resuspension have become the dominant transport sector related particulate matter sources in the Delhi-NCR region (Nagpure et al., 2016). Non-tailpipe emissions such as brake and tire wear and road dust resuspension contribute most to the $PM_{10}$ burden, although they have also become the largest source of transport sector fine mode aerosol and VOC emissions in some countries that have transitioned to Euro-6 norms (Harrison et al., 2021)."

This also reads somewhat contrary to lines 260-264 where petrol vehicles are shown to be major contributors to SOA.

We did not state that petrol vehicles contribute most to the SOA at the receptor sited. Petrol 4W and 2W vehicles contribute significantly to the SOA formation potential because they are the largest source of several aromatic compounds in the NCR. However their contribution to the $PM_{10}$ (10.8 $\mu gm^{-3}$) and $PM_{2.5}$ (4 $\mu gm^{-3}$) mass in the PMF is much smaller than that of CNG vehicles, primarily because non-tailpipe emissions such as break-wear and dust suspension scale with vehicle weight and these vehicles tend to be lightweight. It is very important to keep in mind that 1) SOA particles are smaller and 2) SOA particles are less dense (~1.4 $g/cm^3$) than break wear (up to 5 $g/cm^3$) and road dust (~2.6 $g/cm^3$) and 3) only a small percentage share of the VOC burden is converted to SOA. This is why the non-tailpipe PM emissions of vehicles that are following emissions norms of EURO 4 or better tend to be larger than the tailpipe aerosol emissions and the mass of the SOA formed. However, a large contribution to the SOA formation potential at the receptor site does not necessarily equate a large contribution to the SOA mass at the receptor site, because the 4-wheeler emissions impacting the site mostly have a photochemical age on the timescale of minutes. While the factor contributes the largest percentage share of many C6 to C10 hydrocarbons, it hardly contributes towards the measured mass of the first- and second-generation oxidation products of those very same compounds at the receptor site. We have added the following text to make this clear:

"Figure S9 shows that this factor contributes significantly to the burden of C6- to C10 hydrocarbons, and hence SOA formation potential. However, due to freshly emitted plumes, it hardly contributes to the burden of the first- and second-generation oxidation products of these hydrocarbons at the receptor site. Instead, this factor is likely to contribute to secondary pollution formation downwind of the Delhi NCR."

while a distinction has been made between 2-wheeler and 4-wheeler petrol vehicles, no significant discussion exists on the contribution of diesel vehicles. This needs to be explained in more detail.

Diesel vehicles do have a distinctive source fingerprint and would have been identified by the PMF if they had major impact on the air quality in the Delhi NCR. They do not have major impact, because policies have restricted their usage in recent years including 2022. The only noticeable diesel emissions impacting the receptor site are mixed into the road construction factor and reach the receptor simultaneously with the evaporative emissions of freshly laid asphalt. This is now clarified in section 3.1 as follows:

The OVOC source fingerprint of the road construction factor matched the source fingerprint of asphalt mixture plants and asphalt paving (R=0.9, Li et al., 2020), while the hydrocarbon source fingerprint matched diesel-fuelled road construction vehicles (R=0.6, Che et al., 2023).

Line 650-651: Authors state that "all" previous studies from the region have attributed PM to BB or fossil-fuel burning, and that we need to look beyond these sources. While I agree that a larger set of sources need to be identified, I think there is already some work done on this front. Kumar et al. 2022 ACP https://acp.copernicus.org/articles/22/7739/2022/acp-22-7739-2022.pdf

Kumar et al. 2022 identified "cooking-related OA using EESI-TOF PMF analysis, but sadly did not include this source in the final pie charts of their paper, because their AMS-PMF analysis could not find this particular source, while ESI data was only reported in counts per second and not quantitatively. Instead, the paper reported three primary factor HOA, BBOA-1 and BBOA-2 and one aged biomass burning factor in addition to three SOA factors without naming the activities responsible for the formation of BBOA-1 and BBOA-2. The paper that reported results that most closely relate to the results of our study would be Cash et al. 2021. We have added a discussion of the Cash et al results to these lines

"This new approach of combining VOC tracers with PM measurements provides great potential for improved source apportionment in complex emission environments, at a level of detail that is more meaningful than just attributing emissions to biomass burning or fossil-fuel burning, which has been the case in all previous studies from the region to date. Previously in Delhi-MCR region, Kumar et al. 2022 identified "cooking-related OA using EESI-TOF analysis but due to analytical limitations, the paper only reported quantitative pie charts for three primary factors, namely HOA, BBOA-1 and BBOA-2, without naming the activities responsible for the formation of BBOA-1 and BBOA-2. One of the more comprehensive AMS based studies (Cash et al., 2021) spanning pre-monsoon, monsoon and post monsoon season of the year 2018 identified three different primary biomass burning factors, namely cooking organic aerosol (6% of $PM_1$), solid fuel organic aerosol ($\leq$11% of $PM_1$), and semi-volatility biomass burning organic aerosol ($\leq$13% of $PM_1$), that broadly appear to correspond to our solid fuel-based cooking (4% of $PM_{10}$), residential heating and waste burning (23% of $PM_{10}$), and paddy residue burning (25% of $PM_{10}$) factors. However, the study failed to name and attribute two of these three factors in policy relevant ways, could not identify the significant contribution of coarse mode fly ash to the total aerosol burden, and also was unable to distinguish between different fossil-fuel related sources."

Figure 5: I notice that road construction and solvent factors show opposing temporal trends. Road construction peaks in the afternoon while solvents are higher during early morning or night hours. The authors state in lines 425-426 that the solvents contribute the most to the VOC burden at night. Given that both these sources are evaporative in nature, how could they show opposing temporal trends? Are there any specific sources of solvents in Delhi that are prominent during nighttime?

The road construction factor primarily involves evaporative emissions released during degassing from the road surface, emissions are greatest when the asphalt has just been paved and hence peak during the hours when construction activity is more prevalent. Conversely, compounds associated with the solvent factor predominantly originate from an industrial point sources that appears to operate 24/7. They reach the receptor in episodic but intense plumes that are not accompanied by combustion tracers. Hence, we attribute this factor to the venting of chemicals from some industrial stacks. This type of activity results in the highest concentrations at night when emissions mix into a shallower nocturnal boundary layer. However, the factor also displays episodes with high concentrations during daytime. This indicates that daytime fluxes can actually be quite high and just mix into a larger volume. We looked for some specific types of industrial units located 1) SW of the receptor with the highest source strength after midnight and 2) NE of the receptor with the highest source strength in the evening before midnight. The best source match we found for this type of source was collected from a plot situated opposite a polymer manufacturing unit and next to a pet food manufacturer in an industrial area at Jahangir Puri (R=0.7) NE of the receptor. We have updated the text in Section 3.1 and lines 425ff to make the difference clearer:

"The OVOC source fingerprint of the road construction factor matched the source fingerprint of asphalt mixture plants and asphalt paving (R=0.9, Li et al., 2020), while the hydrocarbon source fingerprint matched diesel-fuelled road construction vehicles (R=0.6, Che et al., 2023). The factors identified as solvent usage and

evaporative emissions matched ambient air grab samples collected from an industrial area at Jahangir Puri (R=0.7), and Dhobighat at Akshar Dham (R=0.5) in this study."

"These compounds point towards stack venting of VOCs from chemical-, food-, or pharmaceutical industries or polymer manufacturing as likely sources of these emissions (Hodgson et al., 2000, Villberg et al., 2001, Jankowski et al., 2017, Gao et al., 2019). This assessment is broadly confirmed by the fact that the best match (R=0.7) for this source was collected from a plot situated opposite a polymer manufacturing unit and next to a pet food manufacturer in an industrial area at Jahangir Puri NE of the receptor site."

One can also check the temporal trends in PCBTF, Texanol and p-dichlorobenzene, D4- and D5-siloxane that are known tracers of VCP sources. Some of these can be measured with PTR-ToF.

Thank you for this comment. We appreciate that Volatile Chemical Products (VCP) have emerged as an important source in recent studies conducted in western countries, in which emissions from industrial and other sources have been regulated to a greater extent and VCP emissions from cosmetic and perfume usage and that of sanitation products have transitioned to becoming major sources of some VOCs in the urban environment. In India, however, these products are expensive and can only be afforded by a limited subset of the urban population. The vast majority of the population, even in a city like Delhi, struggle to meet their daily food needs and expenses on education and healthcare, which likely has kept VCPS from being a major source so far. Hence it is not surprising that this factor did not appear in the 11-factor PMF solution. Except for dichlorobenzene none of the compounds passed our quality control filter, which means either the signal at the m/z was not above the detection limit or the compound presence could not be confirmed via isotopic peaks of the correct height for the isotopes of the compound. At our site, the dominant dichlorobenzene sources appear to be industrial in nature.

The authors should more clearly discuss how they calculated the total VOC mass in the paper. This is important because the fractions of other measured species are drawn from the total, and this can introduce significant bias in the conclusions regarding source contributions if the total VOC mass is not comprehensive enough.

We have added a few sentences to clarify this point:

"The total VOC mass was included as a weak species and was calculated as the sum of the mass of the individual 111 VOC species included in the PMF. Overall, the 111 VOC species included in our analysis and their isotopic peaks explained 86% of the VOC mass detected during our study period. The remaining 119 m/z that accounted for 14% of the detected VOC mass could not be included in our PMF analysis mostly because signals were below the detection limit for close to 50% of the observation period, or because compound identity could not be confirmed via isotopic peaks."

The chemical profiles shown in Figure 3 run up to C10H16 and there is some additional discussion in the paper about IVOCs. However, sources such as road construction emit minimally in the VOC space, and more in the IVOC and SVOC space. The authors should discuss how they prevented biases from creeping into their conclusions.

The main rationale behind choosing these compounds shown in Figure 3 is, that the normalized height of the bar displaying that compound should be at least 0.1x the height of the tallest peak in at least one of the factor profiles. Most of the IVOCs did not meet the inclusion criteria for Figure 3. The figure serves to depict the chemical fingerprints of all the factors and very low bar heights are invisible on the y-axis, while the compound formula will clutter the x-axis of the figure and will make it hard to read. We have now clarified this as follows:

"Figure 3 shows the source profile of the eleven factors that our PMF analyses resolved. Out of the 111 VOCs only those whose normalized source contribution exceeded 0.1 when divided by the most abundant compound in the same source profile in at least one of the sources, were included in the figure."

Fresh asphalt does have a very characteristic VOC signature in the OVOCs space in the form of a very distinct double peak at C3H6O and C4H8O that is not accompanied by methanol peaks as is usually the case in a solvent factor, and neither accompanied by the furanes, aldehydes and organic acids that are usually seen in biomass burning source fingerprints. This makes the identification so easy and clear. The factor identity has been confirmed via cross correlation analysis as follows:

"The OVOC source fingerprint of the road construction factor matched the source fingerprint of asphalt mixture plants and asphalt paving (R=0.9, Li et al., 2020), while the hydrocarbon source fingerprint matched diesel-fuelled road construction vehicles (R=0.6, Che et al., 2023)."

Also, there should be at least some discussion in the paper about the inlet system used upstream of the PTR-TOF as this can prove crucial in the detection of many species (lines 132-133).

A detailed discussion about this as well as the inlet system is given in the companion paper (Mishra et al., 2024 10.5194/egusphere-2024-500) and now also mentioned and included in the revised MS as follows:

"It is worth mentioning again that as a significant improvement over other previous PTR-TOF-MS deployments in Delhi, the inlet system of the instrument used in this work was designed for sampling and detection of low-volatility compounds with the extended volatility range technology (Piel et al., 2021). The inlet system of the instrument as well as the ionization chamber is fully built into a heated chamber and the inlet capillary is further fed through a heated hose to ensure there are no "cold" spots for condensation. The entire inlet system is made of inert material (e.g. PEEK or siliconert treated steel capillaries to keep surface effects minimal. Further the overall inlet residence time was less than 3 seconds, throughout the campaign."

Piel, F., Müller, M., Winkler, K., Skytte af Sätra, J., and Wisthaler, A.: Introducing the extended volatility range proton-transfer-reaction mass spectrometer (EVR PTR-MS), Atmos. Meas. Tech., 14, 1355–1363, https://doi.org/10.5194/amt-14-1355-2021, 2021.

Furthermore:
Lines 182-184: The "pulling up" and "pulling down" should be briefly explained. It sounds vague in its current form.
We regret the confusion, as "pulling up" and "pulling down" are normally common terms used in PMF discussions. We have added the following text to clarify:
"The rotational ambiguity can be reduced using this option with the aid of prior knowledge by encouraging the model to minimize (pull down) or maximize (pull up) the total mass assigned to specific hourly observations or compounds in source profiles as much as possible within a pre-defined permissible penalty on Q."

Lines 187-188: It is quite amazing that the bootstrap found all 100% of the runs stable and well-mapped to the base solution. In principle, this may suggest that your dataset yields only one solution which is super robust. Is this what you are saying?
The reviewer has interpreted this correctly. 100% stable bootstrap solutions for the constraint run indeed indicate that with the help of the constraints a robust solution has been achieved for this particular dataset.

I acknowledge citations, but in lines 180-187, I recommend briefly describing the rationale behind application of different constraints to help the reader assess.
Thank you for highlighting this important aspect. As can be seen from our list of constraints this has been primarily accomplished by pulling down some night-time plumes from combustion sources that the model's base model run had left behind in the BVOC and photochemistry factors. Stopping the model from mixing some of the combustion emissions into non-combustion sources turns out to be an extremely efficient way to force the model to resolve the combustion sources properly, and prevent factor swapping. The solution can be further refined by identifying one major characteristic plume per source and pulling it up. The selected plumes should originate from one dominant source only, should have plume enhancement ratios that matches source samples as closely as possible, and should be among the strongest plumes attributed to the source in the base run. Pulling up as little as one such plume for each of the anthropogenic sources, minimizes the factor swapping between similar sources
We have added the following details:
"The rotational ambiguity can be reduced using this option with the aid of prior knowledge by encouraging the model to minimize (pull down) or maximize (pull up) the total mass assigned to specific hourly observations or compounds in source profiles as much as possible within a pre-defined permissible penalty on Q. The primary problem of the base run solutions is that night-time biomass burning plumes contaminate both the biogenic and the photochemical factor. To minimize this in our constrained run, we have pulled down primary emissions (acetonitrile, toluene, C8 aromatics, and C9 aromatics) in the biogenic and photochemical factors. We also pulled down the top-7 strongest nighttime plumes contaminating the biogenic and photochemical factors. In addition, we pulled up the highest plume event for all the anthropogenic emission-related factors as detailed in Table S2. The overall penalty to Q (the object function) was 4.9 %, which is within the recommended limit of 5 % (Norris et al., 2014; Rizzo & Scheff, 2007)."

Lines 229-234: The comparisons stated here are very on point, which is great. But it is not clear how contributions from heavy vehicles, e.g. road construction vehicles, were separated from other diesel-based sources, such as transport trucks.
We appreciate the referee's comment. As also mentioned in the above replies, the majority of heavy-duty vehicles in Delhi have been switched to CNG. This includes all commercial diesel vehicles such as trucks, buses and taxis. Even private diesel vehicles that are older than 10 years are not permitted to ply in the city. This is why the only diesel emissions we see in the PMF are emissions that reach the receptor site during daytime simultaneously with compounds that are diagnostic of fresh asphalt paving. It appears that construction

machinery is the dominant HDV class that has not yet completely been converted. The revised text now reads as follows:

"The factors identified as CNG (R=1.0), petrol 4-wheelers (R=0.9), and petrol 2-wheelers (R=0.6) matched tailpipe emissions of the respective vehicle types and fuels (Hakkim et al., 2021). The petrol 4-wheelers (R=0.9), and petrol 2-wheelers (R=0.7) also matched traffic junction grab samples from Delhi (Chandra et al., 2018). The OVOC source fingerprint of the road construction factor matched the source fingerprint of asphalt mixture plants and asphalt paving (R=0.9, Li et al., 2020), while the hydrocarbon source fingerprint matched diesel-fuelled road construction vehicles (R=0.6, Che et al., 2023)."

I recommend to put some correlation plots in the SI that compare the chemical profiles of the source factors obtained in this study with the sources from literature that are discussed here.

We thank the referee for the suggestion. We have now modified the text (see below) and added the R values to the main text. but don't think correlation plots are necessary since the source profiles as presented in the revised version can already be visually compared with the help of Figure 3.,

"The PMF factor profile matched best against source samples collected from burning paddy fields (R=0.6, Kumar et al., 2020) for the paddy residue burning factor. The cooking factor matched emissions from a cow-dung-fired traditional stove called angithi (R=0.7, Fleming et al., 2018). The residential heating & waste burning factor had a source fingerprint matching emission from leaf litter burning, (R=0.7, Chaudhary et al., 2022), waste burning (R=0.7, Sharma et al., 2022), and cooking on a chulha fired with a mixture of firewood and cow dung (R=0.9, Fleming et al., 2018). The factors identified as CNG (R=1.0), petrol 4-wheelers (R=0.9), and petrol 2-wheelers (R=0.6) matched tailpipe emissions of the respective vehicle types and fuels (Hakkim et al., 2021). The petrol 4-wheelers (R=0.9), and petrol 2-wheelers (R=0.7) also matched traffic junction grab samples from Delhi (Chandra et al., 2018). The OVOC source fingerprint of the road construction factor matched the source fingerprint of asphalt mixture plants and asphalt paving (R=0.9, Li et al., 2020), while the hydrocarbon source fingerprint matched diesel-fuelled road construction vehicles (R=0.6, Che et al., 2023). The factors identified as solvent usage and evaporative emissions matched ambient air grab samples collected from an industrial area at Jahangir Puri (R=0.7), and Dhobighat at Akshar Dham (R=0.5) in this study. The factor identified as industrial emissions showed the greatest similarity to ambient air grab samples from the vicinity of the Okhla waste-to-energy plant (R=0.8), Gurugram (R=0.7) and Faridabad (R=0.8) industrial area. The biogenic factor showed the greatest similarity to leaf wounding compounds released from Populus tremula (R=0.8, Portillo-Estrada et al., 2015) as well as BVOC fluxes from Mangifera indica (R=0.4, Datta et al., 2021)."

Lines 252-253: As a reader, I was surprised to see a comparison with NW-IGP and Mohali. It was quite sudden and not consistent throughout the paper. This should be rephrased in a way that gives a reader some context on which regions are being compared and why.

We have rephrased this as follows:

"Figure 4 shows the relative contribution of different sources to the total pollution burden of VOCs, PM$_{2.5}$ and PM$_{10}$ at the receptor site. In the megacity of Delhi, transport sector sources contributed most (42±4 %) to the total VOC burden, while it contributed much less (only 24 %) to the total VOC burden in Mohali a suburban site 250 km North of Delhi during the same season (Singh et al., 2023). On the other hand, the contribution of paddy residue burning (6±2 %) and the summed residential sector emissions (17±3 % in Delhi and 18 % in Mohali) to the total VOC burden during post-monsoon season were similar at both sites."

Lines 262-270: Add error values to the average percentages to account for the variability in these fractions during the study period.

This text has been substantially simplified. Uncertainties have been added to the segments that were retained. However, uncertainties reflect the uncertainty of the PMF model imposed by the stability of the bootstrap runs, not the ambient variability. No uncertainty was added to the SOA formation potential because the uncertainty of the widely used SOAP factors has not been quantified so far. This prevents meaningful error propagation from VOC mass to SOAP.

"Figure 4 shows the relative contribution of different sources to the total pollution burden of VOCs, PM$_{2.5}$ and PM$_{10}$ at the receptor site. In the megacity of Delhi, transport sector sources contributed most (42±4 %) to the total VOC burden, while it contributed much less (only 24 %) to the total VOC burden in Mohali a suburban site 250 km north of Delhi during the same season (Singh et al., 2023). On the other hand, the contribution of paddy residue burning (6±2 %) and the summed residential sector emissions (17±3 % in Delhi and 18 % in Mohali) to the total VOC burden during post-monsoon season were similar at both sites. The contribution of the different factors to the SOA formation potential (Fig. 4e), stands in stark contrast to their contribution to primary particulate matter emissions. SOA formation potential was dominated by the transport sector (54 %) while direct

PM$_{10}$ (52±8%) and PM$_{2.5}$ (48±12%) emissions were dominated by different biomass burning sources (Fig. 4 b & c). CNG-fuelled vehicles also contribute significantly to the PM$_{10}$ (15±3 %) and PM$_{2.5}$ (11±3 %) burden."

Line 284: I am not sure whether a correlation R of 0.5 could be considered significant.
We have deleted this.

Line 288: 0.027 and 0.047 are quite small values. What is your error bound on these numbers?
We have added the uncertainties of the slope. Numbers appear to be small because fire counts are very high (several thousands) but the slopes and resulting PM enhancements are very significant:
"Figure S6 shows that the PM$_{2.5}$ and PM$_{10}$ mass loadings at the receptor site increased by 0.027±0.006 and 0.047±0.01 µg m$^{-3}$ respectively for each additional fire count within the 24-hour fetch region whenever the trajectories are arriving through north-west and south-west region. It is very interesting to note that the incremental increase in PM$_{2.5}$ and PM$_{10}$ mass loadings for each additional fire count were almost four times higher than the former regions when the trajectory fetch region was south-east with 0.11±0.01 and 0.19±0.02 µg m$^{-3}$, respectively, likely because the complete burns of entire fields (Figure S7) that are prominent in Punjab can be more easily identified as a fire activity with satellite-based detection (Liu et al., 2019, 2020), while the partial burns (Figure S8) that are more prevalent in the eastern IGP and in Haryana have larger omission errors (Liu et al., 2019, 2020)."

Figure 5: The increase in NOx in petrol 2W panel during morning commute hours is not reflected in 2W or 4W factors. Does this make sense? Also why are the 2-wheeler petrol vehicle factor contributions high throughout the night and drop near the morning commute hours? I would imagine the 2W vehicles on the road to decrease substantially during the night.
Our PMF has a 4W dominated petrol vehicle factor, containing emissions from the immediate vicinity of the receptor (i.e. within Central Delhi), that have a plume age on the timescale of minutes based on the highest correlation of the factor being with NO emissions. The 2W dominated petrol vehicle factor is aged shows a much higher correlation with NO$_2$ than with NO and emissions appear to occur in the rural hinterland and outskirts. Their transport time appears to be on the scale of hours. The 2-wheeler factor has been identified as such primarily based on the benzene to toluene ratio which differs between 2-wheelers and 4-wheelers and is better preserved longer during photochemical aging when compared to the emission ratios of C8 and C9 aromatics. However, it is clear from the presence of OVOCs in the source profile and the low correlation of the factor time series with NO that most plumes in the 2W factor are aged, and hence are expected to reach the receptor several hours after the peak evening traffic. The morning peak in NOx coincides primarily with the peak in cooking emissions and is not triggered by either of these factors. We have revised the text of both sections to make this distinction clearer:
"Figure 4 shows petrol 4-wheeler contributed 20 %, 25 %, and 30 % to the VOC mass loading, OFP, and SOAP, respectively. The source fingerprint of this source matched tailpipe emissions of petrol-fuelled 4-wheelers (Hakkim et al., 2021) and is characterized, in descending rank of contribution, by C8-aromatics, toluene, C9-aromatics (C$_9$H$_{12}$), benzene, butene + methyl tert-butyl ether (MTBE) fragment, propyne, propene, methanol and C2-substituted xylenes + C4-substituted benzenes (C$_{10}$H$_{14}$). Figure 5 shows that emissions peak in the evening between 7 pm and midnight with average VOC mass loadings >70 µg m$^{-3}$ and reach the receptor site from most wind directions. Emissions are strongly correlated with NO (R=0.8), CO (R=0.7), and CO$_2$ (R=0.7) indicating the receptor site is impacted by fresh combustion emissions from this source and the atmospheric age of most plumes is on the timescale of minutes."

"Figure 4 shows petrol 2-wheeler contributed 14 %, 12 %, and 20 % to the VOC mass loading, OFP, and SOAP respectively. The source fingerprint of this source matched tailpipe emissions of petrol-based 2-wheelers (Hakkim et al., 2021) and are characterized, in descending rank of contribution, by toluene, acetone + propanal, C-8 aromatic compounds, acetic acid (C$_2$H$_4$O$_2$), propyne (C$_3$H$_4$), methanol (CH$_3$OH), benzene (C$_6$H$_6$), the MTBE fragment and C-9 aromatics (C$_9$H$_{12}$). A key difference of the petrol 2-wheeler source profile in comparison to the petrol 4-wheeler source profile is the lower benzene to toluene ratio, which is supported by the GC-FID analysis of tailpipe exhaust (Kumar et al., 2020). Figure 5 shows that emissions peak in the evening between 8 pm and 10 pm with average VOC mass loadings >50 µg m$^{-3}$and reach the receptor site from most wind directions. Emissions are strongly correlated with NO$_x$ (R=0.6), CO (R=0.6) and CO$_2$ (R=0.7), but have a lower correlation with NO (R=0.5) (Table S5), and a larger contribution of oxygenated compounds to the source profile, indicating that the emissions have been photochemically aged. This suggests that contrary to 4-wheeler plumes which originate from the immediate vicinity of the site in central Delhi (Figure S1), 2-wheeler plumes reach the receptor after prolonged transport from more distant rural and suburban areas on the outskirts of the city. In such areas, people often favour two-wheelers over four-wheelers."

Line 326: 3.2.2 Title: By waste disposal, do the authors mean waste burning? These can be very different things with different mechanisms of emissions if combustion is not involved in one versus the other.
We appreciate the referee's helpful comments and have changed the names to heating and waste-burning

Line 354: BB emissions are attributed to solid fuel-based cooking and a cow dung-fired traditional stove is discussed. These measurements were made at IMD Lodhi Road, which appears to be a highly urbanized area. How do the authors justify BB-based cooking activities near such location? Is regional transport important for fresh emissions?
In a mega city like Delhi, there is a socio-economic spectrum of society. Among its residents are many who continue to rely on traditional solid fuels for their cooking and heating needs due to financial constraints. It is also important to remember that the air shed is much larger than just Central Delhi and this is not a PMF factor containing fresh emissions as can be seen by the low R=0.1 with NO. Solid fuel usage is very much prevalent in the villages of Haryana and Uttar Pradesh which are located within a radius of less than 60 km from the receptor. We have modified the text as follows:
The activity peaks from 8 am to noon time, with a secondary peak in the early evening hours and persists throughout monsoon and post-monsoon season. Emissions reaching the receptor site show no correlation with NO (R=0.1) indicating plumes are not fresh.

Furthermore, cooking's contribution to PM10 is discussed, which is understandably low. However, what about PM2.5 that can be formed from the oxidation of gas-phase cooking emissions?
Yes cooking appears to be more of a VOC than a PM source. However, for this factor the percentage contribution to the SOA formation potential is lower than its percentage contribution to the VOC burden because most VOCs emitted are small OVOCs with limited SOA formation potential. The volatility oxidation space plot also shows very little evidence of first- and second- generation oxidation products progressing from the VOC into the IVOC region. The text has been modified to include this information:
The cooking factor is a daytime factor and explains 10 % of the total VOC mass loading, 10% and 8 % to the ozone and SOA formation potential (Fig. 4) but only a negligible share of the total $PM_{10}$ ($\leq$4 %) burden. The volatility oxidation space plot (Figure S9) also shows very little evidence of IVOC oxidation products that could partition into the aerosol phase.

Minor points:
Line 86: "at" Lodhi Road.done
Line 190: extra "T" at the start.done
Line 264: "Direct", do you mean "Primary" ? yes, however these details were deleted in response to a comment of reviewer 1
Line 642: ''at this time of the year…" Which time of the year? This is written casually. Revised to:
While several recent efforts in some sectors (e.g. residential biofuel and cooking) appear to have yielded emission reduction benefits, the narrative to blame the post-monsoon pollution exclusively on the more visible sources (e.g. paddy residue burning), needs to be corrected so other sources are also mitigated.
Figure 3: Remove the word "PMF" from all figure legends. We prefer to retain the legend in all panels. The figure is easier to comprehend when legends of all individual panels are complete.
Figure 5: Add y-axis labels to the wind rose plots. These are conditional probability roses showing a probability between 0 and 1. We now explain this more clearly in the figure legend
The polar plots (right column) depict the conditional probability of a factor having a mass contribution above the 75th percentile of the dataset during a certain hour of the day between midnight (centre of rose) and 23:00 local time (outside of rose) from a certain wind direction. This probability is determined by dividing the number of observations above the 75th percentile by the total number of measurements in each bin.

---

## Author Response (AR2)

The Editor,
Atmospheric Chemistry and Physics

Dear Professor Laskin,

Thank you for accepting our paper. We have implemented the technical corrections suggested by reviewer 2 and the editorial office as detailed below.

Yours sincerely,
Baerbel Sinha, on behalf of co-authors

The following comment is from the editorial office
Regarding your Figure 1d - with the next file upload, please add the copyright icon as follows: © Google Earth.
done

Additional private note (visible to authors and reviewers only):
list of technical corrections noted by Reviewer 2:

l.23: add a space between "," and "15"

done l.154: close the ) please / Replace "Further," by "Furthermore," done l.215-216: "industrial OVOC emissions in the 7-factor solution" → the industrial factor is resolved in the 10-factors solution, right?

Response "solvent use and evaporative emissions" at the receptor site originate from an industrial point source that appears to be engaged in the stack venting of VOCs. We have changed the text to:
"Until the PMF opens distinct factors for the OVOC emissions due to industrial solvent usage and stack venting in the 7-factor solution, the partitioning between paddy residue burning and heating and waste burning $PM_{2.5}$ and $PM_{10}$ emissions in the model remains unstable, because these sources with their strong OVOC emissions are most agreeable to accommodating additional OVOC sources in their fingerprint at the expense of explaining the $PM_{2.5}$ and $PM_{10}$ emissions."

l.243: add a space between "ci" and "is"

done

Figure 6: Title is unclear, change to something like "Contribution of PMF factors to VOC species for which different forms of biomass burning contribute the highest percentage share of the atmospheric burden in Delhi". Same for Figure 7 & 8 & 9

We have taken the editor's comments into account and have revised the titles for figures 6, 7, 8, and 9 as per the suggestions. They now read as follows:

Figure 6: Contribution of PMF factors to VOC species for which different forms of biomass burning contribute the highest percentage share of the atmospheric burden in Delhi.

Figure 7: Contribution of PMF factors to VOC species for which the transport sector contributes the highest percentage share of the atmospheric burden in Delhi.

Figure 8: Contribution of PMF factors to VOC species for which the industries, solvent usage, photochemistry, or biogenic sources contribute the highest percentage share of the atmospheric burden in Delhi.

Figure 9: Contribution of PMF factors to VOC species for which the road construction contributes the highest percentage share of the atmospheric burden in Delhi.

l.478: C4H8 corresponds to butene, not butane.

We corrected the typo.

l. 485: repetition of "source"

This has been changed and have deleted the repetition

"The source fingerprint of this matched tailpipe emissions …"

l. 532: replace "to" by "of" and add a space before.

Changed l. 654-655: "The signal at m/z41.035 can potentially be attributed to aC3H4 the 2-methyl-3-butene-2-ol fragment" add a space in "m/z 41.035" / m/z 41.035 (C3H4) is also a known fragment of isoprene (eg, Yuan et al 2017, doi.org/10.1021/acs.chemrev.7b00325) / please delete "a" before "C3H4"

We appreciate the editor's comments and now the revised sentence reads as follows,

"The signal at m/z 41.035 can potentially be attributed to $C_3H_4$ the 2-methyl-3-butene-2-ol fragment (Kim et al., 2010; Park et al., 2013) a known fragment of isoprene (Yuan et al., 2017)."

l. 726 & 794 & 802 & 808 at least: missing space between "PM2.5" and "&", please check whole text for other missing spaces.

We have changed all the missing spaces between "PM2.5" and "&", l.784-785: Secondary ammonium, nitrate and sulfate can also originate from vehicle exhaust oxidation of NH3, NO2 and SO2, especially NH3 can be emitted by gasoline and CNG vehicles (eg, Alanen et al 2017, Link et al 2017)

We thank the reviewer for this valuable suggestion however, the transport sector can not simply be added as an equally important source to this sentence. Hence, we had to extensively rephrase to accommodate mentioning it.

In this context it is important to note that while the contribution of the transport sector to the NH3 burden in relatively "clean" urban environments in Europe and the US is substantial, the same can sadly not be said about the Indian scenario, where it contributes <1% to the NH3 emission budget. In India, ammonia fertilizers are heavily subsidized and overused. Due to this NH3 originates primarily from fertilizer manufacturing and fertilizer usage, as well as poor sewage and manure management practises.

Along similar lines the contribution of the transport sector to $SO_2$ emissions happens to be miniscule (<1%) in comparison to the contribution of power generation (coal fired generation units are responsible for 60% of the Indian $SO_2$ emissions because Indian power plant lacking flue gas desulfurization devices) The transport sector on the other hand uses low sulphur BS VI = Euro VI fuel and has very low SO2 low emissions.

Only when it comes to NOx emissions, the transport sector is the second largest source (25% of the total) after power generation (40%) while industries are the third largest source (21%). However, during the warmer part of the year ammonium nitrate tends to partition into the gas phase and is not thermodynamically favoured. Several past studies have shown that during the time of the year covered by our analysis most of the secondary inorganic aerosol is ammonium sulphate, which is why we only highlight power generation in this sentence. The revised text now reads as follows:

"Power generation is believed to be the dominant source of secondary sulfate aerosol (Atabakhsh et al., 2023), which is the largest contributor to the secondary inorganic aerosol burden in monsoon season (Catch et al., 2021). It is hence likely, that much of our PMF residual can be attributed primarily to this source. While a portion of this residual, particularly during post monsoon season, may also be secondary ammonium nitrate, to which power generation, transport sector and industrial NOx and NH3 emissions contribute (Alanen et al., 2017; Link et al., 2017), ammonium nitrate formation is not thermodynamically favoured during the warm months of the year."

l. 795: change to "one of the world's most…" changed l.806: replace "stronger" by "strongest" done l. 854: close the " done

[revised manuscript text omitted]